# Small Resamples, Sharp Guarantees: Convergence Rates for Resampled Studentized Quantile Estimators

**Imon Banerjee**
Department of Industrial Engineering and Management Science
Northwestern University

**Sayak Chakrabarty**
Department of Computer Science
Northwestern University

## Abstract

The m-out-of-n bootstrap—proposed by Bickel et al. [1992]—approximates the distribution of a statistic by repeatedly drawing $m$ subsamples ($m \ll n$) without replacement from an original sample of size n; it is now routinely used for robust inference with heavy-tailed data, bandwidth selection, and other large-sample applications. Despite this broad applicability across econometrics, biostatistics, and machine-learning workflows, rigorous parameter-free guarantees for the soundness of the m-out-of-n bootstrap when estimating sample quantiles have remained elusive.

This paper establishes such guarantees by analysing the estimator of sample quantiles obtained from m-out-of-n resampling of a dataset of length n. We first prove a central limit theorem for a fully data-driven version of the estimator that holds under a mild moment condition and involves no unknown nuisance parameters. We then show that the moment assumption is essentially tight by constructing a counter-example in which the CLT fails. Strengthening the assumptions slightly, we derive an Edgeworth expansion that delivers exact convergence rates and, as a corollary, a Berry–Esséen bound on the bootstrap approximation error. Finally, we illustrate the scope of our results by obtaining parameter-free asymptotic distributions for practical statistics, including the quantiles for random walk MH, and rewards of ergodic MDP's, thereby demonstrating the usefulness of our theory in modern estimation and learning tasks.

## 1 Introduction

The bootstrap—first proposed by Efron [1979]—quickly became a cornerstone of non-parametric inference. Yet its *orthodox* form, which repeatedly resamples the full data set of size $n$, suffers in two key respects: (i) resampling the entire dataset lacks computational appeal when $n$ is large, and more importantly (ii) it is inconsistent for several common statistics under heavy-tailed or irregular models (see Example 1). To alleviate these drawbacks, Bickel et al. [1992] advocated drawing only $m < n$ observations at each resampling step—the so-called *m-out-of-n bootstrap*.

To set the stage, we introduce some notation. Let $X_1, X_2, \ldots, X_n$ be observations from either an univariate distribution $F$ or a real-valued Markov chain with stationary distribution $F$. Assume that $F$ has a unique $p$-th population quantile $\mu$ and a density $f$ that is positive and continuous in a neighborhood of $\mu$. Let $\bar{F}(t) = n^{-1} \sum_{i=1}^{n} \mathbf{1}\{X_i \leq t\}$ be the empirical distribution of $\{X_i\}$. For $m \leq n$, draw an i.i.d. sample $X_1^*, \ldots, X_m^*$ of size $m$ from $\bar{F}$. Let $F_n^{(m)}(t) = m^{-1} \sum_{i=1}^{m} \mathbf{1}\{X_i^* \leq t\}$

39th Conference on Neural Information Processing Systems (NeurIPS 2025).

be the empirical distribution function of these resampled points. The m-out-of-n bootstrap estimator of the $p$-th quantile is then defined by $\hat{\mu}_m^{(boot)} = \inf\{t : F_n^{(m)}(t) \geq p\}$.

We denote by $\hat{\mu}_n$ the usual sample estimator of $\mu$ from the original data, and we write $\hat{\mu}_n^{(boot)}$ for the m-out-of-n bootstrap estimator of $\hat{\mu}_n$. Let $\hat{\sigma}_n$ be an estimator of $\text{Var}\big(\sqrt{n}(\hat{\mu}_n - \mu)\big)$. The Studentized estimator of $\mu$ is then $T_n = \sqrt{n}(\hat{\mu}_n - \mu)/\hat{\sigma}_n$.

In order to Studentize the m-out-of-n estimator, we analogously use the conditional variance of $\sqrt{m}(\hat{\mu}_m^{(boot)} - \hat{\mu}_n)$, given the original sample $\{X_i\}$, denoted by $\hat{\sigma}_m^{(boot)}$[1]. Consequently, we define

$$T_m^{(boot)}(\mu) := \frac{\sqrt{m}(\hat{\mu}_m^{(boot)} - \hat{\mu}_n)}{\hat{\sigma}_m^{(boot)}}.$$

This is the **Studentized** m-out-of-n bootstrap estimator of the sample quantile. As we remark below, this step is not optional in practice. However, Studentization introduces additional technical challenges. For instance, we show that, without further constraints on $F$, $\sigma_m^{(boot)}$ can tend to infinity even if $\hat{\mu}_m^{(boot)}$ itself is a consistent estimator of $\mu$.

At this point, we make note that we are looking at non-parametric or "model free" bootstrap, and the standard results on maximum likelihood does not apply to our case. Instead, the approach we take relies on empirical process theory [Van Der Vaart and Wellner, 2023]. Therefore, they are generalisable to a wide class of distribution functions.

Empirical studies confirmed the practical values of the m-out-of-n bootstrap, but also revealed new pathologies: Andrews and Guggenberger [2010] showed size distortions for subsampling-based tests, and Simar and Wilson [2011] demonstrated that the naive (full-sample) bootstrap is inconsistent for DEA estimators whereas the m-out-of-n variant is not. Despite these successes, the m-out-of-n bootstrap's fundamental properties remain only partially understood.

Initially, our work was motivated by the need for bootstrap based distribution free tests for practical tasks on Markov chains like Markov Chain Monte Carlo (MCMC) and Reinforcement Learning (RL). However, we quickly realised that this problem was not satisfactorily studied even when the data was i.i.d. We present two out of numerous glaring examples: one significant discrepancy seems to be in Cheung and Lee [2005], which gives an incorrect variance formula for the m-out-of-n estimator of sample quantiles without derivations or citations. Another work Gribkova and Helmers [2007] derive an Edgeworth expansion for Studentized trimmed means in the non-classical $m \gg n$ regime, offering theoretical insights but leaving unanswered the setting when $m \ll m$, which motivates subsampling.

Our work addresses these practically relevant open questions that were left unanswered in previous studies by supplying corrected rates and Edgeworth expansions tailored to the practically relevant $m \ll m$ regime. More details are given in Section 3.

Furthermore, we also show that the m-out-of-n bootstrap does not seem to suffer from the usual pitfalls of orthodox bootstrap pointed out in a seminal PTRF paper by Hall and Martin [1988b, 1991] (see section 4.2 for full details), thus acquitting bootstrap in practical utility. We now briefly mention our contributions.

- **Consistent Studentization under minimal moments.** We establish that the plug-in variance estimator $\hat{\sigma}_m^{(boot)}$ is *consistent* whenever $E|X_1|^\alpha < \infty$ for some $\alpha > 0$ for both i.i.d (Theorem 1) and Markovian (Theorem 3) data. Proposition 1 shows the moment condition is essentially sharp: heavy-tailed distributions preclude a central-limit theorem for the Studentized median. We then demonstrate the applicability of our results by deriving parameter free asymptotic distributions for common types of problems such as the median of random walk Metropolis Hastings, or the median reward in offline MDP's.

- **Exact Edgeworth expansion at classical scaling.** Exploiting a binomial representation, we obtain the first correct $\mathcal{O}(m^{-1/2})$ Edgeworth expansion for the Studentized m-out-of-n quantile (Theorem 2). In contrast to orthodox bootstrap [Hall and Martin, 1991], the expansion is symmetric; reflecting the intrinsic "smoothing" induced by subsampling—and immediately yields a Berry–Esseen bound (Corollary 1).

---

[1] $\hat{\sigma}_m^{(boot)}$ admits a closed form representation; see eq. (B.2).

- **Clarification of prior discrepancies.** Lemmas 6–7 supply a corrected variance formula, by correcting the algebraic sources of error in Cheung and Lee [2005].

*The rest of the paper is organized as follows:* Section 2 outlines a comprehensive discussion of relevant research works. In section 3, we formally introduce the model and the relevant notations. In section 4, we provide our key theoretical results and the proof sketches for these, while the full proofs have been deferred till the appendix due to lack of space. Section 5 contains various applications for our results, and finally, in Section 6, we conclude by mentioning the limitations and broader impact of our work.

## 2 Background and Related Research

The bootstrap, introduced by Efron [Efron, 1979, 1982], has become a cornerstone of modern statistical learning [Nakkiran et al., 2021, Modayil and Kuipers, 2021, Han and Liu, 2016]. Its intuitive appeal and computational simplicity have led to widespread use across diverse fields such as finance, economics, and biostatistics. In many practical settings, especially those involving complex models or nonstandard error bootstrap often creates better confidence intervals [Hall and Padmanabhan, 1992]. For example, in principal component analysis, bootstrap techniques have been employed for sparse PCA and consistency analysis [Rahoma et al., 2021, Babamoradi et al., 2013, Datta and Chakrabarty, 2024]. Time-series applications appear in [Ruiz and Pascual, 2002], whereas econometric and financial analyses are discussed in [Vinod, 1993, Gonçalves et al., 2023]. More recently, bootstrap-based methods have been explored in reinforcement learning [Ramprasad et al., 2023, Zhang et al., 2022, Faradonbeh et al., 2019, Banerjee, 2023, Zhang et al., 2023], Markov and controlled Markov chains [Borkar, 1991, Banerjee et al., 2025], Gaussian and non-Gaussian POMDP's [Krishnamurthy, 2016, Banerjee and Gurvich, 2025] and in clustering/classification [Jain and Moreau, 1987, Moreau and Jain, 1987, Makarychev and Chakrabarty, 2024].

A significant body of work has focused on traditional bootstrap methods for median and quantile estimation. For instance, Ghosh et al. [1984a] studied the convergence of bootstrap variance estimators for sample medians, while Bickel et al. [1992] employed Edgeworth expansions to obtain optimal rates of convergence. Incorporating density estimates for studentized quantiles was examined in Hall and Martin [1988a], Hall and Sheather [1988]. We refer the reader to various textbooks like Hall [2013], Dasgupta [2008] for theoretical treatments, and Singh and Xie [2008], Davison and Hinkley [1997], Freedman et al. [2007] for practical insights. Of particular interest is Liu [1988], which can be thought of a variant of our work for the sample mean rather than the median. Iterative self-alignment with implicit rewards from Direct Preference Optimization (DPO) uses "bootstrapped" preference data to refine model behavior without additional human labels [Yu et al., 2023, Wang et al., 2024, Chakrabarty and Pal, 2025, Zhang et al., 2024, Chakrabarty and Pal, 2024, Geigle et al., 2023, Banerjee et al., 2021]. Within symbolic AI, bootstrapping appears in inductive logic programming (ILP) and neurosymbolic systems as seen in [Natarajan et al., 2010, Bolonkin et al., 2024, Polikar, 2007, Simmons-Duffin, 2015]. This brings us to the following open question.

**Open question.** The paper resolves two open questions in the theory of the *Studentized m-out-of-n bootstrap.* **(i)** Can we provide a *single, parameter-free second-order theory*—variance consistency, a central-limit theorem, an exact $O(m^{-1/2})$ Edgeworth expansion, and a matching Berry–Esseen bound—for bootstrap quantiles under nothing stronger than a finite-moment assumption? **(ii)** Can we extend the analysis to regenerative Markov chains and furnish the inaugural bootstrap-based confidence guarantees for sequential data such as MCMC or RL trajectories?

## 3 Problem Statement

In order to formalise our results, we introduce some notation. Unconditional probability, expectation, and variances are denoted by $\mathbb{P}, \mathbb{E}, \text{Var}$ and their conditional counterparts given the sample are $\mathbb{P}_n, \mathbb{E}_n, \text{Var}_n$. $\sigma^2$ is the asymptotic variance of the estimator of $\hat{\mu}_n$. $\hat{\sigma}_n^2$, and $(\hat{\sigma}_n^{(m)})^2$ are its empirical and m-out-of-n bootstrap counterpart.

The $i$-th order statistic is denoted by $X_{(i)}$. We use $a_n, c(\alpha)$ for scaling constants, and $\xrightarrow{d}, \xrightarrow{p}, \xrightarrow{a.s.}$ for convergence in distribution, probability, and almost surely, respectively. Bachman–Landau notations [Cormen et al., 2022] for order terms are denoted by $\mathcal{O}, o, \mathcal{O}_p, o_p$. Finally, $\Phi$ and

$\phi$ are the standard normal CDF and PDF, and $\mathbb{H}_2(x) = x^2 - 1$ is the 2-nd Hermite polynomial. $\mathbb{C}$ denotes an universal constant whose meaning changes from line to line. $\mathbb{S}_L(\mu) := \{f(x) : \partial^{L-1}f(x)/\partial x^{L-1}\text{exists and is Lipschitz in a neighborhood of } \mu\}$ denotes the set of $L$-Sobolev smooth functions around $\mu$. For any set $A$, $\mathbb{S}_L(A) := \bigcap_{x \in A^\odot} \mathbb{S}_L(x)$ is the set of all functions which are $L$-Sobolev at all interior points of $A$ (denoted by $A^\odot$).

**Data-generating process.** As before, let $X_1, \ldots, X_n \overset{i.i.d}{\sim} F$ be real-valued with unique $p$-th population quantile $\mu$ and density $f$ positive and continuous near $\mu$. Denote the empirical cdf by $\bar{F}(t) = n^{-1}\sum_{i=1}^n \mathbf{1}\{X_i \le t\}$ and the full-sample quantile estimator by

$$\hat{\mu}_n = \inf\{t : \bar{F}(t) \ge p\}.$$

Choose $m \le n$ and resample $X_1^*, \ldots, X_m^* \overset{i.i.d}{\sim} \bar{F}$. The resample cdf is $F_n^{(m)}(t) = m^{-1}\sum_{i=1}^m \mathbf{1}\{X_i^* \le t\}$, and the bootstrap quantile is

$$\hat{\mu}_m^{(boot)} = \inf\{t : F_n^{(m)}(t) \ge p\}.$$

**Studentization.** Write $\hat{\sigma}_n^2 = \mathrm{Var}\big(\sqrt{n}(\hat{\mu}_n - \mu)\big)$ and let $\hat{\sigma}_m^{(boot)}$ be the conditional variance of $\sqrt{m}(\hat{\mu}_m^{(boot)} - \hat{\mu}_n)$ given $\{X_i\}$. Define the Studentized statistics

$$T_n = \frac{\sqrt{n}(\hat{\mu}_n - \mu)}{\hat{\sigma}_n}, \qquad T_m^{(boot)}(\mu) = \frac{\sqrt{m}\big(\hat{\mu}_m^{(boot)} - \hat{\mu}_n\big)}{\hat{\sigma}_m^{(boot)}}.$$

**Remark.** *Studentization is not optional in practice. Typically the un-Studentized estimator converges to a distribution dependent upon unknown parameters. Studentization is the only viable way to derive parameter free tests of sample statistics. Yet, literature regarding its theoretical guarantees is sparse.*

## 3.1 Goals

Our aim is to supply a *complete second–order theory* for Studentized m-out-of-n bootstrap quantiles under the weakest credible assumptions (finite $\alpha$-th moment, $\alpha > 0$). Four concrete targets guide the paper:

1. **Variance consistency and a CLT.** Demonstrate that the resampled variance converges to the population counterpart *and* the Studentized statistic tends to $\mathcal{N}(0, 1)$ whenever $m = o(n)$.

2. **Quantitative error control.** Obtain a one–term, correctly centred *Edgeworth expansion* whose remainder is $O_p(m^{-1/2})$, thereby delivering a uniform Berry–Esseen bound of the same order.

3. **Removal of legacy inconsistencies.** Correct the variance formula of Cheung and Lee [2005]. Furthermore, unlike some previous literature [Gribkova and Helmers, 2007], our new proofs work in the practically relevant regime $m \ll n$ (and even $m = o(n)$).

4. **Beyond i.i.d. data.** Extend variance consistency and the CLT to *regenerative Markov chains*, giving what appears to be the first bootstrap-based confidence guarantees for dependent sequences such as MCMC or reinforcement-learning trajectories.

# 4 Theoretical Results

Recall from the Section 3.1 that our first objective is to provide an asymptotic consistency theorem for the variance of the m-out-of-n bootstrap estimator of the sample quantile. We prove this assuming only the barebones—that $F \in \mathbb{S}_1(\mu)$, and that $E|X_1|^\alpha < \infty$ for some $\alpha > 0$. Later, we discuss the necessity of the assumption $E|X_1|^\alpha < \infty$. Since the techniques in this proof will be reused multiple times, we also provide a detailed proof sketch.

**Theorem 1.** *Let $\mu$ be the unique $p$-th quantile and $F \in \mathbb{S}_1(\mu)$. Furthermore, let $E|X_1|^\alpha < \infty$ for some $\alpha > 0$. Then for any $m = o(n)$, and $m, n \to \infty$.*

$$\big(\hat{\sigma}_m^{(boot)}\big)^2 \xrightarrow{a.s.} \frac{p(1-p)}{f^2(\mu)}.$$

*Furthermore,*

$$T_m^{(boot)}(\mu) = \frac{\sqrt{m}(\hat{\mu}_m^{(boot)} - \hat{\mu}_n)}{\hat{\sigma}_m^{(boot)}} \xrightarrow{d} \mathcal{N}(0,1).$$

**Sketch of proof:** **Step I.** Because the data have a finite $\alpha$-th moment, values larger than $n^{1/\alpha}$ occur only finitely many times (Borel–Cantelli). Consequently, the sample maximum and minimum are shown to be negligible.

$$\frac{|X_{(1)}| + |X_{(n)}|}{n^{1/\alpha}} \xrightarrow{\text{a.s.}} 0.$$

**Step II.** Consider the gap $\hat{\mu}_m^{(boot)} - \hat{\mu}_n$.

- *Medium deviations.* For $t$ up to $\left(\frac{1}{\alpha} + \frac{1}{2}\right)\sqrt{\log m}$, a Taylor expansion plus a Dvoretzky–Kiefer–Wolfowitz bound shows $\mathbb{P}_n\left(\sqrt{m}\,|\hat{\mu}_m^{(boot)} - \hat{\mu}_n| > t\right) = O(t^{-4})$.

- *Large deviations.* For $t$ above that threshold, a Hoeffding–style inequality gives a polynomial bound $O\left(m^{-(1/\alpha+1/2)(2+\delta)}\right)$.

These tail bounds imply the family $m(\hat{\mu}_m^{(boot)} - \hat{\mu}_n)^2$ is uniformly integrable and

$$\left(\hat{\sigma}_m^{(boot)}\right)^2 \xrightarrow{\text{a.s.}} \frac{q(1-q)}{f^2(\mu)}.$$

**Step III.** We then derive a non-Studentized CLT as follows

$$\sqrt{m}\left(\hat{\mu}_m^{(boot)} - \hat{\mu}_n\right) \xrightarrow{d} \mathcal{N}\left(0, \frac{q(1-q)}{f^2(\mu)}\right).$$

**Step IV.** Putting the variance consistency (Step 2) together with the CLT (Step 3) and invoking Slutsky's theorem, we arrive at

$$\frac{\sqrt{m}\left(\hat{\mu}_m^{(boot)} - \hat{\mu}_n\right)}{\hat{\sigma}_m^{(boot)}} \xrightarrow{d} \mathcal{N}(0,1),$$

which is the assertion of Theorem 1.

### 4.1 Importance of the Moment Condition

We briefly discuss the necessity of the condition $\mathbb{E}|X_1|^\alpha < \infty$ in Theorem 1. A close observation at the proof indicates that we only really need the condition in eq. (A.1). However, we found such conditions terse and un-illuminating.

On the other hand, the following counter-example shows that one cannot outright discard all tail conditions. Note that this does not imply that the condition $\mathbb{E}\left[|X_1|^\alpha\right] < \infty$ is a necessary condition, but rather demonstrates that Theorem 1 cannot be expected to hold for arbitrarily heavy-tailed distributions. Although this phenomenon has been observed via simulation [Sakov, 1998], we provide what is, to our knowledge, the first theoretical justification.

**Proposition 1.** *Let $m$ be fixed. Let $F(x)$ be the following class of distribution functions: For some large $C > e$,*

$$F(x) = \begin{cases} 1 - \frac{1}{2\log|x|} & \text{if } |x| > C, \\ G(x) & \text{if } |x| < C \end{cases}$$

*where $G$ is a monotonically increasing function such that $G(0) = 1/2$ and $G(x)$ has a positive derivative in a neighborhood of $0$. Then the variance of the $m$ out of $n$ bootstrap estimator for the median goes to $\infty$, i.e.*

$$\mathbb{E}_n(\hat{\mu}_m^{(boot)} - \hat{\mu}_n)^2 \xrightarrow{a.s.} \infty.$$

As can be seen from Proposition 1 (see also Ghosh et al. [1984b]), there exists regimes of heavy-tailed distributions for which the m-out-of-n (and orthodox) bootstrap estimator of the variance of the sample quantile is inconsistent. However, there are no current results which concretely characterises this regime (even when $m = n$). The citations of Ghosh et al. [1984b] are mostly in the form of remarks in either textbooks or tangentially related papers. It therefore remains an important open theoretical question, especially in the context of finance where heavy tailed distributions are ubiquitous Ibragimov et al. [2015]. However, it is beyond the scope of current work. We now move to establish Edgeworth expansions.

## 4.2 Convergence Rates Via Edgeworth Expansions

The following theorem proves an Edgeworth expansion of the Studentized m-out-of-n bootstrapped distribution of the quantile.

**Theorem 2.** *Assume that $\mu$ is the unique $p$-th quantile such that $F \in \mathbb{S}_2(\mu)$ and $f(\mu) > 0$. Furthermore, let $m = o(n^\lambda)$ for some $\lambda \in (0, 1)$. Then,*

$$\mathbb{P}_n \left( \frac{\sqrt{m}(\hat{\mu}_m^{(boot)} - \hat{\mu}_n)}{\hat{\sigma}_m^{(boot)}} \leq t \right) = \Phi(t) + \frac{(t\sigma)^2}{\sqrt{m}} f'(\mu)\phi \left( \frac{t\sigma f(\mu)}{\sqrt{p(1-p)}} \right) + \mathcal{O}_p \left( \frac{1}{m} \right)$$
$$+ \mathcal{O}_p(m^{-1/4} n^{-1/2}).$$

**Sketch of Proof:** After obtaining a correct form for the variance in Lemma 7, the strategy of our proof will be a combination of an Edgeworth expansion of the Binomial distribution (Proposition 12), followed by an appropriate Taylor series expansion of the CDF (Lemma 10). The proof will then be complete by calculating the rate of decay of the error terms in the Taylor series. See Section C for full details.

**Remark.** *Observe the assumptions on $F$, and $m$, are slightly stronger than those in Theorem 1. It is required to get a **rate** of convergence of the variance. Similar stronger assumptions are standard in literature [Hall and Martin, 1991].*

We discuss some implications of this theorem. Firstly, one can recover a Berry-Esseen type bound as an immediate consequence.

**Corollary 1.** *Under the conditions of Theorem 2*

$$\sup_{t \in (-\infty, \infty)} \left| \mathbb{P}_n \left( \frac{\sqrt{m}(\hat{\mu}_m^{(boot)} - \hat{\mu}_n)}{\hat{\sigma}_m^{(boot)}} \leq t \right) - \Phi(t) \right| = \mathcal{O}_p(m^{-1/2})$$

Next, by comparing Theorem 2 with the main Theorem of Hall and Martin [1991], one can see that the $\mathcal{O}_p(m^{-1/4} n^{-1/2})$ term in Theorem 2 replaces the $\mathcal{O}_p(n^{-3/4})$ in the main Theorem of Hall and Martin [1991] along with an extra $\mathcal{O}_p(1/m)$, recovering previous results. If $m$ is sufficiently large, the $\mathcal{O}_p(m^{-1/4} n^{-1/2})$ term dominates, whereas if $m$ is small, $\mathcal{O}_p(1/m)$ becomes the leading term, and we recover the usual form of Edgeworth expansions. However, setting $m$ too small also deteriorates the rate of decay of the error.

Furthermore, observe in the Edgeworth expansion that the second order term is an even polynomial, whereas for the orthodox bootstrap, the polynomial is neither odd nor even [Hall and Martin, 1991, Van der Vaart, 2000]. Therefore, unlike orthodox bootstrap (See Appendix Hall [2013]), m-out-of-n bootstrap can be used to create two sided parameter-free tests of the sample quantiles with an extra $\mathcal{O}_p(m^{-1})$ error as tradeoff, which is a significant development over orthodox bootstrap.

Finally, the optimal choice of $m$ seems to be $m = \mathcal{O}(n^{-1/3})$ which is known to be the minimax rate for estimating the sample median [Bickel and Sakov, 2008].

**On the choice of $m$:** We believe setting $m = cn^{1/3}$ for some universal constant $c$ would work well in practice. The precise choice of $c$ is somewhat ambiguous but

- for moderate sample sizes where $m > 30$, setting $c = 1$ typically performs well.
- For small $n$, a larger constant (i.e., $c > 1$) may be advisable.
- For large $n$, even $c \ll 1$ may suffice in achieving desirable performance.

## 4.3 Extensions to Regenerating Markov Chains

We now extend our results from the realm of i.i.d data to regenerating Markov chains. Following the theoretical development in the previous sections, let $X_1, \ldots, X_n$ be a sample from a Markov chain with initial distribution $\nu$, stationary distribution $F$, and empirical distribution $\bar{F}$. Then, $X_1^*, \ldots, X_m^*$ is said to be m-out-of-n bootstrap sample for $X_1, \ldots, X_n$ if $X_i^* \overset{i.i.d}{\sim} \bar{F}$, with other terms like $F_n^{(m)}$ and $\hat{\mu}_n^{(m)}$ defined accordingly. Despite its ubiquitousness in modern statistics, the theory of resampling for Markov chains is sparsely studied and there exists no widely accepted method (see Bertail and Clémençon [2006] and citations therein). In particular, no work attempted to study the effects of bootstrapping to create confidence intervals for statistics that are of interest in tasks like reinforcement learning or MCMC. In this section, provide what we believe are some of the first results in this field by extending the results on CLT from Section 4 to the realm of regenerating Markov chains, and then use it to recover a confidence interval for the median reward of an offline ergodic MDP. To that end, we introduce regenerating Markov chains.

The Nummelin splitting method [Nummelin, 1978, Athreya and Ney, 1978, Meyn and Tweedie, 2012] provides a way to recover all the regenerative properties of a general Harris Markov chain. In essence, the method enlarges the probability space so that one can create an artificial atom. To begin, we recall the definition of a regenerative (or atomic) chain [Meyn and Tweedie, 2012].

**Definition 1.** *A $\psi$-irreducible, aperiodic chain $X$ is said to be regenerative (or atomic) if there exists a measurable set $A$, called an atom, with $\psi(A) > 0$ such that for every pair $(x, y) \in A^2$, the transition kernels coincide, that is,*

$$P(x, \cdot) = P(y, \cdot).$$

*Intuitively, an atom is a subset of the state space on which the transition probabilities are identical. In the special case where the chain visits only finitely many states, any individual state or subset of states can serve as an atom.*

**Remark.** *$\psi$ and $\Psi$ are commonly used in Markov chain theory to denote $\psi$-irreducibility and $\Psi$-atoms, respectively [Bertail and Portier, 2019]. We omit formal definitions and refer the reader to standard references [Meyn and Tweedie, 2012, pp. 89, 103]. Intuitively, $\psi$-irreducibility generalizes the notion of irreducibility from finite to infinite state spaces, while $\Psi$-atoms are subsets where transitions behave homogeneously according to a measure $\Psi$—singletons in the finite-state case. A regenerating Markov chain is $\psi$-irreducible and possesses at least one recurrent $\Psi$-atom, with inter-arrival times called regeneration times. The moment conditions on these times determine the chain's ergodic properties.*

One now extends the sample space by introducing a sequence $(Y_n)_{n \in \mathbb{N}}$ of independent Bernoulli random variables with success probability $\delta$. This construction relies on a mixture representation of the transition kernel on a set $S$:

$$P(x, A) = \delta \, \Psi(A) + (1 - \delta) \frac{P(x, A) - \delta \, \Psi(A)}{1 - \delta},$$

where the first term is independent of the starting point. This independence is key since it guarantees regeneration when that component is selected. In other words, each time the chain visits $S$, we randomly reassign the transition probability $P$ as follows:

- If $X_n \in S$ and $Y_n = 1$ (which occurs with probability $\delta \in (0, 1)$), then the next state $X_{n+1}$ is generated according to the measure $\Psi$.

- If $X_n \in S$ and $Y_n = 0$ (with probability $1 - \delta$), then $X_{n+1}$ is drawn from the probability measure

$$(1 - \delta)^{-1} \Big( P(X_n, \cdot) - \delta \, \Psi(\cdot) \Big).$$

The resulting bivariate process $Z = (X_n, Y_n)_{n \in \mathbb{N}}$, is known as the *split chain*, and takes values in $E \times \{0, 1\}$ while itself being atomic, with the atom $A = S \times \{1\}$. We now define the *regeneration times* by setting

$$\tau_A = \tau_A(1) = \inf\{n \geq 1 : Z_n \in A\}, \text{ and for } j \geq 2, \tau_A(j) = \inf\{n > \tau_A(j - 1) : Z_n \in A\}.$$

It is well known that the split chain $Z$ inherits the stability and communication properties of the original chain $X$, such as aperiodicity and $\psi$-irreducibility. In particular, by the recurrence property,

the regeneration times have finite expectation.

$$\sup_{x \in A} \mathbb{E}_x[\tau_A] < \infty \quad \text{and} \quad \mathbb{E}_\nu[\tau_A] < \infty.$$

Regeneration theory [Meyn and Tweedie, 2012] shows that, given the sequence $(\tau_A(j))_{j \geq 1}$, the sample path can be divided into blocks (or cycles) defined by

$$B_j = (X_{\tau_A(j)+1}, \ldots, X_{\tau_A(j+1)}), \quad j \geq 1,$$

corresponding to successive visits to the regeneration set $A$. The strong Markov property then ensures that both the regeneration times and the blocks $\{B_j\}_{j \geq 1}$ form independent and identically distributed (i.i.d) sequences [Meyn and Tweedie, 2012, Chapter 13].

**Assumption 1.** *We impose the following conditions:*

1. *The chain $(X_n)_{n \in \mathbb{N}}$ is a positive Harris recurrent, aperiodic Markov chain with stationary distribution $F$, and initial measure $\nu$ on a compact state space, which we will assume to be $[0, 1]$ without losing generality. Furthermore, there exists a $\Psi$-small set $S$ such that the hitting time $\tau_S$ satisfies*

$$\sup_{x \in S} \mathbb{E}_x[\tau_S] < \infty \quad \text{and} \quad \mathbb{E}_\nu[\tau_S] < \infty.$$

2. *There exists a constant $\lambda > 0$ for which*

$$\mathbb{E}_A[\exp(\lambda \tau_A)] < \infty. \qquad \text{and} \qquad \mathbb{C}_\lambda := \frac{2 \mathbb{E}_A[\exp(\lambda \tau_A)]}{\lambda} > \frac{1}{2}.$$

3. *Let $\mu$ be the $p$-th quantile. Then, $F, \nu \in \mathbb{S}_1(\mu)$.*

**Remark.** *The assumption of compact state space is technical, and can be replaced with uniform ergodicity with significantly more tedium. Geometrically ergodic chains automatically satisfy the previous assumption (more details in the appendix. See also, Bertail and Portier [2019]).*

We now give main theorem of this section.

**Theorem 3.** *Let $X_1, \ldots, X_n$ be a sample from a Markov chain satisfying Assumption 1. Then, for all $m = o(n)$, the m-out-of-n bootstrap estimator of $\mu$ satisfies,*

$$T_m^{(boot)}(\mu) \xrightarrow{d} \mathcal{N}(0, 1)$$

**Sketch of Proof:** Using techniques on VC dimensions (see E.1), we first establish a Dvoretzky–Kiefer–Wolfowitz (DKW) inequality for regenerating Markov chains. This inequality is apparently new, and due to wide applicability in a variety of other tasks like nonparametric density estimation Sen [2018], change point detection Zou et al. [2014], etc. we present an informal version below, while relegating the final version to Section E. With this theorem, the rest of the proof proceeds on a strategically similar path to Theorem 1, but each step is now technically more intricate and requires careful coupling arguments. See Section F for full details.

**Proposition 2** (Informal version of Proposition 17)**.** *Let $X_1, \ldots, X_n$ be a Markov chain satisfying Assumption 1 with stationary distribution $F$, and define*

$$Z := \sup_{t \in [0,1]} |\bar{F}(t) - F(t)|$$

*Then, for all $x > 0$*

$$\mathbb{P}(Z > x) \leq \mathbb{C}^\star(\tau, \lambda) \exp\left(-\frac{\mathbb{C}(\tau, \lambda) n x^2}{\log n}\right) \tag{4.1}$$

*where, constants $\mathbb{C}(\tau, \lambda), \mathbb{C}^\star(\tau, \lambda)$ depend only on the parameters in Assumption 1 and is explicitly defined in Section E.*

**Remark.** *The $\log n$ term is to ensure mixing and is a common compromise when transitioning to Markov chains [Samson, 2000].*

# 5 Applications

We show how our results can be used to derive the asymptotic distributions of some popular statistics. The proofs of the results in this section, as well as more examples can be found in the appendix.

**Random Walk MH:** Random walk Metropolis-Hastings (MH) algorithms are an important algorithm to sample from a posterior density in Bayesian statistics Gelman et al. [1995]. To set the stage, we introduce the random walk Metropolis–Hastings (MH) algorithm with target density $\pi : \mathbb{R} \to \mathbb{R}_{\geq 0}$ and proposal $Q(x, dy) = q(x - y)\, dy$, where $q$ is a positive function on $\mathbb{R} \times \mathbb{R}$ satisfying $\int q(x - y)\, dy = 1$. For any $(x, y) \in \mathbb{R} \times \mathbb{R}$ define

$$\rho(x, y) = \begin{cases} \min\left(1, \dfrac{\pi(y)\, q(y - x)}{\pi(x)\, q(x - y)}\right) & \text{if } \pi(x)q(x - y) > 0, \\ 1 & \text{if } \pi(x)q(x - y) = 0. \end{cases}$$

The MH chain starts at $X_0 \sim \nu$ and moves from $X_n$ to $X_{n+1}$ according to the following rule:

1. Generate

$$Y \sim Q(X_n, dy) \qquad \text{and} \qquad W \sim \text{Bernoulli}\big(\rho(X_n, Y)\big).$$

2. With $\mathbf{1}$ as the indicator function, set

$$X_{n+1} = Y\mathbf{1}[W = 1] + X_n\mathbf{1}[W = 0].$$

We consider the following ball condition on the proposal $q_0$ associated with the random-walk Metropolis–Hastings algorithm which is popular in analysing the drift conditions.

**Assumption 2.** *Let $\pi \in \mathbb{S}_1(\mu)$ be a bounded probability density supported on a bounded convex $E \subset \mathbb{R}$, non-empty interior. Suppose there exist $b > 0$ and $\varepsilon > 0$ such that $\forall x \in \mathbb{R} \times \mathbb{R}$, $q_0(x) \geq b\, \mathbf{1}_{B(\varepsilon)}(x)$ where $B(\varepsilon)$ is the open ball with center $0$ and radius $\varepsilon$.*

We now have the following corollary,

**Corollary 2.** *Let $\mu$ be the unique median of $\pi$, and $X_1, \ldots, X_n$ be a sample from the MCMC rule described above. Then, $\hat{\mu}_n \xrightarrow{p} \mu$ and for all $m = o(n)$*

$$T_m^{(boot)}(\mu) \xrightarrow{d} \mathcal{N}(0, 1).$$

**Offline Ergodic MDP's:** Testing for the mean/median rewards of offline MDP's is an important problem in reinforcement learning [Sutton and Barto, 2018]. Let $\{(X_i, r_i)\}_{i=1}^n$ be an offline sample from an MDP on state space $[0, 1]^2$ under a given policy $\pi$. We will make the following assumption. The rewards are generated by a given reward function $r$. Formally, $r : [0, 1] \to [0, 1]$ and $r_i = r(X_i)$. We note that the state space can be any compact set other than $[0, 1]^2$, and such compactness restriction is common in practice. One can replace this assumption with geometric ergodicity under $\pi$. We will make the following assumption.

**Assumption 3.** *The transition density of $X_i$ under the policy $\pi$ is positive and admits two continuously differentiable derivatives. The reward function $r$ is one-one and onto and admits an inverse that is twice continuously differentiable.*

Under the previous assumptions, we have the following corollary:

**Corollary 3.** *Let $\mu$ be the median reward, and $\hat{\mu}_n$ be its estimator. Then the m-out-of-n bootstrap estimator of $\hat{\mu}_n$ satisfies.*

$$T_m^{(boot)}(\mu) \xrightarrow{d} \mathcal{N}(0, 1).$$

*It follows that with probability* 0.95

$$\hat{\mu}_n \in \hat{\mu}_n^{(boot)} \pm 1.96 \hat{\sigma}_m^{(boot)}/\sqrt{m}.$$

Corollary 3 is to the best of our knowledge, is the first parameter free confidence interval for the median rewards for offline MDP's.

# 6 Conclusion

This paper delivers a unified second-order theory for the *Studentized m-out-of-n bootstrap*. **(i)** We prove the first parameter-free central limit theorem for the resampled quantile estimator under only a finite-moment assumption, establishing rigorous guarantees for a tool already ubiquitous in practice. **(ii)** Leveraging a novel binomial representation, we obtain an exact $O(m^{-1/2})$ Edgeworth expansion together with a matching Berry–Esseen bound, thereby pinpointing when subsampling genuinely sharpens inference. **(iii)** We extend these results to regenerative Markov chains, providing the first bootstrap-based confidence guarantees for sequential data and illustrating the framework through applications such as Metropolis Hastings and offline MDPs. **(iv)** Throughout, we rectify long-standing errors in prior variance formulas and Edgeworth analyses, and we give principled guidance on choosing $m$.

**Limitations and outlook.** Our guarantees still assume mild smoothness and leave open the precise heavy-tail boundary, non-stationary processes, and adaptive variants remain compelling directions for future study. The question regarding the joint normality of the estimators also remain an important open question. Finally, it would also be nice to have an Edgeworth expansion of the Studentized bootstrap estimator for Markov chains. However, the theory of Edgeworth expansion on lattices for regenerating Markov chains seems to be sparse (in particular, we could not find a usable counterpart to Proposition 12 for Markov chains). Deriving such a result is out of scope for this (already lengthy) paper, and we plan to study it in a future work.

Another important assumption is that of regeneration. We note that the exponential moment condition in Assumption 1 is equivalent to the more classical geometric ergodicity of Markov chains (see Theorem 16.0.2 in Meyn and Tweedie [2012]). It is known that a weaker polynomial moment condition is equivalent to arithmatically mixing for Markov chains, and a corresponding result in this regime seems plausible, and warrants future investigation.

Finally, a natural extension of this work involves applying our framework to other estimators, particularly U- and M-estimators, which are commonly used in bootstrapping. Certain classes of M-estimation problems, such as shrinkage problems (see Chapter 1 in Hall which yields the median) or absolute-deviation loss functions, naturally lead to quantile estimation. In other cases, where the M-estimator depends on a function of quantiles, one may appeal to classical tools such as the delta method or CLT for M-estimators (see Theorem 5.21 in Van der Vaart [2000]) to establish asymptotic normality. That said, a comprehensive theory for these broader classes of estimators lies beyond the scope of the current work, but we view it as a compelling direction for future research.

# 7 Acknowledgment

The first author thanks Jorge Loria, Ksheera Sagar, and Ziwei Su for carefully going over an earlier edition of the draft and providing useful comments. The first author acknowledges the IEMS Alumni Fellowship at Northwestern University for financial support during which this research was conducted. The authors acknowledge the five anonymous reviewers for their useful comments and suggestions which significantly improved the readability of the paper.

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

# A  Proof of Theorem 1

*Proof.* We first show that

$$\left(\hat{\sigma}_m^{(boot)}\right)^2 \xrightarrow{a.s.} \frac{q(1-q)}{f^2(\mu)}.$$

Without losing generality, let $\mu$ be the unique median, let $\varepsilon > 0$ and let $q = 1/2$. The proof follows very similarly in other cases. Let $X_1, X_2, \ldots, X_n$ be i.i.d. random variables with their distributions satisfying the hypothesis of the theorem.

**Step 1.** For any $\varepsilon > 0$, we first observe that

$$\sum_{i \geq 0} \mathbb{P}\left(|X_i| > i^{1/\alpha}\varepsilon\right) = \sum_{i \geq 0} \mathbb{P}\left(|X_1| > i^{1/\alpha}\varepsilon\right) = \sum_{i \geq 0} \mathbb{P}\left(|X_1|^\alpha > i\varepsilon^\alpha\right) \leq \mathbb{E}[|X_1|^\alpha] < \infty.$$

Using Borel-Cantelli lemma, $X_i > i^{1/\alpha}\varepsilon$ only finitely many times with probability 1. Therefore,

$$\frac{|X_{(n)}| + |X_{(1)}|}{n^{1/\alpha}} \xrightarrow{a.s.} 0. \tag{A.1}$$

We next establish that $m(\hat{\mu}_m^{(boot)} - \hat{\mu}_n)^2$ is uniformly integrable. In order to do that, we show $\mathbb{E}_n\left[\sqrt{m}(\hat{\mu}_m^{(boot)} - \hat{\mu}_n)\right]^{2+\delta} < \infty$. Observe that

$$\mathbb{E}_n\left|\sqrt{m}(\hat{\mu}_m^{(boot)} - \hat{\mu}_n)\right|^{2+\delta} = (1+\delta) \int_0^\infty t^{1+\delta} \mathbb{P}_n\left(\sqrt{m}|\hat{\mu}_m^{(boot)} - \hat{\mu}_n| > t\right) dt.$$

To complete the proof it is now enough that the integral is finite. Consider first the case $\hat{\mu}_m^{(boot)} - \hat{\mu}_n > 0$, and observe that

$$\begin{aligned}
\{\sqrt{m}(\hat{\mu}_m^{(boot)} - \hat{\mu}_n) > t\} &= \left\{\hat{\mu}_m^{(boot)} > \hat{\mu}_n + \frac{t}{\sqrt{m}}\right\} \\
&= \left\{\frac{1}{2} - \frac{1}{2m} > F_n^{(m)}(\hat{\mu}_n + t/\sqrt{m})\right\} \\
&= \left\{\frac{1}{2} - \frac{1}{2m} - \bar{F}(\hat{\mu}_n + t/\sqrt{m}) > F_n^{(m)}(\hat{\mu}_n + t/\sqrt{m}) - \bar{F}(\hat{\mu}_n + t/\sqrt{m})\right\},
\end{aligned} \tag{A.2}$$

where we recall from that notations section that $\bar{F}(\cdot) = \sum_{i=1}^n \mathbf{1}[X_i \leq \cdot]/n$ is the empirical CDF. Let $c(\alpha) = 1/\alpha + 1/2$. We divide the proof into two cases.

**Step II** ($t \in [1, c(\alpha)\sqrt{\log m}]$)**:** We begin by analysing the left hand side of the event described in eq. (A.2).

$$\begin{aligned}
\frac{1}{2} - \frac{1}{2m} - \bar{F}(\hat{\mu}_n + t/\sqrt{m}) = \underbrace{\frac{1}{2} - \frac{1}{2m} - \bar{F}(\hat{\mu}_n)}_{=:A_{m,n}} + \underbrace{F(\hat{\mu}_n) - F(\hat{\mu}_n + t/\sqrt{m})}_{C_{m,n}} \\
+ \underbrace{\bar{F}(\hat{\mu}_n) - F(\hat{\mu}_n) + F(\hat{\mu}_n + t/\sqrt{m}) - \bar{F}(\hat{\mu}_n + t/\sqrt{m})}_{B_{m,n}}
\end{aligned}$$

Using [Bahadur, 1966, Lemma 1], we obtain $|B_{m,n}| \leq 1/\sqrt{n \log n}$ almost everywhere. Since $m \leq n$, it follows that

$$|B_{m,n}| \leq \frac{1}{\sqrt{m \log m}} \quad \text{almost everywhere.}$$

Turning to $C_{m,n}$ and using Taylor series expansion, we get that

$$
\begin{aligned}
C_{m,n} &= -\frac{t}{\sqrt{m}} f(\hat{\mu}_n) + o(t/m) \\
&= -\frac{t}{\sqrt{m}} f(\mu) + \frac{t}{\sqrt{m}} (\mu - \hat{\mu}_n) f'(\mu) + o\left(t/m\right). \\
&= -\frac{t}{\sqrt{m}} f(\mu) + \frac{t}{\sqrt{m}} (\mu - \hat{\mu}_n) f'(\mu) + \mathcal{O}\left(t/\sqrt{m}\right). \tag{A.3}
\end{aligned}
$$

Recall that under the hypothesis of the theorem, $f$ is continuously differentiable around the median. We first show that

$$
|\hat{\mu}_n - \mu| \le 1/\sqrt{n \log n} \quad \text{almost everywhere.}
$$

It follows from the main theorem of section 2.3.2 in Serfling [2009] that

$$
\mathbb{P}\left(|\hat{\mu}_n - \mu| > \varepsilon\right) \le 2e^{-2n\delta_\varepsilon^2}
$$

where $\delta_\varepsilon = \min\left\{F(\mu + \varepsilon) - 1/2, 1/2 - F(\mu - \varepsilon)\right\}$. Using Taylor series expansion on $F$, we get

$$
\delta_\varepsilon = \varepsilon f(\mu) + \mathcal{O}(\varepsilon^2).
$$

Now, set $\varepsilon = 1/\sqrt{n \log n}$ and take sum over all $n \ge 1$

$$
\sum_{n \ge 1} \mathbb{P}\left(|\hat{\mu}_n - \mu| > 1/\sqrt{n \log n}\right) \le \sum_{n \ge 1} \left(\frac{2}{n^2} + \mathcal{O}\left(\frac{1}{n^n}\right)\right) < \infty.
$$

Now using Borel-Cantelli lemma, it follows that $|\hat{\mu}_n - \mu| \le 1/\sqrt{n \log n}$ infinitely often almost everywhere. Substituting this in eq. (A.3), we get $|C_{m,n}| \le \mathcal{O}(t/\sqrt{m})$.

Turning to $A_{m,n}$, observe that $\mu$ is the median. Therefore, $F(\mu) = 1/2$. We then get

$$
\begin{aligned}
\frac{1}{2} - \frac{1}{2m} - \bar{F}(\hat{\mu}_n) &= -\frac{1}{2m} + F(\mu) - \bar{F}(\hat{\mu}_n) \\
&= -\frac{1}{2m} + \underbrace{(\hat{\mu}_n - \mu) f(\chi)}_{=: \text{ Term 1}} + \underbrace{F(\hat{\mu}_n) - \bar{F}(\hat{\mu}_n)}_{=: \text{ Term 2}}
\end{aligned}
$$

where $\chi \in \left[\mu - |\mu - \hat{\mu}_n|, \, \mu + |\mu - \hat{\mu}_n|\right]$. It follows that $f(\chi) < L$ almost everywhere for all values of $n$ large enough for some non-random constant $L$.

Furthermore, since

$$
|\hat{\mu}_n - \mu| \le 1/\sqrt{n \log n} \tag{A.4}
$$

almost everywhere, it follows that

$$
\text{Term 1} \le L/\sqrt{n \log n}
$$

almost everywhere for some bounded constant $L$ that depends only on $f$ and $\mu$. Combining,

$$
|A_{m,n}| \le \mathcal{O}(1/m)
$$

We move to Term 2.

Observe from Dvoretzky-Kiefer-Wolfowitz's theorem (Theorem A in Section 2.1.5 of Serfling [2009]) that

$$
\mathbb{P}\left(\sup_x |F(x) - \bar{F}(x)| > \varepsilon\right) \le \mathbb{C}e^{-2n\varepsilon^2}.
$$

We can now set $\varepsilon = 1/\sqrt{n \log n}$ similarly as before to get

$$
\mathbb{P}\left(\sup_x |F(x) - \bar{F}(x)| > \frac{1}{\sqrt{n \log n}}\right) \le \frac{2}{n^2}.
$$

Using Borel-Cantelli lemma, it now follows that $\sup_x |F(x) - \bar{F}(x)| < 1/\sqrt{n \log n}$ almost everywhere. This shows that $|B_{m,n}| \leq \mathcal{O}\left(1/\sqrt{n \log n}\right) \leq \mathcal{O}\left(1/\sqrt{m \log m}\right) \leq \mathcal{O}(1/\sqrt{m})$. Combining all three steps, we get that

$$\frac{1}{2} - \frac{1}{2m} - \bar{F}(\hat{\mu}_n + t/\sqrt{m}) = -\frac{t}{\sqrt{m}} + \mathcal{O}\left(\frac{t}{\sqrt{m}}\right). \tag{A.5}$$

For the sake of convenience, we denote $\hat{\mu}_n + t/\sqrt{m}$ by $t_m$. It now follows from eq. (A.2) that

$$\{\sqrt{m}(\hat{\mu}_m^{(boot)} - \hat{\mu}_n) > t\} \subseteq \left\{-\frac{t}{\sqrt{m}} + \frac{t\xi}{\sqrt{m}} \geq F_n^{(m)}(t_m) - \bar{F}(t_m)\right\} \tag{A.6}$$

for some finite real positive number $\xi$ which only depends on the sample $X_1, \ldots, X_n$, and hence is constant under $\mathbb{P}_n$. We now state the following lemma which is proved in the appendix

**Lemma 3.** *Under the conditions of Theorem 1*

$$\mathbb{P}_n\left(-\frac{t}{\sqrt{m}} + \frac{t\xi}{\sqrt{m}} \geq F_n^m(t_m) - \bar{F}(t_m)\right) \leq \frac{3}{t^4 (1-\xi)^4}$$

From equation eq. (A.2)

$$\mathbb{P}_n\left(\sqrt{m}(\hat{\mu}_m^{(boot)} - \hat{\mu}_n) > t\right) \leq \mathbb{P}_n\left(-\frac{t}{\sqrt{m}} + \frac{\xi}{\sqrt{m \log m}} \geq F_n^{(m)}(t_m) - \bar{F}(t_m)\right),$$

and the rest of the arguments follow.

**Step III** ($t > c(\alpha)\sqrt{\log m}$) For $t > c(\alpha) (\log m)^{1/2}$, and for all large $m$ almost surely, it follows by using equations A.5 and A.6 that,

$$\mathbb{P}_n\left(\sqrt{m} \left(\mu_m^{(boot)} - \hat{\mu}_n\right) > t\right) \tag{A.7}$$
$$\leq \mathbb{P}_n\left(\sqrt{m} \left(\mu_m^{(boot)} - \hat{\mu}_n\right) > c(\alpha) (\log m)^{1/2}\right)$$
$$\leq \mathbb{P}_n\left(F_n^{(m)}\left(\hat{\mu}_n + c(\alpha) (\log m)^{1/2} m^{-1/2}\right) - \bar{F}\left(\hat{\mu}_n + c(\alpha) (\log m)^{1/2} m^{-1/2}\right)\right.$$
$$\left.\leq -e\, c(\alpha) (\log m)^{1/2} m^{-1/2}\right).$$

Choose $c(\alpha) = \frac{1}{\alpha} + \frac{1}{2}$. The following lemma provides an upper bound of the term in the right hand side of the previous equation.

**Lemma 4.** *Under the conditions of Theorem 1,*

$$\mathbb{P}_n\left(F_n^{(m)}\left(\hat{\mu}_n + c(\alpha) (\log m)^{1/2} m^{-1/2}\right) - \bar{F}\left(\hat{\mu}_n + c(\alpha) (\log m)^{1/2} m^{-1/2}\right)\right.$$
$$\left.\leq -e\, c(\alpha) (\log m)^{1/2} m^{-1/2}\right).$$
$$\leq \mathcal{O}\left(m^{-\left(\frac{1}{\alpha} + \frac{1}{2}\right)(2+\delta)}\right)$$

Hence, for large $m$,

$$\int_{c(\alpha) (\log m)^{1/2}}^{m^{1/\alpha + 1/2}} t^{1+\delta} \mathbb{P}_n\left(\sqrt{m} \left(\mu_m^{(boot)} - \hat{\mu}_n\right) > t\right) \mathrm{d}t = \mathcal{O}(1) \quad \text{a.s.}$$

Using the previous fact and eq. (A.1) from Step 1, we have

$$\mathbb{P}_n\left(\sqrt{m} \left(\mu_m^{(boot)} - \hat{\mu}_n\right) > m^{1/2 + 1/\alpha}\right) = 0 \quad \text{a.s. for all large } n.$$

Thus, we have proved our result for $\sqrt{m} \left(\mu_m^{(boot)} - \mu\right)$. Similar arguments handle $-\sqrt{m} \left(\mu_m^{(boot)} - \mu\right)$. This completes the proof.

Next, we state the following Lemma which is proved below

**Lemma 5.** *Let $\mu$ be the unique $p$-th quantile of a distribution $F$ with continuous derivative $f$ around $\mu$. Then,*

$$\sqrt{m} \left(\hat{\mu}_m^{(boot)} - \hat{\mu}_n\right) \xrightarrow{d} \mathcal{N}\left(0, \frac{p(1-p)}{f^2(\mu)}\right).$$

Using this lemma, and the fact that

$$\left(\hat{\sigma}_m^{(boot)}\right)^2 \xrightarrow{a.s.} \frac{q(1-q)}{f^2(\mu)}$$

we have via Slutsky's theorem

$$\frac{\sqrt{m}\left(\hat{\mu}_m^{(boot)} - \hat{\mu}_n\right)}{\left(\hat{\sigma}_m^{(boot)}\right)} \xrightarrow{d} \mathcal{N}(0, 1).$$

This proves our Theorem.

$\square$

We now prove Lemmas 3, 4, and 5.

## A.1 Proof of Lemma 3

*Proof.* Recall that $\xi$ is a constant given the sample. Using the conditional version of Markov's inequality we get that

$$\mathbb{P}_n\left(-\frac{t}{\sqrt{m}} + \frac{t\xi}{\sqrt{m}} \geq F_n^{(m)}(t_m) - \bar{F}(t_m)\right) \leq \frac{\mathbb{E}_n\left[F_n^{(m)}(t_m) - \bar{F}(t_m)\right]^4}{(t/\sqrt{m} - t\xi/\sqrt{m})^4}$$

$$= \frac{\mathbb{E}_n\left[\sqrt{m}\left(F_n^{(m)}(t_m) - \bar{F}(t_m)\right)\right]^4}{(t - t\xi)^4}. \qquad \text{(A.8)}$$

Observe that the bootstrapped sample $X_1^*, \ldots, X_m^*$ is an i.i.d. sample from the empirical distribution $\bar{F}$. Therefore, conditioned on the full sample $mF_n^{(m)}(x) \sim Binomial(m, \bar{F}(x))$. Recall that if $X \sim Binomial(n, p)$, $\mathbb{E}[X - np]^4 \leq 3n^2$. Using this fact in eq. (A.8), we get

$$\mathbb{P}_n\left(\sqrt{m}(\hat{\mu}_m^{(boot)} - \hat{\mu}_n) > t\right) \leq \frac{3}{t^4(1-\xi)^4}.$$

This completes the proof.

$\square$

## A.2 Proof of Lemma 4

*Proof.* Recall that we were required to bound

$$\mathbb{P}_n\Big(F_n^{(m)}\big(\hat{\mu}_n + c(\alpha)(\log m)^{1/2} m^{-1/2}\big) - \bar{F}\big(\hat{\mu}_n + c(\alpha)(\log m)^{1/2} m^{-1/2}\big)$$

$$\leq -e\, c(\alpha)(\log m)^{1/2} m^{-1/2}\Big).$$

Using the Hoeffding type bound in Lemma 3.1 of Singh [1981] with

$$p = F_n\big(\hat{\mu}_n + c(\alpha)(\log m)^{1/2} m^{-1/2}\big), \quad B = 1, \quad Z = \left(\frac{1}{\alpha} + \frac{1}{2}\right)(2 + \delta)\log m,$$

and

$$D = e\, c(\alpha)(1 + e/2)^{-1}(\log m)^{1/2} m^{1/2}$$

that

$$\mathbb{P}_n\Big(F_n^{(m)}\big(\hat{\mu}_n + c(\alpha)(\log m)^{1/2} m^{-1/2}\big) - \bar{F}\big(\hat{\mu}_n + c(\alpha)(\log m)^{1/2} m^{-1/2}\big)$$

$$\leq -e\, c(\alpha)(\log m)^{1/2} m^{-1/2}\Big).$$

$$\leq \mathcal{O}\big(m^{-\left(\frac{1}{\alpha} + \frac{1}{2}\right)(2+\delta)}\big).$$

The rest of the proof follows.

$\square$

## A.3 Proof of Lemma 5

*Proof.* We shorthand $t_m = \hat{\mu}_n + t/\sqrt{m}$. We begin following the steps of eq. (A.2) to get

$$\{\sqrt{m}(\hat{\mu}_m^{(boot)} - \hat{\mu}_n) < t\} = \left\{ F_n^{(m)}(t_m) \geq p - \frac{p}{m} \right\}$$

$$= \left\{ \frac{\sqrt{m}(F_n^{(m)}(t_m) - \bar{F}(t_m))}{[\bar{F}(t_m)(1 - \bar{F}(t_m))]^{1/2}} \geq \frac{\sqrt{m}\left(p(1 - \frac{1}{m}) - \bar{F}(t_m)\right)}{[\bar{F}(t_m)(1 - \bar{F}(t_m))]^{1/2}} \right\}.$$

By continuity correction,

$$\mathbb{P}_n\left( \frac{\sqrt{m}(F_n^{(m)}(t_m) - \bar{F}(t_m))}{[\bar{F}(t_m)(1 - \bar{F}(t_m))]^{1/2}} \geq \frac{\sqrt{m}\left(p\frac{m-1}{m} - \bar{F}(t_m)\right)}{[\bar{F}(t_m)(1 - \bar{F}(t_m))]^{1/2}} \right)$$

$$= \mathbb{P}_n\left( \frac{\sqrt{m}(F_n^{(m)}(t_m) - \bar{F}(t_m))}{[\bar{F}(t_m)(1 - \bar{F}(t_m))]^{1/2}} \geq \underbrace{\frac{\sqrt{m}\left(p - \bar{F}(t_m)\right)}{[\bar{F}(t_m)(1 - \bar{F}(t_m))]^{1/2}}}_{=:t_m^*} \right)$$

Observe that given the sample $F_n^{(m)}(x) \sim \text{Binomial}(m, \bar{F}(x))$ and let $n$ be large enough. Using a central limit theorem for Binomial random variables, observe that

$$\frac{\sqrt{m}(F_n^{(m)}(t_m) - \bar{F}(t_m))}{[\bar{F}(t_m)(1 - \bar{F}(t_m))]^{1/2}} \overset{d}{=} \mathcal{N}(0, 1) + \mathcal{R}_m$$

where $\mathcal{R}_m$ is a remainder term that decays in probability to 0 as $m \to \infty$.

Let $m$ be large enough such that $\mathcal{R}_m < \varepsilon$. It follows that,

$$\mathbb{P}_n\left( \frac{\sqrt{m}(F_n^{(m)}(t_m) - \bar{F}(t_m))}{[\bar{F}(t_m)(1 - \bar{F}(t_m))]^{1/2}} \geq \underbrace{\frac{\sqrt{m}\left(p - \bar{F}(t_m)\right)}{[\bar{F}(t_m)(1 - \bar{F}(t_m))]^{1/2}}}_{=:t_m^*} \right) = \mathbb{P}_n\left( \mathcal{N}(0, 1) > t_m^* - \varepsilon \right)$$

$$= 1 - \Phi(t_m^* - \varepsilon).$$

We will now show that

$$t_m^* \xrightarrow{m,n} t\sqrt{\frac{f(\mu)^2}{p(1 - p)}}.$$

Using a method similar to the derivation in eq. (A.5), we have

$$\bar{F}(t_m) = p\frac{m - 1}{m} + \frac{t}{\sqrt{m}} + \mathcal{O}\left( \frac{t}{\sqrt{m}} \right)$$

$$\xrightarrow{m} p.$$

Therefore,

$$[\bar{F}(t_m)(1 - \bar{F}(t_m))]^{1/2} \xrightarrow{m,n} \sqrt{p(1 - p)}.$$

Now we address the numerator. By strong law of large numbers, $\bar{F}(t_m) = F(t_m) + \mathcal{O}_p(1/n)$.

Using a first-order Taylor series expansion on $F(t_m)$, we have

$$F(t_m) = F(\mu) + \left(\hat{\mu}_n + t/\sqrt{m} - \mu\right) f(\chi)$$

$$= p + \left(\hat{\mu}_n + t/\sqrt{m} - \mu\right) f(\chi),$$

where

$$\chi \in \left[\mu - |\mu - \hat{\mu}_n|, \ \mu + |\mu - \hat{\mu}_n|\right].$$

Recall from eq. (A.4) that $|\mu - \hat{\mu}_n| = o_p(1)$. Since $f$ is continuous around $\mu$, it follows using continuous mapping theorem that

$$F(t_m) \xrightarrow{n} p + tf(\mu)/\sqrt{m}$$

and

$$\sqrt{m}\left(p - \bar{F}(t_m)\right) \xrightarrow{n} -tf(\mu).$$

Therefore,

$$\Phi(t_m^* - \varepsilon) = \Phi(-tf(\mu)/\sqrt{p(1-p)} - \varepsilon)$$

and

$$1 - \Phi(t_m^* - \varepsilon) = \Phi(tf(\mu)/\sqrt{p(1-p)} + \varepsilon).$$

$\varepsilon$ is arbitrary. We now use the fact that $\Phi(tf(\mu)/\sqrt{p(1-p)})$ is the CDF of a gaussian distribution with mean 0 and variance $p(1-p)/f^2(\mu)$. This completes the proof. $\square$

# B  On the Variance of the m-out-of-n Bootstrap Estimator for sample quantiles

This section is dedicated to developing the finite sample theory of the variance of the m-out-of-n bootstrap estimators of the sample quantiles. We first write the following lemma which provides a closed form expression for the variance.

**Lemma 6.** *The m-out-of-n bootstrap estimator for the $p$-th sample quantile has the closed form solution*

$$\sum_{j=1}^{n} \left(X_{(j)} - X_{(r)}\right)^2 W_{m,j} \tag{B.1}$$

*where*

$$W_{m,j} = k\binom{m}{k} \int_{(j-1)/n}^{j/n} x^{k-1}(1-x)^{m-k} dx, \quad k = \lfloor mp \rfloor + 1, \ \text{and } r = \lfloor np \rfloor + 1.$$

*Therefore,*

$$\left(\hat{\sigma}_m^{(boot)}\right)^2 = \mathrm{Var}_n\left(\sqrt{m}\left(\hat{\mu}_m^{(boot)} - \hat{\mu}_n\right)\right) = m\sum_{j=1}^{n}\left(X_{(j)} - X_{(r)}\right)^2 W_{m,j}. \tag{B.2}$$

## B.1  Proof of Lemma 6

*Proof.* Without losing generality, let $p = 1/2$ so that $\mu$ is the sample median.

**Case $m = 2q + 1$:** We first consider the case when $m = 2q + 1$ for some integer $q$. The median is therefore $X_{(q+1)}^*$. As before, let $X_1^*, \ldots, X_n^* \overset{i.i.d}{\sim} \bar{F}$. The distribution of the median has the following closed form solution

$$\mathbb{P}_n\left(X_{(q+1)}^* < u\right) = \sum_{j=q+1}^{n} \binom{m}{j} \left[\bar{F}(x)\right]^j \left[1 - \bar{F}(x)\right]^{n-j}.$$

The pdf can be obtained by differentiation. This gives us the following closed form expression for the moments.

$$E\left((\hat{\mu}_n^{(boot)})^r\right) = \frac{(2q+1)!}{(q!)^2} \int_{-\infty}^{\infty} x^r \left[\bar{F}(x)\left(1 - \bar{F}(x)\right)\right]^m f(x)\, dx. \tag{B.3}$$

If we let $y = \bar{F}(x)$ and write $x = \bar{F}^{-1}(y) = \psi(y)$, then eq. (B.3) becomes

$$E\left((\hat{\mu}_n^{(boot)})^r\right) = \frac{(2q+1)!}{(q!)^2} \int_0^1 [\psi(y)]^r [y(1-y)]^m\, dy.$$

Estimating the function $\psi(y)$ by the observed order statistics is therefore equivalent to estimating $\bar{F}(x)$ by $F_n^{(m)}$. Thus $E\left((\hat{\mu}_n^{(boot)})^r\right)$ is estimated by

$$\mathcal{A}_{rm} = \sum_{j=1}^{n} X_{(j)}^r \mathcal{B}_j \tag{B.4}$$

where

$$\mathcal{B}_j = \frac{(2q+1)!}{(q!)^2} \int_{(j-1)/n}^{j/n} y^m (1-y)^m \, dy.$$

The quantity $\mathrm{Var}((\hat{\mu}_n^{(boot)})^r)$ is therefore estimated by

$$\mathrm{Var}((\hat{\mu}_n^{(boot)})^r) = \mathcal{A}_{2m} - \mathcal{A}_{1m}^2.$$

Substituting the values of $\mathcal{A}_{2n}$ and $\mathcal{A}_{1n}$ from eq. (B.4), we have the given result. The case $n = 2q$ is handled similarly. $\qquad\square$

The following lemma establishes the rate of convergence of the variance of the m-out-of-n bootstrap estimator of the sample quantiles.

**Lemma 7.** *Let $\mu$ be the unique p-th quantile such that $F \in \mathbb{S}_2(\mu)$, and $f'(\mu) > 0$. Let $m = o(n^\lambda)$ for some $\lambda > 0$ and $\mathbb{E}|X_1|^\alpha < \infty$. Then,*

$$\left(\hat{\sigma}_m^{(boot)}\right)^2 = \frac{\sigma^2}{m} + \mathcal{O}_p(m^{-3/4} n^{-1/2}).$$

## B.2 Proof of Lemma 7

*Proof.* Let $\varphi$ denote the standard normal density function,

$$Y_{n,j} = (j-1)/n \text{ and } b_{mn} = \frac{(mY_{n,j} - k)}{\sqrt{mY_{n,j}(1 - Y_{n,j})}}$$

where $k = \lfloor mp \rfloor + 1$. The following lemma states a useful expansion for the weight $W_{m,j}$ and is proved below.

**Lemma 8.** *Assume that $m \propto n^\lambda$ for some $\lambda \in (0, 1)$. There exists some constant $C > 0$ such that*

$$W_{m,j} = \frac{\sqrt{m}\phi(b_{mn})}{n\sqrt{Y_{n,j}(1 - Y_{n,j})}} + \mathcal{O}(n^{-1} e^{-Cm(Y_{n,j} - p)^2}).$$

Put $H(x) = F^{-1}(e^{-x})$. Using Rényi's representation, let $Z_1, \dots, Z_n$ be independent unit–mean negative–exponential random variables such that for all $1 \le j \le n$,

$$X_{(j)} = H\left(\sum_{k=j}^{n} k^{-1} Z_k\right).$$

For any integer $r$, following a similar procedure in Hall and Martin [1988b], we define

$$s_j \equiv \mathrm{sgn}(r - j), \quad m_{0j} \equiv \min(r, j), \quad m_{1j} \equiv \max(r, j) - 1, \quad a \equiv H'\left(\sum_{k=r}^{n} k^{-1}\right),$$

and set

$$A_j \equiv \sum_{k=j}^{n} k^{-1} Z_k, \qquad B_j \equiv \sum_{k=m_{0j}}^{m_{1j}} k^{-1} Z_k, \qquad b_j \equiv E(B_j) = \sum_{k=m_{0j}}^{m_{1j}} k^{-1},$$

the latter two quantities being zero if $j = r$.

We first consider the summation over $j$ in eq. (B.1). The summation is divided into two parts.

**Case I:** Let, $|r - j| > \delta n^{1+\beta} m^{-1/2}$ for some $\delta > 0$ and some $\beta < \lambda/12$.
Then $X_{(j)} - X_{(r)} = D_j + R_{1j}$, where

$$D_j \equiv s_j a B_j, \quad R_{1j} \equiv R_{2j} + R_{3j}, \quad R_{2j} \equiv s_j B_j \{H'(A_r) - a\},$$

and

$$R_{3j} \equiv s_j B_j \int_0^1 \{H'(A_r + ts_j B_j) - H'(A_r)\}\, dt.$$

Thus

$$(X_{(j)} - X_{(r)})^2 = a^2 b_j^2 + 2a^2 b_j (B_j - b_j) + a^2 (B_j - b_j)^2 + 2D_j R_{1j} + R_{1j}^2. \tag{B.5}$$

Assuming $E|X|^\eta < \infty$,

$$P\big(|X_j| \le n^{2/\eta} \text{ for all } j \le n\big) = \big\{1 - P(|X| > n^{2/\eta})\big\}^n$$
$$\ge \big(1 - n^{-2} E|X|^\eta\big)^n = 1 + O(n^{-1}).$$

Therefore, with probability tending to one as $n \to \infty$,

$$\max_{j \le n} (X_{(j)} - X_{(r)})^2 \le 4\, n^{4/\eta}.$$

This, combined with Lemma eq. (B.1) implies that, for some constant $C_2 > 0$, $W_{m,j} < C_2 m^{1/2} n^{-1} e^{-Cm(Y_{n,j} - p)^2}$. Thus, with probability tending to one, we have that for some constant $C_3 > 0$ and any $\eta > 0$,

$$\sum_{|j-r| > \delta n^{1+\beta} m^{-1/2}} (X_{(j)} - X_{(r)})^2 W_{m,j} \le 4C_2 m^{1/2} n^{4/\eta} e^{-C_3 n^{2\beta}} = O(n^{-\zeta}).$$

**Case II:** Recall that under the hypothesis $F \in \mathbb{S}_2(\mu)$ $f$ is Lipschitz in a neighbourhood of $\mu$, so that $a = H'(A_r) = -pf(\mu)^{-1} + O(n^{-1})$. Observe that

$$\sum_j (X_{(j)} - X_{(r)})^2 W_{m,j} = S_1 + S_2 + T_1 + T_2 + T_3,$$

where

$$
\begin{aligned}
&S_1 = a^2 \sum_j b_j^2 W_{m,j} && S_2 = 2a^2 \sum_j b_j (B_j - b_j) W_{m,j} \\
&T_1 = a^2 \sum_j (B_j - b_j)^2 W_{m,j} && T_2 = 2 \sum_j D_j R_{1j} W_{m,j}, \\
&T_3 = \sum_j R_{1j}^2 W_{m,j}, && B_j = \sum_{u=m_{0j}}^{m_{1j}} u^{-1} Z_u \\
&b_j = \mathbb{E}(B_j) && D_j = s_j a B_j \\
&R_{1j} = R_{2j} + R_{3j} && R_{2j} = s_j B_j [H'(A_r) - a]
\end{aligned}
$$

and $R_{3j} = s_j B_j \int_0^1 [H'(A_r + ts_j B_j) - H'(A_r)] dt$.
Note also that $B_r = b_r = 0$, and

$$b_j = |j - r| r^{-1} + \frac{1}{2} r^{-2} (j - r)^2 + \mathcal{O}_p(|j - r|^3 r^{-3}).$$

Using eq. (B.1) and the above expansion of $a$,

$$\sum_j b_j^2 W_{m,j} = m^{-1} p^{-1}(1 - p) + \mathcal{O}_p(m^{-3/2}),$$

so that

$$S_1 = m^{-1} p(1 - p) f(\mu)^{-2} + \mathcal{O}_p(m^{-3/2}).$$

For $S_2$, we observe that

$$S_2 = 2a^2 \left\{ \sum_{u=r-\delta n^{1+\beta}m^{-1/2}}^{r-1} u^{-1}(Z_u - 1) \sum_{j=r-\delta n^{1+\beta}m^{-1/2}}^{u} b_j w_{m,j} \right.$$

$$\left. + \sum_{u=r}^{r+\delta n^{1+\beta}m^{-1/2}-1} u^{-1}(Z_u - 1) \sum_{j=u+1}^{r+\delta n^{1+\beta}m^{-1/2}} b_j w_{m,j} \right\}$$

Then, by Lyapunov's central limit theorem,

$$m^{3/4}n^{1/2}S_2 \xrightarrow{d} \mathcal{N}(0, 2\pi^{-1/2}[p(1-p)]^{3/2}f(\mu)^{-4}).$$

In other words,

$$S_2 = \mathcal{O}_p(m^{-3/4}n^{-1/2}).$$

We note that using eq. (B.1) and for $t > 0$,

$$\sum_j |j-np|^t w_{m,j} \sim m^{1/2}n^t \int_{p-\delta n^{\beta}m^{-1/2}}^{p+\delta n^{\beta}m^{-1/2}} |y-p|^t [y(1-y)]^{-1/2} \phi\left(\frac{m^{1/2}(y-p)}{\sqrt{y(1-y)}}\right) dy = \mathcal{O}_p(m^{-t/2}n^t).$$

It follows by substituting appropriate values for $t$ that

$$\mathbb{E}(T_1) = O\left(\sum_j |j-r|r^{-2}w_{m,j}\right) = O(m^{-1/2}n^{-1}),$$

$$\mathbb{E}|T_2| = O\left(\sum_j \left[n^{-2}(j-r)^2 n^{-1/4-1/2} + (n^{-1}|j-r|)^{5/2+1}\right] W_{m,j}\right) = \mathcal{O}(m^{-9/4}n^{-1}),$$

and

$$\mathbb{E}(T_3) = O\left(\sum_j \left[n^{-2}(j-r)^2 n^{-3/2} + (n^{-1}|j-r|)^5\right] W_{m,j}\right) = \mathcal{O}(m^{-5/2}n^{-1}),$$

so that, by Chebyshev's inequality, the terms $T_1, T_2, T_3$ can be shown to satisfy:

$$T_1 = \mathcal{O}_p(m^{-1/2}n^{-1}), \quad T_2 = \mathcal{O}_p(m^{-9/4}n^{-1}), \quad T_3 = \mathcal{O}_p(m^{-5/2}n^{-1}).$$

Combining, we have,

$$\sum_j (X_{(j)} - X_{(r)})^2 W_{m,j} = S_1 + S_2 + T_1 + T_2 + T_3$$

$$= \frac{1}{m}\frac{p(1-p)}{f(\mu)^2} + \mathcal{O}_p(m^{-3/4}n^{-1/2}) + \mathcal{O}_p(m^{-1/2}n^{-1})$$

$$+ \mathcal{O}_p(m^{-9/4}n^{-1}) + \mathcal{O}_p(m^{-5/2}n^{-1})$$

$$= \frac{\sigma^2}{m} + \mathcal{O}_p(m^{-3/4}n^{-1/2}).$$

This completes the proof.

$\square$

### B.3 Proof of Lemma 8

*Proof.* Note that $W_{m,j} = I_{j/n}(k, m-k+1) - I_{(j-1)/n}(k, m-k+1)$, where

$$I_y(a,b) = \sum_{j=a}^{j=a+b-1} \binom{a+b-1}{j} y^j (1-y)^{a+b-1-j}$$

is the incomplete Beta function. Without loss of generality, consider $j = np + q$ with $q \geq 0$. When $0 \leq q \leq Dnm^{-1/2}(\log m)^{1/2}$, for some $D > 0$, we use Edgeworth expansion for the binomial distribution, and when $q > Dnm^{-1/2}(\log m)^{1/2}$, we use Bernstein's inequality. This gives us

$$I_{j/n}(k, m-k+1) - I_{(j-1)/n}(k, m-k+1) = \underbrace{\frac{\sqrt{m}\phi(b_{mn})}{n\sqrt{Y_{n,j}(1-Y_{n,j})}}}_{\text{from Edgeworth expansion}} + \underbrace{\mathcal{O}(n^{-1}e^{-Cm(Y_{n,j}-p)^2})}_{\text{from Bernstein's inequality}}$$

where $Y_{n,j} = (j-1)/n$. This completes the proof. $\qquad\square$

## C  Proof of Theorem 2

*Proof.* The main ingredient of this proof is the Edgeworth expansion of the Binomial distribution in Lemma 9. The lemma is proved below.

**Lemma 9.** *Let $X$ be a binomial random variable with parameters $m$ and $p$. Then,*

$$\mathbb{P}\left(\frac{\sqrt{m}(X-p)}{\sqrt{p(1-p)}} < t\right) = \Phi(t) - \frac{(1-2p)}{6\sqrt{np(1-p)}}\phi(t)\mathbb{H}_2(t) + \mathcal{O}\left(\frac{1}{n}\right) \tag{C.1}$$

*where $\mathcal{O}(n^{-1})$ can be bounded independently of $t$.*

We shorthand $t_m = \hat{\mu}_n + t\hat{\sigma}_m^{(boot)}$. To apply Lemma 9, we represent the event $\{(\hat{\mu}_m^{(boot)} - \hat{\mu}_n)/\hat{\sigma}_m^{(boot)} < t\}$ as a Binomial probability.

**Continuity correction at $t_m$:**  We begin following the steps of eq. (A.2) to get

$$\left\{\frac{\sqrt{m}(\hat{\mu}_m^{(boot)} - \hat{\mu}_n)}{\hat{\sigma}_m^{(boot)}} \leq t\right\} = \left\{F_n^{(m)}(t_m) \geq \frac{1}{2} - \frac{1}{2m}\right\}$$

$$= \left\{\frac{\sqrt{m}(F_n^{(m)} - \bar{F}(t_m))}{[\bar{F}(t_m)(1-\bar{F}(t_m))]^{1/2}} \geq \frac{\sqrt{m}\left(\frac{1}{2} - \frac{1}{2m} - \bar{F}(t_m)\right)}{[\bar{F}(t_m)(1-\bar{F}(t_m))]^{1/2}}\right\}.$$

Observe that given the sample $F_n^{(m)}(x) \sim \text{Binomial}(m, \bar{F}(x))$. Thus, our next step is the following continuity correction.

$$\mathbb{P}_n\left(\frac{\sqrt{m}(F_n^{(m)} - \bar{F}(t_m))}{[\bar{F}(t_m)(1-\bar{F}(t_m))]^{1/2}} \geq \frac{\sqrt{m}\left(\frac{m-1}{2m} - \bar{F}(t_m)\right)}{[\bar{F}(t_m)(1-\bar{F}(t_m))]^{1/2}}\right)$$

$$= \mathbb{P}_n\left(\frac{\sqrt{m}(F_n^{(m)} - \bar{F}(t_m))}{[\bar{F}(t_m)(1-\bar{F}(t_m))]^{1/2}} \geq \underbrace{\frac{\sqrt{m}\left(\frac{1}{2} - \bar{F}(t_m)\right)}{[\bar{F}(t_m)(1-\bar{F}(t_m))]^{1/2}}}_{=:t_m^*}\right) \tag{C.2}$$

Using a complement, and Lemma 9, we now have

$$\mathbb{P}_n\left(\frac{\sqrt{m}(F_n^{(m)} - \bar{F}(t_m))}{[\bar{F}(t_m)(1-\bar{F}(t_m))]^{1/2}} \geq t_m^*\right)$$

$$= 1 - \mathbb{P}_n\left(\frac{\sqrt{m}(F_n^{(m)} - \bar{F}(t_m))}{[\bar{F}(t_m)(1-\bar{F}(t_m))]^{1/2}} \leq t_m^*\right)$$

$$= \underbrace{1 - \Phi(t_m^*)}_{A} + \underbrace{\frac{1 - 2\bar{F}(t_m)}{6\sqrt{m}(\bar{F}(t_m)(1-\bar{F}(t_m))^{1/2}}\mathbb{H}_2(t_m^*)\phi(t_m^*)}_{B} + \mathcal{O}_p\left(\frac{1}{m}\right) \tag{C.3}$$

where $\mathcal{O}_p$ is with respect to the full sample distribution $X_1, \ldots, X_n$.

**Expansion of** $(1 - 2\bar{F}(t_m))/(\bar{F}(t_m)(1 - \bar{F}(t_m)))^{1/2}$: To find the proper Edgeworth expansion, we need to find exact expressions for $A$ and $B$. We use the following lemma which is proved below.

**Lemma 10.** *Under the hypothesis of Theorem 2,*

$$\bar{F}(t_m) = \frac{1}{2} + \frac{t\sigma}{\sqrt{m}}f(\mu) + \frac{(t\sigma)^2}{2m}f'(\mu) + \mathcal{O}_p\left(m^{-1/4}n^{-1/2}\right).$$

For convenience of notation, introduce the notation $T_m$ as follows.

$$T_m := \frac{t\sigma}{\sqrt{m}}f(\mu) + \frac{(t\sigma)^2}{2m}f'(\mu) + \mathcal{O}_p\left(m^{-1/4}n^{-1/2}\right)$$

Let $g(x) = -2x/\sqrt{(1/4 - x^2/4)}$. Then observe that $g(x)$ admits the following Taylor series expansion.

$$g(x) = -4x + o(x^3).$$

Using this fact and Lemma 10, one can expand $(1 - 2\bar{F}(t_m))/(\bar{F}(t_m)(1 - \bar{F}(t_m)))^{1/2} = g(T_m)$ as

$$\frac{(1 - 2\bar{F}(t_m))}{(\bar{F}(t_m)(1 - \bar{F}(t_m)))^{1/2}} = -4T_m + o(T_m^3)$$

$$= -\frac{4t\sigma}{\sqrt{m}}f(\mu) - \frac{2(t\sigma)^2}{m}f'(\mu) + \mathcal{O}_p\left(m^{-1/4}n^{-1/2}\right) + \mathcal{O}_p(1/m^{3/2}).$$

$$\text{(C.4)}$$

Observe that $o(T_m^3) = o(1/m^{3/2})$. Therefore, the leading terms in the expansion arise from $T_m$.

**Expansion of** $t_m^*$ **and** $\mathbb{H}_2(t_m^*)\phi(t_m^*)$: Observe that $t_m^* = \sqrt{m}g(T_m)/2$. Therefore, we immediately recover the following expansion for $t_m^*$.

$$t_m^* = -2t\sigma f(\mu) - \frac{(t\sigma)^2}{\sqrt{m}}f'(\mu) + \mathcal{O}_p(m^{1/4}n^{-1/2}).$$

Let $\phi^{(2)}$ denote the second derivative of $\phi$, and observe that $\mathbb{H}_2(x) = \phi^{(2)}(x)/\phi(x)$. Equivalently, $\mathbb{H}_2(x)\phi(x) = \phi^{(2)}(x)$, and $\frac{d}{dx}\mathbb{H}_2(x)\phi(x) = \phi^{(3)}(x)$. Using this fact, and the fact that $\mathbb{H}_2$ and $\phi$ are both even functions, the Taylor series expansion of $\mathbb{H}_2(t_m^*)\phi(t_m^*)$ around $-2t\hat{\sigma}_m^{(boot)}f(\mu)$ is given by

$$\mathbb{H}_2(-2t\sigma f(\mu))\phi(-2t\sigma f(\mu)) - \frac{(t\sigma)^2}{\sqrt{m}}f'(\mu)\phi^{(3)}(-2t\sigma f(\mu)) + \mathcal{O}_p(m^{1/4}n^{-1/2}) + \mathcal{O}_p\left(\frac{1}{m}\right) \quad \text{(C.5)}$$

$$= \mathbb{H}_2(2t\sigma f(\mu))\phi(2t\sigma f(\mu)) - \frac{(t\sigma)^2}{\sqrt{m}}f'(\mu)\phi^{(3)}(-2t\sigma f(\mu)) + \mathcal{O}_p(m^{1/4}n^{-1/2}) + \mathcal{O}_p\left(\frac{1}{m}\right)$$

**Substitution and Rearrangement:** We now return to eq. (C.3) to substitute the $A$ and $B$.

$$1 - \Phi(t_m^*) = 1 - \Phi\left(-2t\sigma f(\mu) - \frac{(t\sigma)^2}{\sqrt{m}}f'(\mu) + \mathcal{O}_p(m^{1/4}n^{-1/2})\right)$$

$$= 1 - \Phi(-2t\sigma f(\mu)) + \frac{(t\sigma)^2}{\sqrt{m}}f'(\mu)\phi(-2t\sigma f(\mu))\mathcal{O}_p(m^{1/4}n^{-1/2})$$

$$= \Phi(2t\sigma f(\mu)) + \frac{(t\sigma)^2}{\sqrt{m}}f'(\mu)\phi(2t\sigma f(\mu)) + \mathcal{O}_p(m^{1/4}n^{-1/2})$$

Next, we write the full expansion of $B = g(T_m)\mathbb{H}_2(t_m^*)\phi(t_m^*)/(6\sqrt{m})$ and bound it using the following lemma which is proved below.

**Lemma 11.** *Under the conditions of Theorem 2,*

$$\frac{g(T_m)\mathbb{H}_2(t_m^*)\phi(t_m^*)}{6\sqrt{m}} = \mathcal{O}_p\left(\frac{1}{m}\right) + \mathcal{O}_p(m^{-1/4}n^{-1/2})$$

We can now collect all the terms in eq. (C.3) and use Lemma 11 to get

$$\mathbb{P}_n\left(\frac{(\hat{\mu}_m^{(boot)} - \hat{\mu}_n)}{\hat{\sigma}_m^{(boot)}} \leq t\right) = \Phi\left(2t\sigma f(\mu)\right) + \frac{(t\sigma)^2}{\sqrt{m}}f'(\mu)\phi(2t\sigma f(\mu))$$

$$+ \mathcal{O}_p\left(\frac{1}{m}\right) + \mathcal{O}_p(m^{-1/4}n^{-1/2}).$$

Finally, observe that $\sigma = 1/(2f(\mu))$. Therefore, $2\sigma f(\mu) = 1$ and we have,

$$\mathbb{P}_n\left(\frac{(\hat{\mu}_m^{(boot)} - \hat{\mu}_n)}{\hat{\sigma}_m^{(boot)}} \leq t\right) = \Phi\left(t\right) + \frac{(t\sigma)^2}{\sqrt{m}}f'(\mu)\phi(2t\sigma f(\mu))$$

$$+ \mathcal{O}_p\left(\frac{1}{m}\right) + \mathcal{O}_p(m^{-1/4}n^{-1/2}).$$

This completes the proof. □

Now we prove lemmas 9, 10, and 11.

**Proof of Lemma 9**

*Proof.* To prove Lemma 9, we briefly introduce polygonal approximants of CDF's.

**Feller's Edgeworth expansion on Lattice:** To make matters concrete, we first adapt from Feller [1991] the notion of polygonal approximants on lattice. If $F$ is a CDF defined on the set of lattice points $b \pm ih$, with $i \in \mathbb{Z}^+$ and $h$ being the span of $F$, then the *polygonal approximant* $F^\#$ of $F$ is given by

$$F^\# = \begin{cases} F(x) & \text{if } x = b \pm \left(i + \frac{1}{2}\right)h \\ \frac{1}{2}[F(x) + F(x-))] & \text{if } x = b \pm ih \end{cases}$$

Following the terminology of Page 540 in Feller [1991], observe that the empirical m-out-of-n bootstrap CDF $F_n^{(m)}$ is an m-fold-convolution of $\bar{F}$ (i.e. it is the CDF of $m$ i.i.d. samples from $\bar{F}$). The notation in the following result is self-contained. It is due to Feller [1991] (Theorem 2, Chapter XVI, Section 4).

**Proposition 12** (Theorem 2 in Feller, Page 540). *Let $X_1, \ldots, X_n$ be $n$ i.i.d. random variables on a lattice with finite third moment $\mu_3$ and variance $\sigma^2$. Let $F^\#$ be the polygonal approximant of the empirical CDF on a lattice with span $h/(\sigma\sqrt{n})$, and $\mathbb{H}_2(\cdot) = (x^2 - 1)$ be the second order Hermite polynomial. Then,*

$$F^\#(t) = \Phi(t) - \frac{\mu_3}{6\sigma^3\sqrt{n}}\phi(t)\mathbb{H}_2(t) + \mathcal{O}\left(\frac{1}{n}\right).$$

A key observation is that if $X$ is a Binomial random variable with parameters $m$ and $p$. Then, $\sqrt{m}(X - p)/\sqrt{p(1-p)}$ has a lattice distribution with span $1/\sqrt{mp(1-p)}$ and

$$\left\{\frac{\sqrt{m}(X - p)}{\sqrt{p(1-p)}} < t\right\} \tag{C.6}$$

is an event dependent solely on a standardised Binomial distribution. We make two remarks in conclusion:

- Theorem 2 in Feller has $o(1/\sqrt{n})$ in the statement. However, inspecting the proof reveals that the actual rate is $\mathcal{O}(1/n)$.

- The $\mathcal{O}(1/n)$ is independent of the term $t$.

The rest of the proof follows. □

**Proof of Lemma 10**

*Proof.* Recall that by CLT $\bar{F}(\cdot) = \frac{1}{n}\sum_{i=1}^{n} \mathbf{1}[X_i < \cdot] = F(\cdot) + \mathcal{O}_p(1/\sqrt{n})$. Let $f = F'$ be the PDF. We now have,

$$\bar{F}(t_m) = F(t_m) + \mathcal{O}_p\left(\frac{1}{\sqrt{n}}\right)$$

$$(\text{By Taylor Series}) = F(\mu) + (t_m^* - \mu)f(\mu) + \frac{(t_m^* - \mu)^2 f'(\mu)}{2} + o_p((t_m^* - \mu)^2) + \mathcal{O}_p\left(\frac{1}{\sqrt{n}}\right)$$

$$= \frac{1}{2} + \left(\hat{\mu}_n + t\hat{\sigma}_m^{(boot)} - \mu\right)f(\mu) + \frac{(\hat{\mu}_n + t\hat{\sigma}_m^{(boot)} - \mu)^2 f'(\mu)}{2}$$

$$+ o_p((t_m^* - \mu)^2) + \mathcal{O}_p\left(\frac{1}{\sqrt{n}}\right) \tag{C.7}$$

We now carefully substitute the terms. Observe that $\hat{\mu}_n - \mu = \mathcal{O}_p(1/\sqrt{n})$. Furthermore, Lemma 7 implies

$$\left(\hat{\sigma}_m^{(boot)}\right)^2 = \frac{\sigma^2}{m} + \mathcal{O}_p(m^{-3/4}n^{-1/2})$$

Therefore,

$$\hat{\sigma}_m^{(boot)} = \frac{\sigma}{\sqrt{m}} + \mathcal{O}_p\left(m^{-1/4}n^{-1/2}\right).$$

Substituting, we get

$$\hat{\mu}_n + t\hat{\sigma}_m^{(boot)} - \mu = \frac{t\sigma}{\sqrt{m}} + \mathcal{O}_p\left(m^{-1/4}n^{-1/2}\right).$$

and

$$\left(\hat{\mu}_n + t\hat{\sigma}_m^{(boot)} - \mu\right)^2 = \frac{(t\sigma)^2}{m} + \mathcal{O}_p\left(m^{-3/4}n^{-1/2}\right) + \mathcal{O}_p\left(m^{-1/2}n^{-1}\right).$$

$$= \frac{(t\sigma)^2}{m} + \mathcal{O}_p\left(m^{-1/4}n^{-1/2}\right).$$

Substituting this into eq. (C.7) we get

$$\bar{F}(t_m) = \frac{1}{2} + \frac{t\sigma}{\sqrt{m}}f(\mu) + \frac{(t\sigma)^2}{2m}f'(\mu) + \mathcal{O}_p\left(m^{-1/4}n^{-1/2}\right).$$

This completes the proof. $\qquad\square$

**Proof of Lemma 11**

*Proof.* Recall from eq. (C.4) that

$$g(T_m) = \frac{(1 - 2\bar{F}(t_m))}{(\bar{F}(t_m)(1 - \bar{F}(t_m)))^{1/2}} = -\frac{4t\sigma}{\sqrt{m}}f(\mu) - \frac{2(t\sigma)^2}{m}f'(\mu) + \mathcal{O}_p(m^{-3/4}n^{-1/2})$$

Therefore,

$g(T_m)\mathbb{H}_2(t_m^*)\phi(t_m^*)$

$$= \left(-\frac{4t\sigma}{\sqrt{m}}f(\mu) - \frac{2(t\sigma)^2}{m}f'(\mu) + \mathcal{O}_p(m^{-1/4}n^{-1/2})\right)$$

$$\times \left(\mathbb{H}_2(2t\sigma f(\mu))\phi(2t\sigma f(\mu)) - \frac{(t\sigma)^2}{\sqrt{m}}f'(\mu)\phi^{(3)}(-2t\sigma f(\mu)) + \mathcal{O}_p(m^{1/4}n^{-1/2}) + \mathcal{O}_p\left(\frac{1}{m}\right)\right)$$

$$= -\frac{4t\sigma}{\sqrt{m}}f(\mu)\mathbb{H}_2(2t\sigma f(\mu))\phi(2t\sigma f(\mu)) - \frac{2(t\sigma)^2}{m}f'(\mu)\phi^{(3)}(-2t\sigma f(\mu))\mathbb{H}_2(2t\sigma f(\mu))\phi(2t\sigma f(\mu))$$

$$+ \frac{4(t\sigma)^3}{m}f(\mu)f'(\mu)\phi^{(3)}(-2t\sigma f(\mu)) + \mathcal{O}_p(m^{1/4}n^{-1/2}) + o_p\left(\frac{1}{m}\right).$$

Absorbing $o_p(m^{-1})$ into $\mathcal{O}_p(m^{-1})$ terms we now have

$$\frac{g(T_m)\mathbb{H}_2(t_m^*)\phi(t_m^*)}{6\sqrt{m}} = \mathcal{O}_p\left(\frac{1}{m}\right) + \mathcal{O}_p(m^{-1/4}n^{-1/2})$$

This finishes the proof. $\qquad\square$

## D   Supplementary Results and Proofs

The following result is from Shao and Tu [1995].

**Proposition 13.** *Suppose $F$ is a CDF on $\mathbb{R}$ and let $X_1, X_2, \dots$ be i.i.d. $F$. Suppose $\theta = \theta(F)$ is such that $F(\theta) = 1$ and $F(x) < 1$ for all $x < \theta$. Suppose, for some $\delta > 0$ and for all $x$,*

$$P_F\{n^{1/\delta}(\theta - X_{(n)}) > x\} \to e^{-(\frac{x}{\theta})^\delta}$$

*Let*

$$T_n = n^{1/\delta}(\theta - X_{(n)}) \text{ and } T_{m,n}^* = m^{1/\delta}(X_{(n)} - X_{(m)}^*),$$

*and define*

$$H_n(x) = P_F\{T_n \le x\} \quad and \quad H_{Boot,m,n}(x) = P_*\{T_{m,n}^* \le x\}.$$

*Then,*

   (a) *If $m = o(n)$, then $K(H_{Boot,m,n}, H_n) \xrightarrow{P} 0$.*

   (b) *If $m = o\left(\frac{n}{\log\log n}\right)$, then $K(H_{Boot,m,n}, H_n) \xrightarrow{a.s.} 0$.*

**Example 1.** *Let $F(x) = (x/\theta)\mathbf{1}[0 \le x \le \theta]$ be the uniform distribution on $[0, \theta]$ and $X_{(i)}$ is the $i$-th order statistics. $T_n := n\left(\theta - X_{(n)}\right)$. Orthodox bootstrap is inconsistent and m-out-of-n bootstrap is consistent for any $m = o(n)$.*

### D.1   Proof of Example 1

*Proof.* It follows from Proposition 13 that m-out-of-n bootstrap is consistent for any $m = o(n)$. We include a simple proof of the inconsistency of bootstrap. Recall that the survival function of $T_n$ is given by

$$\bar{F}(x) = \mathbb{P}\left(T_n > x\right) = \mathbb{P}\left(n\left(\theta - X_{(n)}\right) > x\right) = \mathbb{P}\left(X_{(n)} < \theta - \frac{x}{n}\right) = \prod_{i=1}^{n}\mathbb{P}\left(X_i < \theta - \frac{x}{n}\right)$$

$$= \left(1 - \frac{x}{n\theta}\right)^n \xrightarrow{n\to\infty} e^{-x/\theta}$$

which is the survival function of an exponential distribution with parameter $\theta$. Now we show that orthodox bootstrap estimator of $T_n$ is inconsistent. Let $X_{(n)}^{boot}$ be the bootstrapped maximum. Then

$$T_{boot} = n\left(X_{(n)}^{boot} - X_{(n)}\right)$$

and

$$\mathbb{P}^{(*)}\left(T_{boot} \le x\right) \ge \mathbb{P}^{(*)}\left(T_{boot} = 0\right) = \mathbb{P}^{(*)}\left(X_{(n)}^{boot} = X_{(n)}\right) = 1 - \mathbb{P}^{(*)}\left(X_{(n)}^{boot} < X_{(n)}\right)$$

$$= 1 - \left(\frac{n-1}{n}\right)^n \xrightarrow{n\to\infty} 1 - e^{-1}.$$

$\qquad\square$

### D.2   Proof of Proposition 1

*Proof.* We recall the following lemma from Ghosh et al. [1984b]

**Lemma 14.** *Let $\{U_i\}_{i\ge 1}$ be a sequence of random variables such that:*

1. $\{U_i\}_{i \geq 1}$ is tight.

2. $E[U_i] \to \infty$, but $E[U_i^2] < \infty$ for all $i$.

*Then,*

$$\mathrm{Var}(U_n) \to \infty.$$

Using this lemma, it is sufficient to show that $\mathbb{E}_n(\hat{\mu}_m^{(boot)} - \hat{\mu}_n)^2 \xrightarrow{a.e} \infty$. The distribution of $\hat{\mu}_m^{(boot)}$ conditioned on the sample $X_1, \ldots, X_n$ can be written as

$$\hat{\mu}_m^{(boot)} = \begin{cases} X_{(n)} & \text{with probability } \frac{1}{n^m} \\ X_{(n-1)} & \text{with probability } 2\left(\frac{n}{2}-1\right)\frac{1}{2n^{m/2+1}} \\ \vdots \\ X_{(1)} & \text{with probability } \frac{1}{n^m} \end{cases}$$

Observe that

$$E_n(\hat{\mu}_m^{(boot)} - \hat{\mu}_n)^2 \geq \frac{1}{n^m}(X_{(n)} - \hat{\mu}_n)^2.$$

Recall that $\hat{\mu}_n \xrightarrow{a.s.} \hat{\mu} = 0$. Thus $\exists \, n_0 \geq 1$ such that $\mathbb{P}(\hat{\mu}_n = 0) = 1$. Letting $n \geq n_0$ we get

$$E_n(\hat{\mu}_m^{(boot)} - \hat{\mu}_n)^2 \geq \frac{X_{(n)}^2}{n^m} \quad \text{almost everywhere.}$$

It is therefore sufficient to show that $X_{(n)}^2/n^m \xrightarrow{a.s.} \infty$. Let $x > 0$ be any positive real number. Then

$$\mathbb{P}_n\left(\frac{X_{(n)}^2}{n^m} \leq x\right) = \mathbb{P}_n\left(X_{(n)} \leq \sqrt{xn^m}\right)$$

$$= \left(\mathbb{P}_n\left(X_1 \leq \sqrt{xn^m}\right)\right)^n$$

$$= F(\sqrt{xn^m})^n$$

$$= \left(1 - \frac{1}{\log(xn^m)}\right)^n$$

$$= \exp\left(n\log\left(1 - \frac{1}{\log(xn^m)}\right)\right).$$

Using the Mclaurin expansion of $\log(1 - x)$, we get

$$\exp\left(n\log\left(1 - \frac{1}{\log(xn^m)}\right)\right) = \exp\left(-n\sum_{k=1}^{\infty}\frac{1}{k!(\log(xn^m))^k}\right)$$

$$\leq \exp\left(-\frac{n}{\log(xn^m)}\right)$$

Therefore,

$$\sum_{n=0}^{\infty}\mathbb{P}_n\left(\frac{X_{(n)}^2}{n^m} \leq x\right) < \infty$$

and it follows using Borel-Cantelli lemma that $X_{(n)}^2/n^m \xrightarrow{a.s.} \infty$. Therefore,

$$E_n(\hat{\mu}_m^{(boot)} - \hat{\mu}_n)^2 \geq \frac{1}{n^m}(X_{(n)} - \hat{\mu}_n)^2 \xrightarrow{a.e} \infty$$

which completes the proof. $\qquad \square$

## D.3 Proof of Corollary 1

*Proof.* For every $t \in \mathbb{R}$ define

$$R_m(t) := \mathbb{P}_n \left( \frac{\sqrt{m}(\hat{\mu}_m^{(\text{boot})} - \hat{\mu}_n)}{\hat{\sigma}_m^{(\text{boot})}} \leq t \right) - \Phi(t).$$

The expansion in Theorem 2 gives

$$R_m(t) = \frac{(t\sigma)^2}{\sqrt{m}} f'(\mu) \phi \left( \frac{t\sigma f(\mu)}{\sqrt{p(1-p)}} \right) + \mathcal{O}_p\left(m^{-1}\right) + \mathcal{O}_p\left(m^{-1/4}n^{-1/2}\right).$$

Recall that

$$\phi \left( \frac{t\sigma f(\mu)}{\sqrt{p(1-p)}} \right) = \mathcal{O} \left( \exp \left( -\frac{t^2 \sigma^2 f(\mu)^2}{p(1-p)} \right) \right)$$

Since the function $g(t) := (t\sigma)^2 \phi(\delta t)$ with $\delta = \sigma f(\mu)/\sqrt{p(1-p)}$ satisfies

$$\sup_{t \in \mathbb{R}} |g(t)| < \infty$$

(because $t^2 e^{-ct^2}$ is bounded for any positive constant $c$), there exists a finite constant $C_0$ (free of $m, n$) such that

$$\sup_{t \in \mathbb{R}} |R_m(t)| \leq C_0 \, |f'(\mu)| \, m^{-1/2} + \mathcal{O}_p\left(m^{-1}\right) + \mathcal{O}_p\left(m^{-1/4}n^{-1/2}\right).$$

Because $m = o(n^\lambda)$ with $\lambda \in (0,1)$, we have $m^{-1/4}n^{-1/2} = o\left(m^{-1/2}\right)$. Hence the leading term $m^{-1/2}$ dominates the remainder terms, and we conclude

$$\sup_{t \in \mathbb{R}} |R_m(t)| = \mathcal{O}_p\left(m^{-1/2}\right),$$

which completes the proof. □

## E   Proof of the DKW-inequality for Markov Chains

### E.1   On VC-dimensions

The notion of VC class is powerful because it covers many interesting classes of functions and ensures suitable properties on the Rademacher complexity. The function $F$ is an envelope for the class $\mathcal{F}$ if $|f(x)| \leq F(x)$ for all $x \in E$ and $f \in \mathcal{F}$. For a metric space $(\mathcal{F}, d)$, the covering number $\mathcal{N}(\varepsilon, \mathcal{F}, d)$ is the minimal number of balls of size $\varepsilon$ needed to cover $\mathcal{F}$. The metric that we use here is the $L_2(Q)$-norm denoted by $\|.\|_{L_2(Q)}$ and given by $\|f\|_{L_2(Q)} = \{\int f^2 dQ\}^{1/2}$.

**Definition 2.** *A countable class $\mathcal{F}$ of measurable functions $E \to \mathbb{R}$ is said to be of VC-type (or Vapnik-Chervonenkis type) for an envelope $F$ and admissible characteristic $(C, v)$ (positive constants) such that $C \geq (3\sqrt{e})^v$ and $v \geq 1$, if for all probability measure $Q$ on $(E, \mathcal{E})$ with $0 < \|F\|_{L_2(Q)} < \infty$ and every $0 < \varepsilon < 1$,*
$$\mathcal{N}\left(\varepsilon \|F\|_{L_2(Q)}, \mathcal{F}, \|.\|_{L_2(Q)}\right) \leq C\varepsilon^{-v}.$$

Let $\mathcal{F}_{\mathcal{S}} := \{\mathbf{1}[\cdot < t], t \in \mathcal{S}\}$ be the set of all half-interval functions. Then, we have the following lemma.

**Lemma 15.** $\mathcal{F}_{[0,1] \cap \mathbb{Q}}$ *is VC with constant envelope 1 and admissible charactaristic $(\mathbb{C}, 2)$ for some universal constant $\mathbb{C} > 4e$.*

Next, we introduce some terminology. We define the Orcliz norm of a random variable $X$ as $\|X\|_{\psi_\alpha} = \text{argmin}\{\lambda > 0 : \mathbb{E}\left[e^{(X/\lambda)^\alpha}\right] \leq 1\}$. We define $\tau_o := \min\{\|\tau(1)\|_{\psi_1}, \|\tau(2)\|_{\psi_1}\}$ and assume that $\tau_o < \infty$.

We now present the following lemma whose proof can be found in SectionE.4

**Lemma 16.** *Let $X_1, \ldots, X_n$ be a Markov chain with stationary distribution $F$, and define*

$$Z := \sup_{f \in \mathcal{F}_{[0,1] \cap \mathbb{Q}}} \left| \sum_{i=1}^{n} \left( f(X_i) - \mathbb{E}_F \left[ f(X) \right] \right) \right|.$$

*Then, for some universal constant $K > 0$,*

$$\mathbb{P}\left( Z > t + KR(\mathcal{F}_{[0,1] \cap \mathbb{Q}}) \right) \leq \exp\left( -\frac{\mathbb{E}_A[\tau_A]}{K} \min\left\{ \frac{t^2}{n\mathbb{E}_A[\tau_A^2]}, \frac{t}{\tau_o^3 \log n} \right\} \right)$$

*where, for any $L > \sqrt{\mathbb{E}_A[\tau_A^2]}$, and $\mathbb{C}_\lambda = 2\mathbb{E}_A[\exp(\tau_A \lambda)]/\lambda$*

$$R(\mathcal{F}_{[0,1] \cap \mathbb{Q}}) = 2(\mathbb{E}_A[\tau_A] + \mathbb{E}_\nu[\tau_A]) + \mathbb{C}\left[ L \log\left( \frac{L}{\sqrt{\mathbb{E}_A[\tau_A^2]}} \right) + \sqrt{n\mathbb{E}_A[\tau_A^2] \log\left( \frac{L}{\sqrt{\mathbb{E}_A[\tau_A^2]}} \right)} \right]$$

$$+ n \exp\left( -L\lambda/2 \right) \mathbb{C}_\lambda.$$

We now present the following proposition whose proof can be found in SectionE.3

**Proposition 17.** *Let $X_1, \ldots, X_n$ be a Markov chain with stationary distribution $F$, and define*

$$Z := \sup_{f \in \mathcal{F}_{[0,1]}} \left| \frac{1}{n} \sum_{i=1}^{n} \left( f(X_i) - \mathbb{E}_F \left[ f(X) \right] \right) \right|.$$

*Then, for some universal constant $K > 0$, and for all $t > 0$*

$$\mathbb{P}(Z > t) \leq \mathbb{C}^\star(\tau, \lambda) \exp\left( -\frac{\mathbb{C}(\tau, \lambda) n t^2}{\log n} \right) \tag{E.1}$$

*where, constants $\mathbb{C}(\tau, \lambda), \mathbb{C}^\star(\tau, \lambda)$ depend only on $\mathbb{E}_A[\tau_A^2]$, $\mathbb{E}_v[\tau_A]$, $\tau_o$, and $\lambda$.*

The following corollary will be useful. It follows easily from Proposition 17 by setting $t = \frac{1}{\sqrt{\mathbb{C}(\tau, \lambda)n}}$ and a simple use of Borel-Cantelli lemma, and is therefore omitted.

**Corollary 4.** *Let $Z_n := \sup_{t \in [0,1]} |\bar{F}(t) - F(t)|$. Then, $Z_n > \sqrt{2/(\mathbb{C}(\tau, \lambda)n)}$ finitely often almost everywhere.*

## E.2 Proofs

### Proof of Lemma 15

*Proof.* We provide a proof for completeness. We begin this proof with some requisite definitions. Given a class of indicator functions $\mathcal{F}$ defined on $\chi$, and a set $\{x_1, \ldots, x_n\} \in \chi^n$, we first define

$$\mathcal{F}(\{x_1, \ldots, x_n\}) := \{(f(x_1), \ldots, f(x_n)) \in \{0,1\}^n : f \in \mathcal{F}\}$$

The *growth function* of the $\mathcal{F}$ is then defined as

$$\Delta_n(\mathcal{F}) = \max_{\{x_1, \ldots, x_n\} \in \chi^n} |\mathcal{F}(\{x_1, \ldots, x_n\})|$$

The **VC-dimension** of $\mathcal{F}$ is then defined as

$$VC(\mathcal{F}) := \operatorname*{argmax}_n \{n : \Delta_n(\mathcal{F}) = 2^n\}$$

We will now show that $VC(\mathcal{F}_{[0,1] \cap \mathbb{Q}}) = 1$. Let $\{x_1, \ldots, x_n\}$ be any ordered sample. That is, $x_1 < x_2 <, \ldots, < x_n$. For any $t \in [0,1] \cap \mathbb{Q}$, observe that $(\mathbf{1}[x_1 < t], \mathbf{1}[x_2 < t], \ldots, \mathbf{1}[x_n < t])$ has the form $(1, 1, 1, \ldots, 1, 0, 0 \ldots, 0)$. In particular, the values of $\mathcal{F}_{[0,1] \cap \mathbb{Q}}(\{x_1, \ldots, x_n\})$ has to be within the following set

$$(0, 0, 0, \ldots, 0),$$
$$(1, 0, 0, \ldots, 0),$$
$$(1, 1, 0, \ldots, 0),$$
$$\vdots$$
$$(1, 1, 1, \ldots, 1)$$

Therefore $\Delta_n(\mathcal{F}_{[0,1] \cap \mathbb{Q}}) = n + 1$. This implies that

$$VC(\mathcal{F}_{[0,1] \cap \mathbb{Q}}) = \underset{n}{\operatorname{argmax}}\{\Delta_n(\mathcal{F}_{[0,1] \cap \mathbb{Q}}) = 2^n\}$$
$$= \underset{n}{\operatorname{argmax}}\{n + 1 = 2^n\}$$
$$= 1.$$

Now, using standard results of covering number bounds, (Theorem 7.8 of Sen [2018], see also Theorem 2.6.4 Van der Vaart [2000]) we have the following result. For some universal constant $\mathbb{C} > 0$

$$\mathcal{N}\left(\varepsilon\|F\|_{L_2(Q)}, \mathcal{F}_{[0,1] \cap \mathbb{Q}}, \|.\|_{L_2(Q)}\right) \leq \mathbb{C} \times VC(\mathcal{F}_{[0,1] \cap \mathbb{Q}})(4e)^{VC(\mathcal{F}_{[0,1] \cap \mathbb{Q}})}\left(\frac{1}{\varepsilon}\right)^{2VC(\mathcal{F}_{[0,1] \cap \mathbb{Q}})}$$
$$\overset{(i)}{\leq} \frac{\mathbb{C}'}{\varepsilon^2}.$$

where $(i)$ follows by substituting $VC(\mathcal{F}_{[0,1] \cap \mathbb{Q}})$. This completes the proof. $\qquad\square$

### E.3   Proof of Proposition 17

*Proof.* Proof of Proposition 17 Since there is always a rational number between any two real numbers, it holds almost everywhere that

$$\sup_{f \in \mathcal{F}_{[0,1]}}\left|\sum_{i=1}^{n}(f(X_i) - \mathbb{E}_F[f(X)])\right| \leq \sup_{f \in \mathcal{F}_{[0,1]} \cap \mathbb{Q}}\left|\sum_{i=1}^{n}(f(X_i) - \mathbb{E}_F[f(X)])\right| + 2$$

Therefore,

$$\mathbb{P}(nZ > nt + KR(\mathcal{F}_{[0,1] \cap \mathbb{Q}}))) \leq \mathbb{P}\left(\underbrace{\sup_{f \in \mathcal{F}_{[0,1]} \cap \mathbb{Q}}\left|\sum_{i=1}^{n}(f(X_i) - \mathbb{E}_F[f(X)])\right| > nt + KR(\mathcal{F}_{[0,1] \cap \mathbb{Q}}) - 2}_{=:A}\right)$$

$t \geq 3/n$ by hypothesis. Therefore $nt - 2 \geq 1$ and it follows using Lemma 16 that the right hand side of the previous equation is bounded above by

$$\exp\left(-\frac{\mathbb{E}_A[\tau_A]}{K}\min\left\{\frac{(nt-2)^2}{n\mathbb{E}_A[\tau_A^2]}, \frac{(nt-2)}{\tau_o^3 \log n}\right\}\right)$$

By setting $L = (2/\lambda)\log(n/2\mathbb{C}_\lambda)$ and observing that under Assumption 1 (EM), $1/(2\mathbb{C}_\lambda) < 1$ we get

$$R(\mathcal{F}_{[0,1] \cap \mathbb{Q}})) \leq 2(\mathbb{E}_A[\tau_A] + \mathbb{E}_\nu[\tau_A]) + \mathbb{C}\left[2\frac{\log(n)}{\lambda}\log\left(\frac{2\log(n)/\lambda}{\sqrt{\mathbb{E}_A[\tau_A^2]}}\right) + \sqrt{n\mathbb{E}_A[\tau_A^2]\log\left(\frac{2\log(n)/\lambda}{\sqrt{\mathbb{E}_A[\tau_A^2]}}\right)}\right]$$

Observe that $\sqrt{\mathbb{E}_A[\tau_A^2]} \geq 1$. Now, with a constant $\mathbb{C}(\tau, \lambda)$ depending on $\lambda$ and $\mathbb{E}_A[\tau_A^2]$, and $\mathbb{E}_v[\tau_A]$ we have with some standard manipulations

$$R(\mathcal{F}_{[0,1] \cap \mathbb{Q}})) \leq \mathbb{C}(\tau, \lambda)\sqrt{n \log n}$$

Then, with $R_n = \mathbb{C}(\tau, \lambda)\sqrt{n \log n}$, we have

$$\mathbb{P}(nZ > nt + KR_n)) \leq \exp\left(-\frac{\mathbb{E}_A[\tau_A]}{K}\min\left\{\frac{(nt-2)^2}{n\mathbb{E}_A[\tau_A^2]}, \frac{(nt-2)}{\tau_o^3 \log n}\right\}\right).$$

Now dividing both sides of $A$ by $n$ and trivially upper bounding 2 by $2K$, we have for some universal constant $K > 0$, and for all $t > 3/n$

$$\mathbb{P}\left(Z > t + KR_n/n\right) \leq \exp\left(-\frac{\mathbb{E}_A[\tau_A]}{K}\min\left\{\frac{(nt-2)^2}{n\mathbb{E}_A[\tau_A^2]}, \frac{(nt-2)}{\tau_o^3 \log n}\right\}\right) \qquad (E.2)$$

where, for some constant $\mathbb{C}(\tau, \lambda)$ depending only on $\mathbb{E}_A[\tau_A^2], \mathbb{E}_v[\tau_A], \lambda$

$$R_n = \mathbb{C}(\tau, \lambda)\sqrt{n \log n}.$$

Next, observe that

$$\mathbb{P}(Z > t) = \mathbb{P}(Z - \mathbb{E}Z > t - \mathbb{E}Z)$$

Since $\mathbb{E}Z < R_n/n = \mathcal{O}(\sqrt{\log n / n})$, there exists a constant $\mathbb{C}'(\tau, \lambda) > 3/n$ such that

$$t - \mathbb{E}Z > t - \mathbb{C}'(\tau, \lambda).$$

Then,

$$\mathbb{P}(Z > t) \le \exp\left(-\frac{\mathbb{E}_A[\tau_A]}{K} \min\left\{\frac{(n(t - \mathbb{C}'(\tau, \lambda)) - 2)^2}{n\mathbb{E}_A[\tau_A^2]}, \frac{(n(t - \mathbb{C}'(\tau, \lambda)) - 2)}{\tau_o^3 \log n}\right\}\right).$$

We now make 2 cases.

**Case I:** When $t > 2\mathbb{C}'(\tau, \lambda)$, $t - \mathbb{C}'(\tau, \lambda) - 2/n > t/2$ and hence

$$\mathbb{P}(Z > t) \le \exp\left(-\frac{\mathbb{E}_A[\tau_A]}{K} \min\left\{\frac{(nt/2)^2}{n\mathbb{E}_A[\tau_A^2]}, \frac{nt/2}{\tau_o^3 \log n}\right\}\right).$$

**Case II:** When $0 < t \le 2\mathbb{C}'(\tau, \lambda)$, there exists a large enough constant $\mathbb{C}^\star(\tau, \lambda)$ such that

$$\mathbb{P}(Z > t) \le \mathbb{C}^\star(\tau, \lambda) \exp\left(-\frac{\mathbb{E}_A[\tau_A]}{K} \min\left\{\frac{(nt/2)^2}{n\mathbb{E}_A[\tau_A^2]}, \frac{nt/2}{\tau_o^3 \log n}\right\}\right).$$

It therefore follows that, for some large enough constant $\mathbb{C}(\tau, \lambda)$ and for all $t > 0$

$$\mathbb{P}(Z > t) \le \mathbb{C}^\star(\tau, \lambda) \exp\left(-\frac{\mathbb{E}_A[\tau_A]}{K} \min\left\{\frac{(nt/2)^2}{n\mathbb{E}_A[\tau_A^2]}, \frac{nt/2}{\tau_o^3 \log n}\right\}\right).$$

It follows that

$$\mathbb{P}(Z > t) \le \mathbb{C}^\star(\tau, \lambda) \exp\left(-\frac{\mathbb{E}_A[\tau_A]}{K} \min\left\{\frac{(nt/2)^2}{n\mathbb{E}_A[\tau_A^2]}, \frac{nt/2}{\tau_o^3 \log n}\right\}\right)$$

$$\le \mathbb{C}^\star(\tau, \lambda) \exp\left(-\frac{\mathbb{E}_A[\tau_A]}{K} \frac{n \min\{t, t^2\}}{4 \log n} \min\left\{\frac{1}{\mathbb{E}_A[\tau_A^2]}, \frac{1}{\tau_o^3}\right\}\right).$$

Define

$$\mathbb{C}(\tau, \lambda) := \frac{\mathbb{E}_A[\tau_A]}{4K} \min\left\{\frac{1}{\mathbb{E}_A[\tau_A^2]}, \frac{1}{\tau_o^3}\right\}.$$

It now follows that, for all $t > 0$

$$\mathbb{P}(Z > t) \le \mathbb{C}^\star(\tau, \lambda) \exp\left(-\frac{\mathbb{C}(\tau, \lambda) n \min\{t, t^2\}}{\log n}\right).$$

$\square$

### E.4  Proof of Lemma 16

*Proof.* Proof of Lemma 16 To prove this lemma, we use Lemma 15 in conjunction with theorems 4, 5, and 6 Bertail and Portier [2019] (see also theorem 7 Adamczak [2008]).

Observe from part (ii) of theorem 4 Bertail and Portier [2019] that under Assumption 1, the Rademacher complexity $R(\mathcal{F}_{[0,1] \cap \mathbb{Q}})$ (as defined in definition 7 Bertail and Portier [2019]) for any class of VC functions with constant envelope $U$ and characteristic $(C_1, v)$ can be upper bounded as

$$R(\mathcal{F}_{[0,1] \cap \mathbb{Q}}) \le \mathbb{C}\left[vLU \log \frac{C_1 LU}{\sigma'} + \sqrt{vn\sigma' \log \frac{C_1 LU}{\sigma'}}\right] + nU \exp(-L\lambda/2)\mathbb{C}_\lambda, \quad \text{(E.3)}$$

where $(\sigma')^2$ is any number such that

$$\sup_{f \in \mathcal{F}_{[0,1] \cap \mathbb{Q}}} \mathbb{E}_A \left[ \left( \sum_{i=1}^{\tau_A} f(X_i) \right)^2 \right] \le (\sigma')^2$$

and $L$ is any number such that $0 < \sigma' < LU$.

Recall from Lemma 15 that the class of all half intervals on rationals $\mathcal{F}_{[0,1] \cap \mathbb{Q}}$ are VC with a constant envelope $U$ and characteristic $(\mathbb{C}, 2)$ for some universal constant $\mathbb{C}$. Substituting this in eq. (E.3), we get

$$R(\mathcal{F}_{[0,1] \cap \mathbb{Q}})) \le \mathbb{C} \left[ 2L \log \frac{\mathbb{C}L}{\sigma'} + \sqrt{2n\sigma' \log \frac{\mathbb{C}L}{\sigma'}} \right] + n \exp(-L\lambda/2)\mathbb{C}_\lambda.$$

Next, we observe that $f(\cdot)$ are indicators of half-intervals. Hence $f(\cdot) \le 1$ and

$$\left( \sum_{i=1}^{\tau_A} f(X_i) \right)^2 \le \tau_A^2.$$

Therefore, choosing $(\sigma')^2 = \mathbb{E}_A[\tau_A^2]$ suffices and we get

$$R(\mathcal{F}_{[0,1] \cap \mathbb{Q}})) \le \mathbb{C} \left[ 2L \log \frac{\mathbb{C}L}{\sqrt{\mathbb{E}_A[\tau_A^2]}} + \sqrt{2n\sqrt{\mathbb{E}_A[\tau_A^2]} \log \frac{\mathbb{C}L}{\sqrt{\mathbb{E}_A[\tau_A^2]}}} \right] + n \exp(-L\lambda/2)\mathbb{C}_\lambda.$$

Finally, substituting this into theorem 5 Bertail and Portier [2019] and trivially substituting $\log(x\mathbb{C}) \le \mathbb{C}\log(x)$ for all large enough constant $\mathbb{C}$, we arrive at the required bound

$$R(\mathcal{F}_{[0,1] \cap \mathbb{Q}})) = 2(\mathbb{E}_A[\tau_A] + \mathbb{E}_\nu[\tau_A]) + \mathbb{C} \left[ L \log \left( \frac{L}{\sqrt{\mathbb{E}_A[\tau_A^2]}} \right) + \sqrt{n\mathbb{E}_A[\tau_A^2] \log \left( \frac{L}{\sqrt{\mathbb{E}_A[\tau_A^2]}} \right)} \right]$$
$$+ n \exp\left(-L\lambda/2\right) \mathbb{C}_\lambda.$$

Now, using the exponential tail bound for the suprema of additive functions of regenerative Markov chains (theorem 6 in Bertail and Portier [2019], or theorem 7 in Adamczak [2008]), we arrive at the conclusion. □

## F   Proof of Theorem 3

*Proof.* As before, we first show that

$$\left( \hat{\sigma}_m^{(boot)} \right)^2 \xrightarrow{a.s.} \frac{q(1-q)}{f^2(\mu)}.$$

Without losing generality, let $\mu$ be the unique median, let $\varepsilon > 0$ and let $q = 1/2$. The proof follows very similarly in other cases. Let $X_1, X_2, \ldots, X_n$ be a sample from a Markov chain satisfying Assumption 1, and recall (for instance from Remark 5 Bertail and Portier [2019]) that it is equivalent to assuming geometric ergodicity of the Markov chain. Furthermore, let $\{X_i^\dagger\}$ be another Markov chain with the same transition density starting from the stationary distribution $F$.

Next recall the well-known fact (see for instance [Jones, 2004, pg. 304]. See also Meyn and Tweedie [2012]) that for a geometrically ergodic Markov chain, the coupling time $T$ (formally defined in eq. (F.1)) is finite a.e. In other words, almost everywhere,

$$T := \inf \left\{ n : X_n = X_n^\dagger \right\} < M. \tag{F.1}$$

**Step 1.** For any $\varepsilon > 0$, we first observe that

$$\sum_{i \geq 0} \mathbb{P}\left(|X_i| > i^{1/\alpha}\varepsilon\right) = \sum_{i \geq 0} \mathbb{P}\left(|X_i| > i^{1/\alpha}\varepsilon \bigcap T > i\right) + \sum_{i \geq 0} \mathbb{P}\left(|X_i| > i^{1/\alpha}\varepsilon \bigcap T \leq i\right)$$

$$\leq \sum_{i \geq 0} \mathbb{P}\left(|X_i| > i^{1/\alpha}\varepsilon \bigcap T \leq i\right) + \sum_{i \geq 0} \mathbb{P}(T > i)$$

$$\stackrel{(i)}{\leq} \sum_{i \geq 0} \mathbb{P}\left(|X_i^\dagger| > i^{1/\alpha}\varepsilon \bigcap T \leq i\right) + \sum_{i \geq 0} \mathbb{P}(T > i)$$

$$\leq \sum_{i \geq 0} \mathbb{P}\left(|X_i^\dagger| > i^{1/\alpha}\varepsilon\right) + \sum_{i \geq 0} \mathbb{P}(T > i)$$

$$= \sum_{i \geq 0} \mathbb{P}\left(|X_1^\dagger| > i^{1/\alpha}\varepsilon\right) + \sum_{i \geq 0} \mathbb{P}(T > i)$$

$$\leq \mathbb{E}_F[|X_1^\dagger|^\alpha] + \mathbb{E}[T]$$

$$\stackrel{(ii)}{<} \infty.$$

where $(i)$ follows by coupling property, $(ii)$ follows since $T < M$ almost everywhere for some $M$, and the other inequalities are trivial.

Now, using Borel-Cantelli lemma, $X_i > i^{1/\alpha}\varepsilon$ only finitely many times with probability 1. Therefore,

$$\frac{|X_{(n)}| + |X_{(1)}|}{n^{1/\alpha}} \xrightarrow{a.s.} 0. \tag{F.2}$$

We next establish that $m(\hat{\mu}_m^{(boot)} - \hat{\mu}_n)^2$ is uniformly integrable. As in the proof of Theorem 1, we have

$$\mathbb{E}_n \left|\sqrt{m}(\hat{\mu}_m^{(boot)} - \hat{\mu}_n)\right|^{2+\delta} = (1+\delta)\int_0^\infty t^{1+\delta}\mathbb{P}_n\left(\sqrt{m}|\hat{\mu}_m^{(boot)} - \hat{\mu}_n| > t\right)dt, \tag{F.3}$$

with

$$\{\sqrt{m}(\hat{\mu}_m^{(boot)} - \hat{\mu}_n) > t\} = \left\{\frac{1}{2} - \frac{1}{2m} - \bar{F}(\hat{\mu}_n + t/\sqrt{m}) > F_n^{(m)}(\hat{\mu}_n + t/\sqrt{m}) - \bar{F}(\hat{\mu}_n + t/\sqrt{m})\right\}, \tag{F.4}$$

Let $c(\alpha) = 1/\alpha + 1/2$. As before, we divide the proof into two cases.

**Step II** ($t \in [1, c(\alpha)\sqrt{\log m}]$)**:** We begin by analysing the left hand side of the event described in eq. (A.2).

$$\frac{1}{2} - \frac{1}{2m} - \bar{F}(\hat{\mu}_n + t/\sqrt{m}) = \underbrace{\frac{1}{2} - \frac{1}{2m} - \bar{F}(\hat{\mu}_n)}_{=:A_{m,n}} + \underbrace{F(\hat{\mu}_n) - F(\hat{\mu}_n + t/\sqrt{m})}_{C_{m,n}}$$

$$+ \underbrace{\bar{F}(\hat{\mu}_n) - F(\hat{\mu}_n) + F(\hat{\mu}_n + t/\sqrt{m}) - \bar{F}(\hat{\mu}_n + t/\sqrt{m})}_{B_{m,n}}$$

We begin with $B_{m,n}$. Using triangle inequality, we have,

$$|B_{m,n}| \leq \left|\bar{F}(\hat{\mu}_n) - F(\hat{\mu}_n)\right| + \left|F(\hat{\mu}_n + t/\sqrt{m}) - \bar{F}(\hat{\mu}_n + t/\sqrt{m})\right|.$$

Using Corollary 4 twice, we obtain $|B_{m,n}| \leq \sqrt{8/(\mathbb{C}(\tau, \lambda)n)}$ almost everywhere. Since $m \leq n$, it follows that

$$|B_{m,n}| \leq \frac{2}{\sqrt{2m\mathbb{C}(\tau, \lambda)}} \quad \text{almost everywhere.}$$

Turning to $C_{m,n}$ and using Taylor series expansion, we get that

$$C_{m,n} = -\frac{t}{\sqrt{m}} f(\hat{\mu}_n) + o(t/m)$$

$$= -\frac{t}{\sqrt{m}} f(\mu) + \frac{t}{\sqrt{m}} (\mu - \hat{\mu}_n) f'(\mu) + o\left(t/m\right).$$

$$= -\frac{t}{\sqrt{m}} f(\mu) + \frac{t}{\sqrt{m}} (\mu - \hat{\mu}_n) f'(\mu) + \mathcal{O}\left(t/\sqrt{m}\right). \tag{F.5}$$

Recall that under the hypothesis of the theorem, $f$ is continuously differentiable around the median. Using [Bertail and Portier, 2019, Proposition 11], we have that

$$|\hat{\mu}_n - \mu| \leq \mathcal{O}_p\left(\sqrt{\frac{\log\log n}{n}}\right) \quad \text{almost everywhere.}$$

Turning to $A_{m,n}$, observe that $\mu$ is the median. Therefore, $F(\mu) = 1/2$. We get similarly as before,

$$\frac{1}{2} - \frac{1}{2m} - \bar{F}(\hat{\mu}_n) = -\frac{1}{2m} + F(\mu) - \bar{F}(\hat{\mu}_n)$$

$$= -\frac{1}{2m} + \underbrace{(\hat{\mu}_n - \mu) f(\chi)}_{=: \text{ Term 1}} + \underbrace{F(\hat{\mu}_n) - \bar{F}(\hat{\mu}_n)}_{=: \text{ Term 2}}$$

where $\chi \in \left[\mu - |\mu - \hat{\mu}_n|, \, \mu + |\mu - \hat{\mu}_n|\right]$.

To upper bound Term 1, observe that from the hypothesis of the Theorem that $f$ is bounded and continuous around $\mu$. Since

$$\chi \in \left[\mu - |\mu - \hat{\mu}_n|, \, \mu + |\mu - \hat{\mu}_n|\right]$$

it follows that $f(\chi) < L$ almost everywhere for all values of $n$ large enough for some non-random constant $L$. Since by [Bertail and Portier, 2019, Proposition 11], we have that

$$|\hat{\mu}_n - \mu| \leq \mathcal{O}_p\left(\sqrt{\frac{\log\log n}{n}}\right) \quad \text{almost everywhere} \tag{F.6}$$

it follows that

$$\text{Term 1} \leq \mathcal{O}_p\left(\sqrt{\frac{\log\log n}{n}}\right).$$

almost everywhere. It follows from Corollary 4 that Term 2 $\leq \mathcal{O}_p(1/\sqrt{n})$. Combining,

$$|A_{m,n}| \leq \mathcal{O}_p\left(\sqrt{\frac{\log\log n}{n}}\right).$$

Combining all three steps, we get that

$$\frac{1}{2} - \frac{1}{2m} - \bar{F}(\hat{\mu}_n + t/\sqrt{m}) = -\frac{t}{\sqrt{m}} + \mathcal{O}\left(\frac{t}{\sqrt{m}}\right). \tag{F.7}$$

For the sake of convenience, we denote $\hat{\mu}_n + t/\sqrt{m}$ by $t_m$. It now follows from eq. (F.4) that

$$\{\sqrt{m}(\hat{\mu}_m^{(boot)} - \hat{\mu}_n) > t\} \subseteq \left\{-\frac{t}{\sqrt{m}} + \frac{t\xi}{\sqrt{m}} \geq F_n^{(m)}(t_m) - \bar{F}(t_m)\right\} \tag{F.8}$$

for some finite real positive number $\xi$ which only depends on the sample $X_1, \ldots, X_n$, and hence is constant under $\mathbb{P}_n$. Recall that, $X_i^* \overset{i.i.d}{\sim} \bar{F}$. By using Lemma 3, we have from equation eq. (F.4)

$$\mathbb{P}_n\left(\sqrt{m}(\hat{\mu}_m^{(boot)} - \hat{\mu}_n) > t\right) \leq \mathbb{P}_n\left(-\frac{t}{\sqrt{m}} + \frac{\xi}{\sqrt{m}\log m} \geq F_n^{(m)}(t_m) - \bar{F}(t_m)\right)$$

$$\leq \frac{3}{t^4(1-\xi)^4}.$$

Substituting this into eq. (F.3), we have shown that the integral is finite.

**Step III** ($t > c(\alpha)\sqrt{\log m}$) For $t > c(\alpha)(\log m)^{1/2}$, and for all large $m$ almost surely, it follows by using equations F.7 and F.8 that,

$$\mathbb{P}_n\big(\sqrt{m}\,(\mu_m^{(boot)} - \hat{\mu}_n) > t\big) \tag{F.9}$$

$$\leq \mathbb{P}_n\big(\sqrt{m}\,(\mu_m^{(boot)} - \hat{\mu}_n) > c(\alpha)(\log m)^{1/2}\big)$$

$$\leq \mathbb{P}_n\left(F_n^{(m)}\big(\hat{\mu}_n + c(\alpha)(\log m)^{1/2}\,m^{-1/2}\big) - \bar{F}\big(\hat{\mu}_n + c(\alpha)(\log m)^{1/2}\,m^{-1/2}\big) \leq -\,e\,c(\alpha)\sqrt{\frac{\log m}{m}}\right).$$

Choose $c(\alpha) = \frac{1}{\alpha} + \frac{1}{2}$. the rest follows similarly as before.

Hence, for large $m$,

$$\int_{c(\alpha)(\log m)^{1/2}}^{m^{1/\alpha + 1/2}} t^{1+\delta}\mathbb{P}_n\big(\sqrt{m}\,(\mu_m^{(boot)} - \hat{\mu}_n) > t\big)\mathrm{d}t = \mathcal{O}(1) \quad \text{a.s.}$$

Using the previous fact and eq. (F.2) from Step 1, we have

$$\mathbb{P}_n\big(\sqrt{m}\,(\mu_m^{(boot)} - \hat{\mu}_n) > m^{1/2 + 1/\alpha}\big) = 0 \quad \text{a.s. for all large } n.$$

Thus, we have proved our result for $\sqrt{m}\,(\mu_m^{(boot)} - \mu)$. Similar arguments handle $-\sqrt{m}\,(\mu_m^{(boot)} - \mu)$. This completes the proof.

Next, we state the following Lemma which is proved below

**Lemma 18.** *Let $\mu$ be the unique $p$-th quantile of a Markov chain satisfying Assumption 1. Then,*

$$\sqrt{m}\,\big(\hat{\mu}_m^{(boot)} - \hat{\mu}_n\big) \xrightarrow{d} \mathcal{N}\left(0, \frac{p(1-p)}{f^2(\mu)}\right).$$

Using this lemma, and the fact that

$$\big(\hat{\sigma}_m^{(boot)}\big)^2 \xrightarrow{a.s.} \frac{q(1-q)}{f^2(\mu)}$$

we have via Slutsky's theorem

$$\frac{\sqrt{m}\,\big(\hat{\mu}_m^{(boot)} - \hat{\mu}_n\big)}{\big(\hat{\sigma}_m^{(boot)}\big)} \xrightarrow{d} \mathcal{N}(0, 1).$$

This proves Theorem 3. We now prove Lemma 18. $\qquad\square$

## F.1 Proof of Lemma 18

*Proof.* As before, we shorthand $t_m = \hat{\mu}_n + t/\sqrt{m}$ and begin following the steps of eq. (A.2) to get

$$\mathbb{P}_n\{\sqrt{m}(\hat{\mu}_m^{(boot)} - \hat{\mu}_n) < t\} = \mathbb{P}_n\left(\frac{\sqrt{m}(F_n^{(m)}(t_m) - \bar{F}(t_m))}{[\bar{F}(t_m)(1 - \bar{F}(t_m))]^{1/2}} \geq \underbrace{\frac{\sqrt{m}\,(p - \bar{F}(t_m))}{[\bar{F}(t_m)(1 - \bar{F}(t_m))]^{1/2}}}_{=:t_m^*}\right).$$

Observe that given the sample $F_n^{(m)}(x) \sim \text{Binomial}(m, \bar{F}(x))$ and let $n$ be large enough. Using a central limit theorem for Binomial random variables, observe that

$$\frac{\sqrt{m}(F_n^{(m)}(t_m) - \bar{F}(t_m))}{[\bar{F}(t_m)(1 - \bar{F}(t_m))]^{1/2}} \stackrel{d}{=} \mathcal{N}(0, 1) + \mathcal{R}_m$$

where $\mathcal{R}_m$ is a remainder term that decays in probability to 0 as $m \to \infty$.

Let $m$ be large enough such that $\mathcal{R}_m < \varepsilon$. It follows that,

$$\mathbb{P}_n\left( \frac{\sqrt{m}(F_n^{(m)}(t_m) - \bar{F}(t_m))}{[\bar{F}(t_m)(1 - \bar{F}(t_m))]^{1/2}} \geq \underbrace{\frac{\sqrt{m}\,(p - \bar{F}(t_m))}{[\bar{F}(t_m)(1 - \bar{F}(t_m))]^{1/2}}}_{=:t_m^*} \right) = \mathbb{P}_n\left(\mathcal{N}(0,1) > t_m^* - \varepsilon\right)$$

$$= 1 - \Phi(t_m^* - \varepsilon).$$

We will now show that

$$t_m^* \xrightarrow{m,n} t\sqrt{\frac{f(\mu)^2}{p(1-p)}}.$$

Using a method similar to the derivation in eq. (F.7), we have

$$\bar{F}(t_m) = p\frac{m-1}{m} + \frac{t}{\sqrt{m}} + \mathcal{O}\left(\frac{t}{\sqrt{m}}\right)$$

$$\xrightarrow{m} p$$

and,

$$[\bar{F}(t_m)(1 - \bar{F}(t_m))]^{1/2} \xrightarrow{m,n} \sqrt{p(1-p)}.$$

Now we address the numerator. Since a geometrically ergodic Markov chain is also ergodic, we have by Birkhoff's ergodic theorem [Dasgupta, 2008, Theorem 3.4], that

$$\bar{F}(t_m) = F(t_m) + \mathcal{O}_p(1/n).$$

Using a first-order Taylor series expansion on $F(t_m)$, we have

$$F(t_m) = F(\mu) + \left(\hat{\mu}_n + t/\sqrt{m} - \mu\right)f(\chi)$$
$$= p + \left(\hat{\mu}_n + t/\sqrt{m} - \mu\right)f(\chi),$$

where

$$\chi \in \left[\mu - |\mu - \hat{\mu}_n|,\ \mu + |\mu - \hat{\mu}_n|\right].$$

Recall from eq. (F.6) that $|\mu - \hat{\mu}_n| = o_p(1)$. Since $f$ is continuous around $\mu$, it follows using continuous mapping theorem that

$$F(t_m) \xrightarrow{n} p + tf(\mu)/\sqrt{m}$$

and

$$\sqrt{m}\left(p - \bar{F}(t_m)\right) \xrightarrow{n} -tf(\mu).$$

Therefore,

$$\Phi(t_m^* - \varepsilon) = \Phi(-tf(\mu)/\sqrt{p(1-p)} - \varepsilon)$$

and

$$1 - \Phi(t_m^* - \varepsilon) = \Phi(tf(\mu)/\sqrt{p(1-p)} + \varepsilon).$$

$\varepsilon$ is arbitrary. We now use the fact that $\Phi(tf(\mu)/\sqrt{p(1-p)})$ is the CDF of a gaussian distribution with mean 0 and variance $p(1-p)/f^2(\mu)$. This completes the proof. $\qquad\square$

# G   Proof of Corollaries

## G.1   Proof of Corollary 2

We only need to show that the MH algorithm satisfies Assumption 1. The proof is analogous to the proof of Proposition 9 in Bertail and Portier [2019], and we provide it for completion.

**Proposition 19** (Uniform minorisation). *Let $P$ be a Markov transition kernel on $(\mathbb{R}^d, \mathcal{B}(\mathbb{R}^d))$ and let $\Phi$ be a positive measure whose support*

$$E = \operatorname{supp}(\Phi)$$

*is bounded, convex, and has non–empty interior. Assume that there exists $\varepsilon > 0$ such that for every $x \in E$*

$$P(x, \mathrm{d}y) \geq \mathbf{1}_{B(x,\varepsilon)}(y)\, \Phi(\mathrm{d}y).$$

*Then there are constants $C > 0$ and $n \geq 1$ for which*

$$P^n(x, A) \geq C\, \Phi(A), \qquad \forall x \in E, \forall A \in \mathcal{B}(E). \tag{G.1}$$

*Proof.* The argument is split into four parts.

Fix $0 < \gamma \leq \eta$. There is a constant $c > 0$ such that for all $x, y \in E$

$$\int \mathbf{1}_{B(x,\eta)}(x_1)\, \mathbf{1}_{B(y,\gamma)}(x_1)\, \Phi(\mathrm{d}x_1) \geq c\, \mathbf{1}_{B(x,\eta+\gamma/4)}(y). \tag{G.2}$$

Indeed, if $y \notin B(x, \eta + \gamma/4)$ the right–hand side vanishes, so only the case $y \in B(x, \eta + \gamma/4)$ matters. Since $E$ is convex, one can select a point $m$ on the segment connecting $x$ and $y$ such that $B(m, \gamma/4) \subset B(x, \eta) \cap B(y, \gamma)$; this yields the bound with $c := \inf_{m \in E} \Phi\big(B(m, \gamma/4)\big) > 0$.

Iterating (G.2) shows that for each $n \geq 1$ there is $C_n > 0$ satisfying

$$\int \cdots \int \mathbf{1}_{B(x,\varepsilon)}(x_1)\, \mathbf{1}_{B(x_2,\varepsilon)}(x_1) \cdots \mathbf{1}_{B(x_n,\varepsilon)}(x_{n-1})\, \mathbf{1}_{B(y,\varepsilon)}(x_n)\, \Phi(\mathrm{d}x_1) \cdots \Phi(\mathrm{d}x_n)$$
$$\geq C_n\, \mathbf{1}_{B(x,\varepsilon(1+n/4))}(y).$$

Choose $n$ so large that $\varepsilon(1 + n/4)$ exceeds $\sup_{u,v \in E} \|u - v\|$. Then $B(x, \varepsilon(1 + n/4))$ contains $E$ for every $x \in E$, and the integral in Step 2 is bounded below by a positive constant $C_n$ that does not depend on $x$ or $y$.

Combining the minorisation assumption on $P$ with the integral lower bound from Step 3 yields, for any $x \in E$ and measurable $A \subset E$,

$$P^n(x, A) \geq \int \cdots \int \mathbf{1}_{B(x,\varepsilon)}(x_1) \cdots \mathbf{1}_{B(y,\varepsilon)}(x_n)\, \mathbf{1}_A(y)\, \Phi(\mathrm{d}x_1) \cdots \Phi(\mathrm{d}x_n)\Phi(\mathrm{d}y) \geq C_n\, \Phi(A).$$

Setting $C := C_n$ completes the proof. $\qquad\square$

Applying Proposition 19 to the random–walk Metropolis–Hastings (MH) kernel yields exponential moment bounds, summarised below.

**Proposition 20.** *Let $\pi$ be a bounded density supported by a bounded, convex set $E \subset \mathbb{R}^d$ with non–empty interior. Assume the proposal density $q$ of the random–walk MH algorithm satisfies*

$$q(x, y) \geq b\, \mathbf{1}_{B(x,\varepsilon)}(y), \qquad \forall x, y \in \mathbb{R}^d,$$

*for some constants $b, \varepsilon > 0$. Then the resulting MH chain is aperiodic, $\pi$–irreducible, and enjoys the exponential moment property* (EM).

*Proof.* Because the acceptance probability obeys $\rho(x, y) \geq \pi(y)/\|\pi\|_\infty$, the MH kernel satisfies

$$P(x, \mathrm{d}y) \geq \|\pi\|_\infty^{-1}\, \mathbf{1}_{B(x,\varepsilon)}(y)\, \pi(y)\, \mathrm{d}y, \qquad x \in E.$$

Thus every ball of radius $\varepsilon/2$ is $\pi$–small in the sense of Roberts and Tweedie [1996], which implies aperiodicity. Applying Proposition 19 with $\Phi(\mathrm{d}y) = \|\pi\|_\infty^{-1}\pi(y)\, \mathrm{d}y$ gives the minorisation (G.1); Theorem 16.0.2 of Meyn and Tweedie [2012] then verifies Assumption 1. $\qquad\square$

### G.2 Proof of Corollary 3

*Proof.* We first show that $X_i$ is geometrically ergodic. We will show that it is $\phi$-mixing, which implies geometric ergodicity [Bradley, 2005].

Recall from Bradley [2005, Eq. 1.2] the definition of the $\alpha$- and $\phi$–mixing coefficients $\alpha_{i,j}$, $\phi_{i,j}$ and from Bradley [2005, Eq. 1.11] that $\alpha_{i,j} \leq \phi_{i,j}$, so it suffices to bound $\phi_{i,j}$.

Introduce the *weak-mixing* coefficients

$$\bar{\theta}_{i,j} := \sup_{s_1,s_2 \in [0,1]} \left\| \Pr(X_j \in \cdot \mid X_i = s_1) - \Pr(X_j \in \cdot \mid X_i = s_2) \right\|_{\mathrm{TV}},$$

for $1 \leq i < j \leq n$. By Banerjee et al. [2025, Lemma 1], $\phi_{i,j} \leq \bar{\theta}_{i,j}$, so bounding $\bar{\theta}_{i,j}$ is enough.

Let $P_i(x, \cdot)$ be the (time-dependent) Markov kernel at step $i$ and $p_i(x,t)$ its density w.r.t. Lebesgue measure on the state space $[0,1]$. The chain is assumed to satisfy a *Doeblin minorisation*:

$$\inf_{x \in [0,1]} p_i(x,t) \geq \kappa, \qquad \forall t \in [0,1],\ \forall i \geq 1. \tag{G.3}$$

For a Markov kernel $P$, its Dobrushin coefficient is

$$\delta(P) := 1 - \inf_{x_1,x_2 \in [0,1]} \int_{[0,1]} \min\{p(x_1,t),\, p(x_2,t)\}\, dt.$$

Hajnal's theorem for products of stochastic matrices [Hajnal and Bartlett, 1958, Theorem 2] gives

$$\bar{\theta}_{i,j} \leq \prod_{p=i}^{j-1} \delta(P_p).$$

Using the minorisation (G.3),

$$\int_{[0,1]} \min\{p_p(x_1,t), p_p(x_2,t)\}\, dt \geq \int_{[0,1]} \kappa\, dt = \kappa,$$

so $\delta(P_p) \leq 1 - \kappa$ for every $p$. Hence

$$\bar{\theta}_{i,j} \leq (1 - \kappa)^{j-i}.$$

Combining the above inequalities yields

$$\alpha_{i,j} \leq \phi_{i,j} \leq \bar{\theta}_{i,j} \leq (1 - \kappa)^{j-i}.$$

Thus, $X_i$ is geometrically ergodic. Since $r$ is one-one onto, it is invertible. Therefore, $r_i$ is also geometrically ergodic. Therefore, this satisfies Assumption 1. The differentiability assumption can be easily seen to hold. The rest of the proof follows. □

## H  Simulations

We conducted Monte Carlo experiments with sample sizes $n \in \{10000, 20000, 50000\}$. For each $n$, we examined three block sizes $m = \log n$, $n^{1/3}$, and $\sqrt{n}$. Following the procedure described in Section 1 (formula for the bootstrap statistic) and Equation (B.2) (closed-form variance estimate), we repeatedly evaluated the test statistic $T$ to approximate its sampling distribution.

Three data-generating mechanisms were considered:

1. i.i.d. Gaussian observations;

2. Observations obtained by reflecting a simple random walk onto the interval $[-1,1]$;

3. a Markov chain generated by the Random-Walk Metropolis–Hastings (RWMH) algorithm (see Section 5), employing a Laplace proposal distribution on a standard normal target density.

For each configuration, we measured the discrepancy between the empirical distribution of $T$ and the standard normal distribution using the Kolmogorov–Smirnov (KS) distance.

Table 1: Results for $n = 10,000$

| $m$ | $m$ value | Case | Mean($T$) | Var($T$) | KS |
|---|---|---|---|---|---|
| $\log(n)$ | 9 | Gaussian | -0.030959 | 0.992566 | 0.033 |
| $\log(n)$ | 9 | Reflected RW | 0.041787 | 0.784624 | 0.069 |
| $\log(n)$ | 9 | MH RW | 0.005385 | 0.937350 | 0.057 |
| $n^{1/3}$ | 21 | Gaussian | -0.053800 | 1.114615 | 0.028 |
| $n^{1/3}$ | 21 | Reflected RW | -0.006052 | 0.965366 | 0.040 |
| $n^{1/3}$ | 21 | MH RW | -0.045641 | 0.862029 | 0.028 |
| $\sqrt{n}$ | 100 | Gaussian | -0.037565 | 1.059072 | 0.039 |
| $\sqrt{n}$ | 100 | Reflected RW | 0.021268 | 0.905786 | 0.069 |
| $\sqrt{n}$ | 100 | MH RW | -0.081919 | 0.978142 | 0.061 |

Table 2: Results for $n = 20,000$

| $m$ | $m$ value | Case | Mean($T$) | Var($T$) | KS |
|---|---|---|---|---|---|
| $\log(n)$ | 9 | Gaussian | -0.035324 | 0.975723 | 0.033 |
| $\log(n)$ | 9 | Reflected RW | 0.015931 | 0.771220 | 0.052 |
| $\log(n)$ | 9 | MH RW | -0.003805 | 0.977506 | 0.033 |
| $n^{1/3}$ | 27 | Gaussian | -0.025948 | 0.976584 | 0.028 |
| $n^{1/3}$ | 27 | Reflected RW | -0.011587 | 0.956381 | 0.027 |
| $n^{1/3}$ | 27 | MH RW | 0.006103 | 0.970455 | 0.032 |
| $\sqrt{n}$ | 141 | Gaussian | 0.048491 | 0.972170 | 0.049 |
| $\sqrt{n}$ | 141 | Reflected RW | 0.027213 | 1.045555 | 0.032 |
| $\sqrt{n}$ | 141 | MH RW | 0.025495 | 0.967099 | 0.040 |

Table 3: Results for $n = 50,000$

| $m$ | $m$ value | Case | Mean($T$) | Var($T$) | KS |
|---|---|---|---|---|---|
| $\log(n)$ | 10 | Gaussian | 0.023658 | 0.920129 | 0.039 |
| $\log(n)$ | 10 | Reflected RW | 0.024419 | 0.794373 | 0.044 |
| $\log(n)$ | 10 | MH RW | 0.011908 | 0.860669 | 0.025 |
| $n^{1/3}$ | 36 | Gaussian | -0.001047 | 0.959031 | 0.028 |
| $n^{1/3}$ | 36 | Reflected RW | 0.001399 | 0.872152 | 0.027 |
| $n^{1/3}$ | 36 | MH RW | 0.007086 | 0.971482 | 0.022 |
| $\sqrt{n}$ | 223 | Gaussian | -0.051904 | 1.046060 | 0.051 |
| $\sqrt{n}$ | 223 | Reflected RW | -0.028429 | 0.957545 | 0.028 |
| $\sqrt{n}$ | 223 | MH RW | -0.021154 | 0.967758 | 0.023 |

