# OpenReview forum: "Small Resamples, Sharp Guarantees: Convergence Rates for Resampled Studentized Quantile Estimators"
_NeurIPS.cc/2025/Conference — NeurIPS 2025 poster_

### Official Review · Reviewer_gHZF · 2025-06-11

**Clarity:** 4
**Significance:** 4
**Originality:** 3
**Rating:** 5
**Confidence:** 3

**Summary:**

This is a theoretical paper about the n-out-of-m bootstrap (i.e., drawing $m \ll n$ samples for each bootstrap iteration). The paper specifically considers a model free setting, where nothing is assumed about the data-generating process except for a weak moment condition. The paper points out that despite the over 30 year history of the m-out-of-n bootstrap, no (correct) theory exists proving its limiting behavior. The paper derives a central limit theorem (CLT) for the n-out-of-m bootstrap, giving a correct description of its limiting behavior. The paper goes on to prove a Berry-Esseen bound for the m-out-of-n bootstrap's behavior, showing that the distribution of the estimator converges to normal at a $1/\sqrt{m}$ rate. The paper then does a significant amount of extra theoretical work to extend these results to derive confidence intervals for the rewards in a Markov decision process, which the paper notes are the first of their type.

**Questions:**

1. What should a practitioner who is considering using the m-out-of-n bootstrap take away from reading this paper?

2. In Section 5, are the $X_i$ sampled from the MCMC algorithm or from the stationary distribution?

3. What are $\psi, \Psi$, and $E$ in Section 5?

**Ethical Concerns:**

["NO or VERY MINOR ethics concerns only"]

**Final Justification:**

The authors' responses to my and other reviewers' comments makes me more confident about the strength of the paper, and I've maintained my vote for an acceptance.

**Limitations:**

Yes

**Quality:**

3

**Strengths And Weaknesses:**

# Strengths

I think this is overall a pretty solid paper. There were a few points that I thought it did especially well:

1. The comparison to previous literature is very precise, and I found it very clear how this work compared with previous papers in this space.

2. Good proof sketches are provided; I really felt like I got a good sense of how each proof was going to proceed from the outline in the main text.

3. Study of necessiry of conditions: a very weak moment condition is assumed (there exists some $\alpha > 1$ such that $E|X|^\alpha < \infty$) to prove the theory. Instead of noting that this is a very standard and mild assumption in the literature, the paper provides a counterexample to the theory when this moment condition does not hold.

4. Overall easy to read: despite being a purely theoretical paper about a typically mathematical topic, I found the paper pretty easy to follow. And I would say that I'm knowledgeable about the bootstrap, but not an expert on its theory by any means.

# Weaknesses

I had two medium-level weaknesses:

1. I think the paper is a little mean towards the previous work of Cheung and Lee (2005) (CL). The paper points out that part of its contribution is to correct a mistake in CL. As part of doing that, of course it needs to say CL is wrong. But I thought the paper crossed the line towards being rude:
   "[The] most damning [gap in previous literature] seems to be Cheung and Lee (2005), which which gives an incorrect variance formula"

   I would be more polite and also just generally avoid using the word "damning", especially to describe other people's mistakes.

2. Lack of practical advice. The paper doesn't provide any practical advice about the use of the m-out-of-n bootstrap. I know this is a purely theoretical paper, but I usually think of the goal of theory as providing guidance for practice. I think the paper could dedicate a little space to discussing the takeaways of the results. E.g., should we be spending our time on the m-out-of-n bootstrap? Or should we just use the central limit theorem to derive uncertainty? On the face of it, it seems like the CLT variance shrinks more quickly, as it goes with $1/\sqrt{n}$, whereas the variance from Theorem 1 goes with $1/\sqrt{m}$.

I also have a collection of minor-level comments; I don't think these meaningfully contribute to the strength of the paper, with the exception of \#8 and \#12, which I was really unclear on.

1. There's some jargon used in the introduction that I don't think was defined: "DEA estimator" and "PTRF paper" (I had to google what PTRF is -- I'm pretty sure typical NeurIPS attendees will not know this journal by name, and definitely not by abbreviation). Additionally, the paper uses the term "Bachman–Landau notations" to refer to big-$O$ and little-$o$ notation, which isn't standard as far as I know for computer science venues.

2. Line 93: "For example, In" the capitalization here is a typo.

3. $\mathcal{C}$ is used to denote a constant. I think it's a little bit of a mistake to use that symbol to represent anything besides the complex numbers.

4. Theorem 1 uses $q$ to denote the quantile of interest, whereas the rest of the paper uses $p$.

5. Step 1 of the  sketch of Theorem 1 uses $i$ without defining it or it being clear from the context what it means.

6. Step 2 of the sketch of Theorem 1 states a Hoeffding bound, but I don't see $t$ on the right hand side of it. Should the $\delta$ be $t$?

7. Line 234: "... and empirical distribution $\bar F$" -- is this supposed to be "empirical stationary distribution"?

8. Section 4.3 on extending to Markov chains has some undefined notation: $\psi, \Psi, E$.

9. In section 5, $q$ is a positive function satisfying some conditions. Do these conditions hold for all $x$? Or just some?

10. In Section 5, I don't think $q_0$ was ever defined.

11. At the beginning of Section 5, $x$ is real-valued, but later it takes values in $R \times R$.

12. Corollary 2 assumes that $X_1, \dots, X_n$ are sampled from the trajectory of a MCMC algorithm. But at the start of section 5, it's assumed that $X_1, \dots, X_n$ are sampled from the stationary distribution of some Markov chain. These aren't the same thing, so which are we supposed to be assuming when reading through Section 5?

13. I think Corollary 3 should state its assumptions within the theorem environment.

---

> ### Author Rebuttal · Authors · 2025-07-25
>
> We thank the anonymous reviewer for the valuable comments. We are grateful for getting helpful suggestions, and a list of typos. We will address all the feedback we received in the final version of this paper and, of course, fix the typos and add necessary clarifications.
>
> # Response to Strengths and Weaknesses
>
> __Tone towards prior work:__ We completely agree with the reviewer in hindsight that it came out harsher than we intended and thank the reviewer for highlighting this important point. We intend to replace it with ``One significant discrepancy seems to be in Cheung and Lee [2005], which gives an incorrect variance formula for the m-out-of-n estimator of sample quantiles without derivations or citations." which we hope will be more acceptable.
>
> **Lack of practical advice:**  We thank the reviewer for this question and would like to take this opportunity to remind them that the practical benefits of bootstrapping has been relatively well studied and we refer to the citations on line 107 of our paper (Singh and Xie [2008], Davison and Hinkley [1997], Freedman et al. [2007], Dasgupta [2008]) for references. As we also point out to reviewer gjZu, we also provide a way to choose $m$ based on our current theory. Our main objective here (as we mentioned in line 116, line 142, etc.) was to attack one specific corner of this problem which lay largely unanswered: the theoretical properties of the studentized bootstrapping of the sample quantiles for IID and regenerative Markov chains. Our work had four primary motivations (as outlined between lines 63-80, section 3.1, and section 4.2)  (1) acquitting the oddities of Studentization in bootstrapping the median, (2) give minimal conditions for asymptotic normality, (3) produce second order Edgeworth expansions, (4) extend bootstrap to Markov chains to align it with modern tasks like RL and MCMC.
>
> **Minor comments and typos:** We thank the reviewer for pointing out the typos and will correct those, as well as provide full names for the acronyms for DEA (Data Envelopment Analysis), and PTRF (Probability Theory and Related Fields) in the revised version of the paper. Now we proceed to answer some of the specific comments pointed out by the reviewer.
>
> **Point 5**, We thank the reviewer for pointing out this typo. $i^{1/\alpha}$ should be $n^{1/\alpha}$, and hopefully this clears up the discrepancy.
>
> **Point 8** We thank the reviewer for pointing out this important omission. The definitions are largely technical, and the key objective is to build towards the definition of regeneration, which endows the Markov chain with desirable properties. Therefore, while the context is interesting, including all formal definitions will probably be distracting for our purposes, and we believe that pointing towards classical textbooks is the correct approach. We would  point the readers to the definitions in Meyn and Tweedie  (Markov chains and Stochastic Stability, defined in pages 89 and 103 respectively). That said, there is an intuitive explanation which we include in the revised version, to better illustrate the key concepts to a prospective reader.
>
> >$\psi$ and $\Psi$ are somewhat common in Markov chain literature to denote $\psi$-irredicibility and $\Psi$-atoms respectively [Bertail and Portier, 2019]. We will not require the formal definitions, and therefore we refer the interested reader to classical textbooks on Makov chains [pages 89 and 103, Meyn and Tweedie, 2013] for more details. Intuitively, $\psi$-irredicibility extends the classical notion of irreducibility for finite state Markov chains to infinite state spaces, whereas $\Psi$-atoms are sets from which the transitions behave homogeneously with some probability measure $\Psi$. For finite state Markov chains, the atoms are the individual states (i.e. singletons). Informally, a regenerating Markov chain is a $\psi$-irreducible Markov chain which has at least one $\Psi$-atom that is repeatedly visited (with the inter-arrival times termed as the regeneration time). Conditions on the moments of the regeneration time charactises the ergodic properties of the Markov chain.
>
>      Citation: P. Bertail and F. Portier. Rademacher complexity for Markov chains: Applications to kernel smoothing and Metropolis–Hastings. Bernoulli, 25(4B):39123938, Nov. 2019. ISSN 1350-7265. doi: 10.3150/19-BEJ1115.
>
> **Point 10**, The confusion regarding $q_0$ is a result of a typo. $q(\cdot,\cdot)$ is a bivariate function, and $q_0$ is an univariate function such that $q(x,y)=q_0(x-y)$. This notation standard in the MCMC literature, and we will correct it in the revised version.
>
> **Point 12**, the reviewer is indeed correct that for this example we have considered a stationary MCMC which is a limitation of our current work. However, this example is primarily a demonstration of the applicability of our assumptions and we believe that the theory can be generalised with significantly more tedium to non-stationary ergodic Markov chains. Considering that bootstrap for Markov chains is sparsely studied (see line 238), this is a good topic for future studies.
>
> # Response to Questions:
>
> Regarding questions 2 and 3, we hope you find our responses to the weaknesses sections satisfactory. As for the first question—what a practitioner should take away—we offer the following 60-second summary:
>
> Practitioners interested in sampling from the distribution of the sample quantiles may continue to rely on the bootstrap with confidence. Our results suggest that standard practice often remains valid even in dependent settings, such as Markov chains encountered in MCMC or offline reinforcement learning. Moreover, the m-out-of-n bootstrap provides a robust alternative when orthodox bootstrap is compromised by lack of symmetry (as we discuss in lines 224-228).
>
> **_Once again, we thank the reviewer for such a great evaluation of our work._**

---

> > ### Comment · Reviewer_gHZF · 2025-08-03
> >
> > Thanks for the replies here and to the other reviewers! I have just have a few follow-up comments/questions:
> >
> > 1. On the choice of $m$: thank you for pointing out the discussion on lines 218-228. It might be worth re-phrasing this to be clearer in terms of the advice it's giving to users of the bootstrap. Is line 229-230 suggesting that we always set $m = n^{1/3}$? If so, I would just make this explicit in the text. Maybe getting even more at what other reviewers asked in terms of heuristics for setting $m$ -- what about the constant? Can we just set $m = n^{1/3}$ and call it a day? Or should we sometimes put a constant on there? I think it's fine to not have an answer here, I would just be really clear what we do / don't know about practice given the current results.
> > 2. That one of the practical results is "practitioners ... may continue to rely on the bootstrap with confidence" is a major finding! I would highlight this in the paper.
> > 3. I'm still a little confused about the meaning of Corollary 1. This seems to just show the convergence rate is $1/\sqrt{m}$, which is worse than $1/\sqrt{n}$, which is what we get from the CLT. So is Corollary 1 telling us we should ignore the $m$-out-of-$n$ bootstrap in favor of the CLT? Or is it just a theoretical check that the $m-n$ is meeting a bear minimum? I'm guessing the later, but I would just make this clear.
> > 4. I don't necessarily agree with kcDW that the paper is poorly written overall, but I kind of take their point that the paper does jump a little quickly into the deep end (I do remember thinking when reading the paper for the first time "OK I guess we're skipping the intro!") It could be nice to have the typical few paragraphs laying out the goal of the paper and the significance of the results before jumping into notation. I don't think this is necessary for the success of the paper, but this might make the paper accessible to a broader portion of the NeurIPS audience.

---

> > > ### Author Response · Authors · 2025-08-05
> > > **Further Clarifications**
> > >
> > > We thank the anonymous reviewer for their thoughtful clarifications and are pleased to respond to their questions.
> > >
> > > - On the choice of $m$:
> > >   We believe setting $m = c n^{1/3}$ for some universal constant $c$ would work well in practice. The precise choice of $c$ is somewhat ambiguous, as finite-sample convergence rates in nonparametric statistics, unlike asymptotic rates, are generally characterized up to universal constants (see *A Gentle Introduction to Empirical Process Theory* by Bodhisattva Sen). In practice, (from the lens of a non-parametric statistician):
> > >   - For moderate sample sizes where $ m > 30$, setting $c = 1$ typically performs well.
> > >   - For small $n$, a larger constant (i.e., $c > 1$ ) may be advisable.
> > >   - For large $n$, even $c \ll 1$ may suffice in achieving desirable performance.
> > >
> > >
> > > - We thank the reviewer for this suggestion and will highlight the summary in the conclusion section of the revised manuscript.
> > >
> > > - If someone could calculate the variances, (perhaps if the parameters completely identify the distribution), then CLT will always beat bootstrap. However, in practice, one typically estimates the variances via bootstrap. The reviewer is indeed correct that about their assertion regarding this result being a theoretical check, and we understand how the writing could have caused this confusion, and would make this point clear in the final manuscript.
> > >
> > > - We appreciate the reviewer’s comments, as well as those from other reviewers. We believe that the modifications proposed in our responses (e.g., to reviewer gHZF’s question, gjZu’s weakness 2, and others) will substantially improve the clarity and presentation in the final version. We will also make further clarifications about the significance of our results in the Introduction.

---

> > > > ### Comment · Reviewer_gHZF · 2025-08-07
> > > >
> > > > This all sounds great to me; thank you for all the responses! This totally addresses all of my issues.
> > > >
> > > > Just a side-note on harping about practicality in what's clearly a very theoretical paper: I very much think pure theory is valuable. But I also think it's good to tie it back to practice as much as possible. And this is pure theory about a statistical method *widely used in practice*. So to the extent that it's possible, I think it makes the paper even stronger to give small snippets of practical advice that fall out of the theory (or call out places where the theory provides some advice but doesn't provide the full picture -- maybe like the selection of the constant $c$ in the scaling $m = c n^{1/3}$!).
> > > >
> > > > Anyway, I've updated my review to affirm my vote for accepting the paper.
> > > >
> > > > Best,
> > > > gHZF

---

### Official Review · Reviewer_gjZu · 2025-06-29

**Clarity:** 2
**Significance:** 3
**Originality:** 2
**Rating:** 4
**Confidence:** 4

**Summary:**

This work investigates the $m$-out-of-$n$ bootstrap, a resampling method that draws a subsample of size $m\ll n$ from a dataset of size $n$, focusing on estimating sample statistics. The authors developed a rigorous second-order theory for Studentized versions of the bootstrap quantile estimator under relatively general conditions. They proved variance consistency and the central limit theorem (CLT), derived a parameter-free Edgeworth expansion with an $O(m^{-1/2})$ convergence rate, and established a Berry-Esseen bound. These techniques and results are applicable to both IID and regenerative Markov chain settings. Additionally, they correct errors and inconsistencies in previous variance formulas. The work is complemented by illustrative applications in Random Walk MH and Offline Ergodic MDPs.

**Questions:**

Please carefully review the points, questions, and suggestions listed in other sections, including the identified strengths, noted weaknesses, and areas for clarification or improvement, as they aim to enhance the paper's technical rigor, practical relevance, and overall presentation.

**Ethical Concerns:**

["NO or VERY MINOR ethics concerns only"]

**Final Justification:**

My current score of 4 already indicates that I am inclined to accept the paper. The rebuttal helps to improve the paper and provides more clarity, but does not invalidate the claims made in my original review. Hence, I choose to keep my original score.

Below are recaps of some points made during the author-reviewer discussion.

- **The paper lacks detailed empirical validation.**
I agree that the paper's results have been proven. And "validation" in my original review mainly refers to "verifying if the theory applies to practical scenarios (e.g., for RL and MCMC)," where some of the assumptions may not entirely hold, but remain true to some extent.

- **Restrictiveness of assumption 1.**
I acknowledge the authors' inability to address such "restrictiveness" in the current submission. And I look forward to seeing generalized results in the authors' or other researchers' follow-up work.

**Limitations:**

The work is mainly theoretical and does not pose a direct societal risk. However, practical uses in sensitive areas (such as finance and healthcare) should consider the constraints of the required moment conditions and the potential for variance inflation in extreme, heavy-tailed situations. The authors also included a section on "limitations and outlook," which reads well.

**Paper Formatting Concerns:**

The paper is dense and seems mathematically rigorous, but some sections (like Section 4) could benefit from an intermediate summary to better guide readers.

Figures or simulation plots showing convergence behavior or other theoretical representations would strengthen the narrative.

**Quality:**

3

**Strengths And Weaknesses:**

Assessment of strengths:
1. The paper addresses a well-known but underexplored setup: quantile estimation using the Studentized m-out-of-n bootstrap method. The results are technically sound and seem to fill a genuine gap in second-order theory.
2. The derivation steps are clear, as shown in the proof sketches of Theorems 1 and 2. The authors also thoroughly explain the implications of the main results and the intuitions behind them.
3. The assumptions are broad, including finite $\alpha$-order moments and weak smoothness conditions, making the results applicable to other problems.
4. The authors extended the approach to Markov chain data, which is well-motivated for RL and MCMC settings, and they corrected previous errors and inconsistencies in the literature.

Assessment of weaknesses:
1. While the theoretical contributions are non-trivial, the paper lacks detailed empirical validation, although some practical implications (e.g., for RL and MCMC) are discussed.
2. The counterexample used to justify the necessity of moment conditions (Proposition 1) is insightful but somewhat abrupt. It does not imply that a moment condition is necessary. A broader discussion on the characteristics that heavy-tailed distributions must have for the variance of the bootstrap estimator to approach infinity would be helpful.
3. Relying on regenerative structure and conditions in Assumption 1 could limit the generality of the Markov chain results. Would it be possible to include a broader discussion on the limitations and restrictiveness of these assumptions?
 4. I wonder if the authors have some non-heuristic, principled methods to guide the choice of $m$, especially those that are adaptive to practical settings.

---

> ### Author Rebuttal · Authors · 2025-07-26
>
> Thank you for your careful review and meticulous reading and assessment. We are very happy that you pointed out a few weaknesses which we would like to address below.
>
> # Response to Strengths and Weaknesses
>
>
> **1. The paper lacks detailed empirical validation** - Regarding the empirical validation, we respectfully note that the principal contribution of this manuscript is theoretical. The core results have been established through rigorous mathematical proofs; therefore, additional numerical experiments are not required to validate their correctness. We are also pleased that you found our attempt demonstrate the practicality of our assumptions in machine learning tasks like RL and MCMC to be useful. Therefore, especially when considering the current length of the paper, we do not find that simulations to be significantly more illuminating. However, we still agree about the need for some empirical illustrations in the form of simulation, and we hope you find our response below satisfactory.
>
> We conducted Monte Carlo experiments with sample sizes n ∈ {10000,20000,50000}. For each n we examined three block sizes $m = log n, n^{1/3}, and √n$. Following the procedure described in lines 33–39 (formula for the bootstrap statistic), and Equation (B.2) (closed form variance estimate), we repeatedly evaluated the test statistic T to approximate its sampling distribution.
>
> Three data-generating mechanisms were considered:
>
> 1. i.i.d. Gaussian observations;
> 2. i.i.d. observations obtained by reflecting a simple random walk onto the interval [−1, 1];
> 3. a Markov chain generated by the Random-Walk Metropolis–Hastings (RWMH) algorithm (see line 305-312), employing a Laplace proposal distribution on a standard normal target density.
>
> For each configuration we measured the discrepancy between the empirical distribution of T and the standard normal distribution using the Kolmogorov–Smirnov (KS) distance.
>
> n = 10K
>
> | m |   m value | Case         | Mean(T)   | Var(T)   | KS   |
> |-------------|-----|--------------|-----------|----------|------|
> | $\log(n)$   |   9 | Gaussian     | −0.030959 | 0.992566 | 0.033 |
> | $\log(n)$   |   9 | Reflected RW |  0.041787 | 0.784624 | 0.069 |
> | $\log(n)$   |   9 | MH RW        |  0.005385 | 0.937350 | 0.057 |
> | $n^{1/3}$   |  21 | Gaussian     | −0.053800 | 1.114615 | 0.028 |
> | $n^{1/3}$   |  21 | Reflected RW | −0.006052 | 0.965366 | 0.040 |
> | $n^{1/3}$   |  21 | MH RW        | −0.045641 | 0.862029 | 0.028 |
> | $\sqrt{n}$  | 100 | Gaussian     | −0.037565 | 1.059072 | 0.039 |
> | $\sqrt{n}$  | 100 | Reflected RW |  0.021268 | 0.905786 | 0.069 |
> | $\sqrt{n}$  | 100 | MH RW        | −0.081919 | 0.978142 | 0.061 |
> ---
> n = 20K
>
> | m |   m value | Case         | Mean(T)   | Var(T)   | KS   |
> |-------------|-----|--------------|-----------|----------|------|
> | $\log(n)$   |   9 | Gaussian     | −0.035324 | 0.975723 | 0.033 |
> | $\log(n)$   |   9 | Reflected RW |  0.015931 | 0.771220 | 0.052 |
> | $\log(n)$   |   9 | MH RW        | −0.003805 | 0.977506 | 0.033 |
> | $n^{1/3}$   |  27 | Gaussian     | −0.025948 | 0.976584 | 0.028 |
> | $n^{1/3}$   |  27 | Reflected RW | −0.011587 | 0.956381 | 0.027 |
> | $n^{1/3}$   |  27 | MH RW        |  0.006103 | 0.970455 | 0.032 |
> | $\sqrt{n}$  | 141 | Gaussian     |  0.048491 | 0.972170 | 0.049 |
> | $\sqrt{n}$  | 141 | Reflected RW |  0.027213 | 1.045555 | 0.032 |
> | $\sqrt{n}$  | 141 | MH RW        |  0.025495 | 0.967099 | 0.040 |
>
> n = 50K
>
> | m |   m value | Case         | Mean(T)   | Var(T)   | KS   |
> |-------------|-----|--------------|-----------|----------|------|
> | $\log(n)$   |  10 | Gaussian     |  0.023658 | 0.920129 | 0.039 |
> | $\log(n)$   |  10 | Reflected RW |  0.024419 | 0.794373 | 0.044 |
> | $\log(n)$   |  10 | MH RW        |  0.011908 | 0.860669 | 0.025 |
> | $n^{1/3}$   |  36 | Gaussian     | −0.001047 | 0.959031 | 0.028 |
> | $n^{1/3}$   |  36 | Reflected RW |  0.001399 | 0.872152 | 0.027 |
> | $n^{1/3}$   |  36 | MH RW        |  0.007086 | 0.971482 | 0.022 |
> | $\sqrt{n}$  | 223 | Gaussian     | −0.051904 | 1.046060 | 0.051 |
> | $\sqrt{n}$  | 223 | Reflected RW | −0.028429 | 0.957545 | 0.028 |
> | $\sqrt{n}$  | 223 | MH RW        | −0.021154 | 0.967758 | 0.023 |
> ---
>
> **2. The counter-example is abrupt** - We agree with the reviewer that the counterexample does not imply that the moment condition is necessary, and this, along with the language in Section 4.1 creates some confusion. We have **revised** the section to more carefully explain the point of the counterexample as can be seen below.
>
> >**Importance of the Moment Condition**
> We briefly discuss the importance of the condition  $\mathbb{E}\bigl[|X_{1}|^{\alpha}\bigr] < \infty$
> in **Theorem&nbsp;1**. A close inspection of the proof indicates that we really only need the condition in eq.&nbsp;(A.1). However, we found such statements terse and un-illuminating.
>
> >On the other hand, the following counter-example shows that one cannot outright discard all tail conditions. Note that this does **not** imply that the condition $\mathbb{E}\bigl[|X_{1}|^{\alpha}\bigr] < \infty$ is a *necessary* condition, but rather demonstrates that Theorem&nbsp;1 cannot be expected to hold for arbitrarily heavy-tailed distributions. Although this phenomenon has been observed via simulation&nbsp;[Sakov, 1998], we provide what is, to our knowledge, the first theoretical justification.
>
> **3. Restrictiveness of assumption 1** - We thank the reviewer for pointing out this important comment and agree on the necessity of discussing the regenerative condition, and we will revise the ``limitations and outlook" section in the following manner to include it.
>
> >Our guarantees still assume mild smoothness and leave open the precise heavy‐tail boundary and non‐stationary processes, and adaptive variants remain compelling directions for future study. The question regarding the joint normality of the estimators also remains an important open problem. Finally, it would be desirable to derive an Edgeworth expansion of the Studentized bootstrap estimator for Markov chains. However, the theory of Edgeworth expansions on lattices for regenerating Markov chains is sparse (in particular, we could not find a usable counterpart to Proposition 12 for Markov chains). Deriving such a result is out of scope for this (already lengthy) paper, and we plan to investigate it in future work.
>
> >Another important assumption is that of regeneration. We note that the exponential moment condition in Assumption 1 is equivalent to the more classical geometric ergodicity of Markov chains (see Theorem 16.0.2 in Meyn and Tweedie [2009]). It is known that a weaker polynomial moment condition is equivalent to arithmatically mixing for Markov chains (see Bolthausen [1982]), and a corresponding result in this regime seems plausible, and warrants future investigation.
>
>     Bolthausen, E. (1982). On the central limit theorem for stationary mixing random fields. Ann. Probab.
>     10 1047–1050. MR0672305
>     Meyn, S. and Tweedie, R.L. (2009). Markov Chains and Stochastic Stability, 2nd ed. Cambridge: Cambridge Univ. Press. With a prologue by Peter W. Glynn. MR2509253
>
> **4. Non‑heuristic, principled methods to guide the choice of $m$** - We do provide such a guidance (barring a small typo) in line 229 with $m=O(n^{1/3})$ as follows:
>
> > Finally, the optimal choice of m seems to be $m = O(n^{1/3})$ which is known to be the minimax rate for estimating the population median [Bickel and Sakov, 2008].
>
> **Formatting:** We thank the reviewer for raising this point and will include the following summary of Section 4 between lines 160 and 161 for better guiding of the prospective readers
>
> > This section presents Theorem 1 on the asymptotic normality of the Studentized bootstrap statistic under a mild moment condition, with a proof sketch (full details in Appendix A). Proposition 1 provides a counterexample in the absence of this condition. Theorem 2 establishes an Edgeworth expansion for the statistic.
>
> Finally, we cordially thank you again for reading our work in-depth and we very much welcome your further suggestion to improve our final version.
>
> **_We hope we have addressed all the questions, suggestions and weaknesses listed and kindly request you to revise your score._**

---

> > ### Comment · Reviewer_gjZu · 2025-08-06
> > **Appreciate the revisions. Some feedback from the reviewer.**
> >
> > ### Feedback on "The paper lacks detailed empirical validation"
> > Yes, I agree that the paper's results have been proven. And "validation" in my original review mainly refers to "verifying if the theory applies to practical scenarios (e.g., for RL and MCMC)," where some of the assumptions may not entirely hold, but remain true to some extent. That said, the additional experiments are still helpful. Please try to incorporate them into the paper or its appendix.
> >
> > ### Feedback on "The counterexample is abrupt"
> > Thanks for making the revision and confirming that "the counterexample does not imply that the moment condition is necessary," which aligns with my original comment.
> >
> > ### Feedback on "Restrictiveness of assumption 1"
> > Thanks for the additional context. I look forward to seeing generalized results in your or other researchers' follow-up work. Given the limited time, I understand that it may not be feasible to address such "restrictiveness" in the current submission.
> >
> > ### On score revision.
> > My current score of 4 already indicates that I am inclined to accept the paper. The rebuttal helps to improve the paper and provides more clarity, but does not invalidate the claims made in my original review. Hence, I choose to keep my original score. Thanks.

---

### Official Review · Reviewer_NAQc · 2025-07-02

**Clarity:** 4
**Significance:** 2
**Originality:** 3
**Rating:** 4
**Confidence:** 3

**Summary:**

This paper is well written and well presented. It provides sharp consistency and CLT results for m our of n boot strap estimators of the quantile.  The significance is
1) the results hold in practically motivated setting where the number of bootstrap sample m is large but not as large as the number of samples n; and
2) the the assumptions on the underlying distribution are minimal (a tail bound and a local smoothness condition).


Comments:

- Line 51 Please define DEA estimator

- Theorem 1 - It is also required that $m \to \infty$ correct?  If so, this condition should be made explicit since $m = o(n)$ includes the case $m  = O(1)$.

- Line 350 - "join normality" -> "joint normality"

- I recommend more careful language with respect to what has and has not been done previously.  It's not clear to me that analysis of Gribkova and Helmers [2007] in the regime $m \gg n$ should be called a "deficiency". Also, the statement in line 239 that "no work attempted to study the effects of bootstrapping to create confidence intervals for statistics that are of interest in tasks like reinforcement learning or MCMC." is impossible to validate.

**Questions:**

Is there a reason that the tail conditions cannot be alleviated via truncations techniques?

**Ethical Concerns:**

["NO or VERY MINOR ethics concerns only"]

**Final Justification:**

I agree with the weaknesses raised by other reviewers, particularly KcDW

While I appreciate the mathematically precise analysis, the practical motivation for this work is still unclear to me. If the main contribution of this paper is that it fills in some missing gaps in the theoretical literature that is perfectly fine (though perhaps a bit narrow). If there are meaningful practical implications (partially for the setting of quantiles covered by the theory), they should be highlighted. Since I am not aware of applications where computing the empirical quantiles is the main bottleneck, the purported gains in this direction are not obvious to me.

**Limitations:**

Yes.

**Paper Formatting Concerns:**

no concerns.

**Quality:**

4

**Strengths And Weaknesses:**

Strengths:
- This paper fills in gaps in the literature w.r.t bootstrap.
- The details analysis of Markov chains goes beyong the standing IID setting in a meaningful way.
- The paper is well writing and the proofs careful and precise.

Weakness:
- The results require smoothenss conditions on the distribution (e.g., reward). What happens if these are vioiated? For applications in reinforcement learning, this rules out reward functions that may be of interest.
- The interest in this topic within the neurips community might be limited.
- The paper does not make a strong case for the importance of this analysis. There are no empirical results. The authors claim that prior work is impractical because it requires m to large to be computationally practical. But the theoretical results in this paper are valid only in the limit as m goes to infinity, so the same argument would seem to apply here. If one wants to make a convincing argument it would help to mention the computational complexity and typical scalings.

---

> ### Author Rebuttal · Authors · 2025-07-26
>
> Thank you for your careful review and meticulous reading and assessment. We will be happy to address the few weaknesses you have pointed out. You have provided us an opportunity to discuss answers to a few very valuable questions.
>
> **Typos pointed out in the summary:** We want to thank the reviewer for pointing out the minor typos, including acronyms where DEA stands for Data Envelopment Analysis, and added clarification that in our context, the convergence in distribution $\xrightarrow{d}$ is when both $m$ and $n$ is going to infinity.
>
> # Response to Strengths and Weaknesses
>
> **1. Smoothness assumption on reward function in the MDP example:** - We would like to highlight three important aspects of this example.
>
> a. We have found that most literature on reinforcement learning theory assumes finite state-action, and rigorous theoretical treatment of RL as a continuous MDP is somewhat rarer. In the finite setting, the reward function can be described as a finite dimensional vector with minimal assumptions. Therefore, while generalising, we made a largely simplifying assumption that serves to illustrate the amenability of the regenerative assumption towards the analysis of the reward function of MDP's. If one looks at the proof, what we really require is the ergodicity of the reward function, which follows as a consequence of this assumption and some calculations. However, we think directly assuming that the rewards are ergodic is somewhat terse and un-illuminating.
>
> b. We agree with the reviewer about the broader need to have theoretical results on resampling of MDP's under weak assumptions, which we believe is possible with considerably more work. However, since this was a corollary to a more general theorem (Theorem 3), it made sense for us to somewhat simplify the problem for the sake of illustration. We are working on one follow up work which builds upon the theory of the current paper to tackle resampling in offline RL under significantly more generality. Given the rapidly rising popularity treating MDP's as ergodic Markov chains (see for example, Fabien and Maillard, [2022]), we think our current work will also be relevant for other prospective authors in this field.
>
>     Pesquerel, Fabien, and Odalric-Ambrym Maillard. "IMED-RL: Regret optimal learning of ergodic Markov decision processes." Advances in Neural Information Processing Systems 35 (2022): 26363-26374.
>
> **2. The interest in this topic within the NeurIPS community might be limited.** - We would like to respectfully point out to the reviewer that some of the *other reviewers explicitly agree on the interest of this paper to the NeurIPS community*. While resampling and empirical process theory for regenerative Markov chains in isolation align more with probability theory, we provide illustrative examples for MCMC and RL, which we think are of strong interest to the NeurIPS community. As you and other reviewers note in the review, this paper extends the theory beyond IID in a meaningful way and is well motivated for machine learning. We hope that our rebuttal also helps convince you of the merits of this paper through the lenses of modern ML tasks.
>
> **3. Importance of this analysis and empirical results** - We address this in several parts.
>
> a. **Practicality over prior work:** We would like to take this opportunity to clarify some misunderstanding. What we really meant to say in the paper (line 58-59) was that prior theoretical work assumes $m\gg n$, which is not only unnecessary (as demonstrated in Theorem 2), but also which defeats the entire practical motivation of resampling fewer than $n$ samples [Bickel et al. 1992], and thus is uncommon in practice (see also the numerical simulations outlined below which uses $m=o(n)$).
>
> b. **Computational complexity:** We agree about the need to clarify the computational complexity of estimating the resampled quantile and we will add the following line as a remark between lines 42 and 43 in the revised manuscript.
>
> > The computational complexity of resampling and of estimating the quantile (via selection algorithm) is $O(m)$.
>
> c. **Asymptotics:** At this point, we would like to gently remind the reviewer that Theorem 2 (which concerns Edgeworth expansion) is *not asymptotic*, but finite sample. One of our goals in this paper (as mentioned in Section 3.1) was to demonstrate theoretically that $m\ll n$ is actually sufficient to derive an Edgeworth expansion for Studentized quantiles which we felt was an important gap left by Gribkova and Helmers.
>
>     P. J. Bickel, F. Götze, and W. R. van Zwet. Resampling fewer than n observations: gains, losses, and approximations. Statistica Sinica, 2(1):1–31, 1992
>
> **4. Exposition:** As also pointed out in the strengths section your review, our main motivation was to study the practical necessity of $m\ll n$. However, we completely agree with the reviewer about the need to reword "deficiency" to accurately point out the gaps in literature, and **change the lines from 58-61** into
>
> >"Gribkova and Helmers [2007] derive an Edgeworth expansion for Studentized trimmed means in the non‑classical $m\gg n$ regime, offering theoretical insights but leaving unanswered the setting when $m\ll m$ which motivates subsampling. Our work addresses these practically relevant open questions that were left unanswered in previous studies by supplying corrected rates and Edgeworth expansions tailored to the practically relevant $m\ll m$ regime."
>
> We also agree with the reviewer on the topic regarding line 239 and will remove it from our revised manuscript.
>
> **Empirical Results:**   We would also like to gently underline the fact that all our theory is **exact** and free from unknowable constants, in the sense that the asymptotic distributions are free of unknowable parameters, and first order expansions are precise. Furthermore, we theoretically demonstrate the reasonableness of our assumptions with examples in Section 5. Therefore, especially when considering the current length of the paper, we do not find that simulations to be significantly more illuminating. However, we still agree about the need for some empirical illustrations in the form of simulation, and we hope you find our response to weakness 1. of reviewer gjZu satisfactory.
>
> # Response to Questions
>
> **Is there a reason that the tail conditions cannot be alleviated via truncations techniques?** - We do not think that the tail condition be removed via more sophisticated analysis, and provide a counter-example (Proposition 1) in the paper to address this point. We do agree with the reviewer that the condition can probably be weakened, and discuss this in the paper (see lines 183-185, as well as 194-201). However, the assumption is already mild, and weakening it seems to require new analytical machinery, which makes it beyond the scope of the current work.
>
>
> **_We hope we have addressed all the questions, suggestions and weaknesses listed and kindly request you to revise your score._**

---

> > ### Comment · Reviewer_NAQc · 2025-08-04
> >
> > I agree with the weaknesses raised by other reviewers, particularly KcDW
> >
> > While I appreciate the mathematically precise analysis, the practical motivation for this work is still unclear to me. If the main contribution of this paper is that it fills in some missing gaps in the theoretical literature that is perfectly fine (though perhaps a bit narrow). If there are meaningful practical implications (partially for the setting of quantiles covered by the theory), they should be highlighted. Since I am not aware of applications where computing the empirical quantiles is the main bottleneck, the purported gains in this direction are not obvious to me.

---

> ### Author Response · Authors · 2025-08-05
> **Response to Reviewer – Clarifications on Rebuttal, Practical Relevance, Scope, and Computation**
>
> We thank the reviewer for acknowledging the rebuttal.
>
> ### **Addressing Reviewer Concerns**
> With regard to the first point, we would like to respectfully note that we have **addressed all weaknesses raised by other reviewers** in our initial responses and we hope that you found those responses to have satisfactorily clarified any outstanding concerns.
>
> We would like to clarify a potential misunderstanding: our point was not that **computation is the bottleneck** when $n$ is large, but rather that **in practice**, even for moderately sized datasets (e.g., $n \sim 10^5$), practitioners routinely use the $m$-out-of-$n$ bootstrap. Thus, as we note on **line 53** of the manuscript, it was surprising that despite its widespread use, the $m$-out-of-$n$ bootstrap remained **theoretically underdeveloped**. This gap is what our work seeks to fill. We also note that reviewer **gHZF** explicitly highlighted this point as a **major strength** of our paper.
>
> #### **Scope of Contribution**
> While we appreciate the reviewer’s observation that our work fills important gaps in the literature, we respectfully submit that the assessment understates the **full extent of our contributions**. As elaborated in **Section 3.1**, the objective of this paper is to provide a **complete second-order theory** for the Studentized $m$-out-of-$n$ bootstrap estimator of sample quantiles, which we organize into three main contributions:
>
> - **Filling theoretical gaps:** We show that the resampled variance converges to its population counterpart and that the Studentized statistic is asymptotically Gaussian. We also derive a **one-term, correctly centered Edgeworth expansion** with a sharp remainder bound.
>
> - **Extending to dependent data:** We extend the theory to regenerating Markov chains, enabling practical application in MCMC and RL. To the best of our knowledge, this is among the **first rigorous treatments** of the $m$-out-of-$n$ bootstrap in these settings.
>
> - **Correcting prior inconsistencies:** We identify and resolve inconsistencies in earlier work, including correcting the variance formula and providing guarantees in the **practically relevant regime** where $m \ll n$.
>
> #### **Practical Implications**
> This paper provides **rigorous theoretical guarantees** for the **Studentized $m$-out-of-$n$ bootstrap estimator** of sample quantiles. Initially, we provided theoretical justifications (see Section 5), which we have now complemented with **numerical simulations** (as described in our response to reviewer **gjZU**) to further reinforce the practical value of our results.
>
> We hope these clarifications help to convey the broader scope, rigor, and practical significance of our work.

---

> > ### Comment · Reviewer_NAQc · 2025-08-05
> >
> > To be clear, I have been recommending acceptance of your paper with my score of 4. I am trying to suggest how you can improve your presentation at least for a reader like me. I find the following part of your argument distracts from your overall contributions.
> >
> > - You state "practitioners routinely use the -out-of- bootstrap" but you do not provide any indication of whether it is used for quantiles or other more complicated statistics.
> > - You mention in the first paragraph of your introduction that computation is one of the motivations for m our of n boot strap. But you never make the case this is relevant for quantiles  (I don't think it is).
> > - You provide theory for quantiles. But you do not address how your theory goes beyond quantiles.

---

> ### Author Response · Authors · 2025-08-06
> **Clarification for our misunderstandings**
>
> We sincerely thank the reviewer for the positive evaluation and recommendation for acceptance. We are especially grateful for the thoughtful clarifications provided and would like to apologize for the earlier misunderstanding on our part. Below, we offer specific clarifications, along with the corresponding suggested revisions to the manuscript.
>
> - The reviewer is absolutely correct in noting that the bootstrap is widely used beyond quantile estimation. A helpful reference for practitioners on this point is the book *Statistics* by Freedman and Pisani, among others, and we will make this clear in the introduction of the revised manuscript.
>
> - We deeply appreciate the reviewer’s clarification on this point and again apologize for the prior misinterpretation. The primary issue is now clear which we believe stemmed from the wording in line 24 of the current manuscript. To address this, we propose the following revised sentence:
>
>   > "... (1) resampling the entire dataset lacks computational appeal when $n$ is large, and more importantly (2) ..."
>
>   As the reviewer rightly points out, computational gains are not the main appeal of resampling fewer than $n$; the failure of bootstrap is, and we believe this revised phrasing more clearly emphasizes the central motivation behind the $m$-out-of-$n$ bootstrap.
>
> - We also thank the reviewer for emphasizing the importance of extending the discussion beyond quantiles. We fully agree that such a perspective would benefit the reader, and in response, we will include the following exposition in the revised conclusion section
>
> >[existing] Our guarantees still assume mild smoothness and leave open the precise heavy‐tail boundary and non‐stationary processes, and adaptive variants remain compelling directions for future study. The question regarding the joint normality of the estimators also remains an important open problem. Finally, it would be desirable to derive an Edgeworth expansion of the Studentized bootstrap estimator for Markov chains. However, the theory of Edgeworth expansions on lattices for regenerating Markov chains is sparse (in particular, we could not find a usable counterpart to Proposition 12 for Markov chains). Deriving such a result is out of scope for this (already lengthy) paper, and we plan to investigate it in future work.
>
> >[existing] Another important assumption is that of regeneration. We note that the exponential moment condition in Assumption 1 is equivalent to the more classical geometric ergodicity of Markov chains (see Theorem 16.0.2 in Meyn and Tweedie [2009]). It is known that a weaker polynomial moment condition is equivalent to arithmatically mixing for Markov chains (see Bolthausen [1982]), and a corresponding result in this regime seems plausible, and warrants future investigation.
>
> >[added part] Finally, a natural extension of this work involves applying our framework to other estimators, particularly U- and M-estimators, which are commonly used in bootstrapping. Certain classes of M-estimation problems, such as shrinkage problems (see Chapter 1 in Hall [2013], which yields the median) or absolute-deviation loss functions, naturally lead to quantile estimation. In other cases, where the M-estimator depends on a function of quantiles, one may appeal to classical tools such as the delta method or CLT for M-estimators (see Theorem 5.21 in Van der Vaart [2000]) to establish asymptotic normality. That said, a comprehensive theory for these broader classes of estimators lies beyond the scope of the current work, but we view it as a compelling direction for future research.
>
>     Citations:
>     P. Hall. The bootstrap and Edgeworth expansion. Springer Science \& Business Media, 2013.
>     A. W. Van der Vaart. Asymptotic statistics, volume 3. Cambridge university press, 2000

---

### Official Review · Reviewer_kcDW · 2025-07-02

**Clarity:** 1
**Significance:** 2
**Originality:** 2
**Rating:** 4
**Confidence:** 3

**Summary:**

The authors provide asymptotic normality, Edgeworth expansion and Berry-Esseen theorems for the studentized m-out-of-n bootstrap estimate of the distribution of the estimate of the q-th quantile of a scalar distribution with either iid data or markov chain data, from a markov chain that satisfies a regenerative condition.

The results fill a gap in the theoretical literature on understanding the consistency of the m-out-of-n bootstrap for quantiles.

**Questions:**

1. How do you formally define the hat{sigma}_m^(boot) and how do you compute it in practice?
2. Why are the main theorem statements only about asymptotic normality of the bootstrap statistic? How does one translate that into a confidence interval for the population quantile mu?  If you only care about the sample quantile, then why do you need the m out of n bootstrap? Why not just compute the sample quantile?
3. Why do you only analyze the sample quantile and not a broader class of estimands? What is particular to the sample quantile in your theorems (apart from the limit variance characterization)?
4. In proposition 1, why does the statement that E[(mu_m^boot - mu_n)^2] -> \infty, imply that "the bootstrap estimator of the variance of the sample quantile is inconsistent"?? Here you just show that the bootstrap estimate does not converge to the sample estimate. Why do you mention "the variance of the sample quantile"? In this regime of the proposition, how do we know that the variance of the sample quantile does not go to infinity too?
5. What is the technical difference between your Theorem 2 and the results of Hall and Martin 1991
6. What do you mean with the jargon "parameter-free" throughout the paper. You never spell it out. Please be more specific, as parameter free can mean many different things across sub-communities.
7. Can you expand a bit more around your comment on the fact that the tsecond order term is an even polynomial on page 6 and hence this leads to parameter free two sided tests? This is a very cryptic comment.
8. What is technically different between median (for which many more results in prior works exist) and sample quantiles?
9. Shouldn't corollary 3 be about confidence intervals for mu and not mu_n?

**Ethical Concerns:**

["NO or VERY MINOR ethics concerns only"]

**Final Justification:**

The authors have addressed my concerns in their rebuttal

**Limitations:**

Yes

**Quality:**

2

**Strengths And Weaknesses:**

Strengths:
1. The authors provide rigorous finite sample guarantees for the m-out-of-n bootstrap for quantiles
2. The problem is quite fundamental and basic and has a long history. Hence, they are filling missing gaps to a well-established literature.
3. Their result is extended to non-iid data which has applications in MCMC sampling (of interest to the neurips community)

Weaknesses:
1. The paper is poorly written and uses a lot of jargon at many places without explanation and for no real reason. For instance:
1a. It is not even clear as to what you really want to estimate? Do you want to estimate the variance of hat{mu}_n? Do you want a confidence interval for $mu$? Do you want a confidence interval for $\hat{mu}_n$? (and if the latter then why? why can't you just compute hat{mu}_n)? Do you want to approximate the whole distribution of \hat{mu}_n?
1b. You have statements of the form: The studentized estimator of mu is then Tn = sqrt{n} (hat{mu}_n - mu) / hat{sigma}_n. Tn is not an estimator of mu. It even involves mu on the right hand side. So this sentence is just weird. This to the least can be called a studentized statistic (which is an oracle statistic that is not computable). Similarly, you call Tm(mu) the m-out-of-n bootstrap estimator of the sample quantile. But this is not an estimator again of the sample quantile. hat{mu}_m^{boot} is an estimator of the sample quantile and hat{mu}_n is an estimator of the population quantile.
1c. Just say that hat{mu}_n is the sample quantile and hat{mu} is the population quantile.
1d. The prior work section is too vague and sometimes irrelevant (e.g. the LLM alignment comment). This paper is solving a very specific problem in a very vast and well established literature. With the current prior work section it is very hard to judge whether the paper makes a novel contribution in the literature and exactly what it is. Why doesn't the quantile fall into the many statistics for which consistency of m-out-of-n bootstrap confidence intervals has been established (e.g. it was not at all clear what the technical comparison is of your work and that of Hall and Martin 1991. Or of the work of Stephen Lee; "on a class of m-out-of-n bootstrap confidence intervals" (JRSSB) "on  m-out-of-n bootstrapping for non-standard m-estimation with nuisance parameters).

2. The method applies only to a very particular estimand, the quantile of a scalar random variable. Typically m-out-of-n bootstrap approaches are used for much more complex objects, such as outputs of an M-estimation or model selection procedure, or other irregular estimands. It is not clear why would one use the m-out-of-n bootstrap for constructing confidence intervals for a quantile? Unless the whole point of the paper is purely computational and about being able to construct the sample quantile among n samples using only sample quantile calculations among m << n samples. If this is the case, the paper is written in a very confusing manner and this point does not come across at all.

3. Even though the authors claim that their analysis has implications for MCMC sampling, no realistic implication was provided and no implementation and experiments on how their results have practical consequences was given.

---

> ### Author Rebuttal · Authors · 2025-07-26
>
> We thank the reviewer for their careful review and meticulous assessment. We provide responses to their questions and weaknesses as below.
>
> ## Weaknesses
>
> 1a. We refer to Dasgupta [2008, Ch. 29], which explains the idea of using replicated samples to estimate the distribution of a statistic. Since we typically get only one dataset, bootstrap samples from the empirical CDF serve as proxies for such replicates. Revisiting the paper, we agree the need to make this immediately apparent in the introduction, and believe it can be achieved by rewording line 22 to be more specific, as
>
> > The bootstrap—first proposed by Efron [1979]—quickly became the go-to method for sampling from, and testing about, the distribution of a statistic.
>
> 1b.  We will reword  $T_n$ to Studentized ``statistic" in the revision.
>
> 1c. We would like to gently point out that the general consensus in Statistics is Greek alphabets (like $\mu,\sigma$ etc.) stand for parameters and their hatted version (like $\hat \mu,\hat\sigma$ etc.) stand for the estimates.
>
> 1d. We respectfully disagree with the assertion that the manuscript is “poorly written”. Nevertheless, we will revise the literature review to remove any residual vague statements. We gently remind the reviewer that the development of the m-out-of-n bootstrap leading to our work is outlined comprehensively in the Introduction (lines 22, 26, 46, 49, 50). The limitations of earlier work, motivating our results in the i.i.d. setting and their subsequent extension to Markov chains are discussed in lines 53–62. Additionally, the shortcomings of the most directly relevant prior work are addressed in lines 63–66 and further elaborated in Section 4.2. We address the limitations in our current work in section 6 (see also the revision according to suggestions of reviewer gjZu). We would like to respectfully highlight the fact that reviewer gHZF highlighted the paper’s clarity as a strength, stating:
>
> >The comparison to previous literature is very precise, and I found it very clear how this work compared with previous papers in this space.
>
> That said, we would be glad to consider any specific suggestions for improvement during the discussion period. For the other part, please see the response to Question 5.
>
>  **On M-estimation:** The benefits of $m$-out-of-$n$ has by now been well studied in the literature, and we would like to point the reviewer towards the citations in line 107 of our paper (particularly highlighting Chapter 22 in Dasgupta [2008]). We divide the rest of our responses up into two distinct parts:
>
> The M-estimator is indeed an interesting use case for bootstrapping, however we would also like to point out that the full generality of this problem is outside the scope of the current work. That being said, there are many M-estimators which end up producing quantiles when maximised like $L_2$-shrinkage problems [Chapter 1, Hall, 2013] (which leads to the median) or absolute-deviation loss functions. On the other hand, as the reviewer suggests, if the M-estimator depends on some function of the quantiles one would use existing theory (like delta-method, or CLT for M-estimates [Theorem 5.21, Van der Vaart, 1998]) to derive asymptotic normality. While the most general cases are interesting in their own rights, we found that even this basic case (Studentized quantiles) lacked rigorous theoretical guarantees, something which this paper aims to address.
>
> For the second question, we take this opportunity to gently remind the reviewer that our objective in this paper (as we mentioned in line 116, line 142, etc.) was to attack one specific corner of this problem which lay largely unanswered: the theoretical properties of the **Studentized** bootstrapping of the sample quantiles for IID and regenerative Markov chains. While we agree that it is also interesting in the context of M-estimation, we would like to point out that testing for the quantile is quite prominent even in "banal" statistical tasks like MCMC, where one might want to test for the sample median of an MCMC algorithm. But even beyond quantile estimation, $m$-out-of-$n$ bootstrapping is almost ubiquitous in statistical learning and features prominently in tasks bootstrap aggregating typically uses $\sqrt{n}$ many resamples when training the smaller models (Chapter 8, Hastie et al. [2009]).
>
>       Hastie, T., Tibshirani, R., Friedman, J. H., \& Friedman, J. H. (2009). The elements of statistical learning: data mining, inference, and prediction (Vol. 2, pp. 1-758). New York: springer.
>       P. Hall. The bootstrap and Edgeworth expansion. Springer Science \& Business Media, 2013
>
> **Experiments:** Regarding the first point we would like to remind the reviewer that we provide theoretical justifications for resampling of MCMC in section 5, and for the second point we hope that the reviewer finds our response to reviewer gjZu satisfactory. Having said that, we would also like to underline the fact that all our theory is **exact** and free from unknowable constants, in the sense that the asymptotic distributions are free of unknowable parameters, and first order expansions are precise. Furthermore, we theoretically demonstrate the reasonableness of our assumptions with examples in Section 5. Therefore, especially when considering the current length of the paper, we believe that simulations would not be significantly more illuminating.
>
> ## Questions
>
> 1.  The complete derivation of the closed-form expression is already provided in Supplementary Section B, and its location is explicitly flagged by a footnote on page 2 (line 39). In view of this clear cross-reference, the material should have been readily accessible. Nevertheless, to preempt any further oversight, we will replace the footnote with a prominent remark in the main text, immediately above line 40, in the revised manuscript.
>
> 2.  We hope our response to Weakness 1a has already addressed this concern in part. To clarify further, the primary focus of this paper is on the behavior of bootstrap statistics. Under the standard assumption that sample quantiles are consistent for their population counterparts, confidence intervals can be derived as a straightforward consequence of Slutsky’s theorem.
>
> 3.  We appreciate the reviewer’s interest in broader estimands and hope our response regarding M-estimation in the weaknesses section was helpful. We also refer the reviewer to our response to Comment 1b by reviewer NAQc for additional context. While we acknowledge the importance of extending the analysis to other settings, we believe such generalizations fall outside the scope of this work, which is already substantial in length and technical depth.
>
> 4.  We would like to clarify that the purpose of the counterexample was to justify the moment condition imposed on the random variables. This implication follows directly by definition (see also our response to the first question). Additionally, we note that the counterexample demonstrates divergence of the error variance, a stronger result than simply showing that the bootstrap estimate fails to converge to the sample estimate.
>
> 5.  We would like to respond in two parts:
>
>    **(a)** Hall and Martin’s 1991 paper concerns the classical (orthodox) bootstrap, whereas our work focuses on the \( m \)-out-of-\( n \) bootstrap.
>
>    **(b)** Since \( m < n \), we are able to apply a convenient continuity correction. This, in turn, allows us to leverage more modern results on Edgeworth expansions for polygonal approximants (see Lemma 9 at line 1075 and its proof at line 1106), rather than deriving the expansion from first principles as Hall and Martin did through substantial effort. A further implication of this smoothing is that the Studentized \( m \)-out-of-\( n \) bootstrap estimator of the sample quantiles admits an Edgeworth expansion with an even first polynomial, a property not shared by the classical bootstrap, where the first polynomial is neither even nor odd.
>
> 6.  We agree with the reviewer and will revise line 72 to enhance clarity. In the updated draft, we will rephrase the sentence as:
>
> > “... deriving asymptotic distributions free of unknown parameters (parameter-free) for ...”
>
> 7.  We hope our response to Question 6 has addressed the first part. Regarding the second part, because the polynomial in our Edgeworth expansion is even, it follows that $\( P(T_m^{(boot)} > t) \)$ agrees with $\( P(T_m^{(boot)} < -t) \)$ up to a $\( 1/m \)$ term. This symmetry does not hold for the classical bootstrap, where the leading polynomial is neither odd nor even. Prior literature (e.g., Hall and Martin, Van der Vaart) has noted the incompatibility of such expansions with two-sided tests, a limitation that the \( m \)-out-of-\( n \) bootstrap appears to avoid. We also point the reviewer to the technical justification provided in our response to Question 5b.
>
> 8.  This is an excellent question. We note that our assumption that $\( F \in \mathbb{S}_1(\mu) \)$ implicitly excludes extreme quantiles such as the maximum or minimum. These fall within the so-called “extreme” regime, which is inherently more challenging than the central regime (e.g., the median); see Dasgupta [2008, p. 91] for further discussion. However, we emphasize that no prior work has addressed Studentization for the \( m \)-out-of-\( n \) bootstrap estimator of sample quantiles in the practical regime $\( m \ll n \)$. Thus, our contributions focus not on extending results from the median to other quantiles, but on developing new theory for Studentized quantiles, specifically including the median. We hope this helps clarify our intent.
>
> 9.  Since Corollary 3 establishes the asymptotic distribution of $\( T_m^{(boot)} \)$, we believe it is stated correctly as is.
>
> We would be happy to answer any further questions in the discussions
>
> **_We hope we have addressed all the questions, and kindly request you to revise your score._**

---

> > ### Comment · Area_Chair_9t2q · 2025-08-07
> >
> > Dear Reviewer kcDW,
> >
> > Please read the author's answer and comment on which of your concerns were mitigated and which remain (if any). We have one more day left for these discussions and your participation is crucial (and mandatory).

---

> > > ### Comment · Reviewer_kcDW · 2025-08-09
> > >
> > > The authors have satisfactorily addressed my concerns and questions in their rebuttal

---

### Official Review · Reviewer_nMYW · 2025-07-03

**Clarity:** 4
**Significance:** 3
**Originality:** 3
**Rating:** 5
**Confidence:** 4

**Summary:**

The paper derives convergence guarantees for the m-out-of-n bootstrap estimator by developing a second–order theory for Studentized m-out-of-n bootstrap quantiles under relatively weak moment assumptions. Specifically, they prove variance consistency, a central-limit theorem, an exact O(m^{−1/2}) Edgeworth expansion that delivers exact convergence rates, and a matching Berry–Esseen bound on the bootstrap approximation error of bootstrap quantiles. As a by-product, their variance formula correct an error of Cheung and Lee [2005]. Finally, they extend the analysis to regenerative Markov chains.

**Questions:**

I have no questions for the authors.

**Ethical Concerns:**

["NO or VERY MINOR ethics concerns only"]

**Final Justification:**

I have read the authors extensive replies for all the reviewers and feel very confident in keeping my score.

**Limitations:**

The authors adequately addressed the limitations and potential negative societal impact of their work.

**Paper Formatting Concerns:**

There are no major formatting issues in this paper.

**Quality:**

3

**Strengths And Weaknesses:**

Strengths: The authors develop the asymptotic theory of the m-out-of-n bootstrap quantile estimator, which is a broadly used method that somehow lacks rigorous soundness guarantees. They also extend beyond the i.i.d. case to regenerative Markov chains and provide useful applications. The presentation is clear and the proof sketches are intuitive and capture the gist of the proofs.
Weaknesses: There are some mild assumptions and only stationary process are considered, but I don't consider this a problem but an intersting direction for future works.

---

> ### Author Rebuttal · Authors · 2025-07-25
>
> We deeply appreciate your positive review and your accurate summarization. We hope that the rigorous theoretical guarantees, the correction of key prior work, and the clear exposition across both i.i.d. and regenerative Markov settings underscore the paper’s strong impact and align with the highest possible recommendation.

---

> > ### Comment · Reviewer_nMYW · 2025-08-05
> >
> > I have read the authors extensive replies for all the reviewers and feel very confident in keeping my score.

---

> > > ### Author Response · Authors · 2025-08-05
> > > **Thank you for your positive feedback**
> > >
> > > Thank you for carefully reviewing our responses; we are delighted that you found our responses in the rebuttals useful and we appreciate your continued confidence in our work.

---

### Note · Authors · 2025-08-12

We take this opportunity to once again thank the reviewers for their thorough review and are grateful for the various useful comments which substantially improve the quality and flow of the paper.

We have incorporated all recommended suggestions in the revised manuscript, in particular, correcting all the typos; including the full versions of all acronyms; setting the tone towards previous literature; streamlining the expositions, and the inclusion of requisite definitions. The expository changes are crucial but cosmetic, and fits in the main draft, while the experiments have been deferred to the supplements. We thank the reviewers for their positive reviews, and illuminating discussions, and hope that our paper gets their highest recommendation.

---

### Decision · Program_Chairs · 2025-09-17

**Decision:**

Accept (poster)

**Comment:**

This paper develops a framework for resampling with small sample sizes, yielding sharp non-asymptotic guarantees. The results are mathematically careful, the presentation is clear, and the reviewers consistently judged the technical quality to be high.

The strengths highlighted were rigor and clarity. The main reservation is that the contribution feels narrow and not especially exciting from the perspective of broader NeurIPS impact. Reviewers nevertheless felt that the contribution was valuable, emphasizing that the sharpness of the guarantees advances the state of the art. In their rebuttal, the authors clarified aspects of positioning and provided further examples, which the reviewers found adequate.

That said, the reviewers were positive, finding the work correct, solid, and of value to the statistical theory community. Given the strength of their assessments, I follow their recommendation to accept.